# Measurement report: Vertical and temporal variability in the near-surface ozone production rate and sensitivity in an urban area in the Pearl River Delta region, China

**Jun Zhou**[1,2,★], **Chunsheng Zhang**[3,★], **Aiming Liu**[3], **Bin Yuan**[1,2], **Yan Wang**[1,2], **Wenjie Wang**[1,4], **Jie-Ping Zhou**[1,2], **Yixin Hao**[1,2], **Xiao-Bing Li**[1,2], **Xianjun He**[1,2], **Xin Song**[1,2], **Yubin Chen**[1,2], **Suxia Yang**[1,2], **Shuchun Yang**[1,2], **Yanfeng Wu**[1,2], **Bin Jiang**[1,2], **Shan Huang**[1,2], **Junwen Liu**[1,2], **Yuwen Peng**[1,2], **Jipeng Qi**[1,2], **Minhui Deng**[1,2], **Bowen Zhong**[1,2], **Yibo Huangfu**[1,2], **and Min Shao**[1,2]

[1]Institute for Environmental and Climate Research, Jinan University, Guangzhou, 511443, China
[2]Guangdong–Hong Kong–Macau Joint Laboratory of Collaborative Innovation for Environmental Quality, Guangzhou, 511443, China
[3]Shenzhen National Climate Observatory, Shenzhen, 518040, China
[4]Multiphase Chemistry Department, Max Planck Institute for Chemistry, 55128 Mainz, Germany
★These authors contributed equally to this work.

**Correspondence:** Bin Yuan (byuan@jnu.edu.cn), Xiao-Bing Li (lixiaobing@jnu.edu.cn), and Min Shao (mshao@jnu.edu.cn)

**Abstract.** Understanding the near-ground vertical and temporal photochemical $O_3$ formation mechanism is important to mitigate $O_3$ pollution. Here, we measured the vertical profiles of $O_3$ and its precursors at six different heights, ranging from 5 to 335 m, using a newly built vertical observation system in the Pearl River Delta (PRD) region of China. The net photochemical ozone production rate ($P(O_3)_{net}$) and $O_3$ formation sensitivities at various heights were diagnosed using an observation-based model coupled with the Master Chemical Mechanism (MCM v3.3.1). Moreover, to assess model performance and identify the causative factors behind $O_3$ pollution episodes, the $P(O_3)_{net}$ was measured at 5 m above ground level with a custom-built detection system. In total, three $O_3$ pollution episodes and two non-episodes were captured. The identified $O_3$ pollution episodes were found to be jointly influenced by both photochemical production and physical transport, with local photochemical reactions playing a major role. The high index of agreement (IOA) calculated by comparing the modelled and measured $P(O_3)_{net}$ values indicated the rationality of investigating the vertical and temporal variability in $O_3$ formation mechanisms using model results. However, the measured $P(O_3)_{net}$ values were generally higher than the modelled $P(O_3)_{net}$ values, particularly under high-$NO_x$ conditions, which may indicate a potential underestimation of total $RO_2$ by the model. Throughout the measurement period, the contribution of different reaction pathways to $O_3$ production remained consistent across various heights, with $HO_2 + NO$ as the major $O_3$ production pathway, followed by $RO_2 + NO$. We observed that $P(O_3)_{net}$ decreased with an increase in measurement height, which was primarily attributed to the reduction in $O_3$ precursors, such as oxygenated volatile organic compounds (OVOCs) and non-methane hydrocarbons (NMHCs). The $O_3$ formation regimes were similar at different heights during both episodes and non-episodes, either being located in the VOC-sensitive regime or in the transition regime that is more sensitive to VOCs. Diurnally, photochemical $O_3$ formation typically remained in the VOC-sensitive regime during the morning and noon, but it transitioned to the transition regime and was more sensitive to VOCs in the afternoon at around 16:00 LT (local time). Vertical and temporal photochemical $O_3$ formation is most sensitive to OVOCs, suggesting that targeting specific VOCs for control measures is more practical and feasible at the observation site. The vertical temporal analysis of $O_3$ formation mechanisms near

the ground surface in this study provides critical foundational knowledge that can be used to formulate effective short-term emergency and long-term control strategies to combat $O_3$ pollution in the PRD region of China.

## 1   Introduction

Tropospheric ozone ($O_3$), which has adverse effects on ecosystems, climate change, and human health (Fiore et al., 2009; Anenberg et al., 2012; Seinfeld and Pandis, 2016), has become an important factor resulting in severe regional air pollution in China (Zhu et al., 2020). Tropospheric $O_3$ mainly comes from stratospheric intrusions and the photochemical reactions of $O_3$ precursors, involving volatile organic compounds (VOCs) and nitrogen oxides ($NO_x = NO + NO_2$). The $O_3$-precursor relationship can be split into "$NO_x$-limited", "VOC-limited", or "mixed-sensitivity" regimes (Seinfeld and Pandis, 2016; Sillman, 1999). An $NO_x$-limited regime has higher VOC / $NO_x$ ratios and the $O_3$ formation is sensitive to $NO_x$ concentration changes, whereas a VOC-limited regime has lower VOC / $NO_x$ ratios and the $O_3$ formation decreases with increasing $NO_x$ and increases with increasing VOCs. In a mixed-sensitivity regime, $O_3$ formation responds positively to changes in both $NO_x$ and VOC emissions (P. Wang et al., 2019). Local $O_3$ concentrations can be further influenced by meteorological conditions and the regional transport of $O_3$ and its precursors (Gong and Liao, 2019; Chang et al., 2019). The Pearl River Delta (PRD) stands out as one of the most rapidly developing economic and urbanized regions in China, which currently is suffering from severe ground-level $O_3$ pollution (Lu et al., 2018; Yang et al., 2019). To date, many scholars have analysed the relationship of tropospheric $O_3$ pollution and its precursors with meteorological elements in the PRD region, and results show that the surface $O_3$ pollution is determined by both local photochemistry and physical transport, with long-range transport contributing 30 %–70 % to surface $O_3$ concentrations (Mao et al., 2022; Shen et al., 2021; Li et al., 2012, 2013). However, the distribution of $O_3$ is highly variable at different altitudes (Wang et al., 2021), due to vertical differences in VOC concentrations and sources, as is the sensitivity of $O_3$ formation (Liu et al., 2023; Tang et al., 2017). Due to the presence of strong vertical mixing, driven by the surface heating effect in the daytime boundary layer, the $O_3$ budget at ground level as well as at an arbitrary height in the daytime boundary layer is closely related to the formation and removal of $O_3$ at other heights (Tang et al., 2017). In addition, the difference in vertical gradients of precursors may drive the vertical change in the photochemical formation regimes of $O_3$ (Zhao et al., 2019). Using data from only one height to understand the photochemical reactions in the planetary boundary layer imposes a great limitation. Thus, diagnosing the $O_3$ formation mechanism at

different heights is essential to achieve effective control of $O_3$ pollution.

To date, remote sensing techniques with a high time resolution and real-time response, such as lidar and optical absorption spectroscopy, have been utilized to measure the vertical distribution of $O_3$ (Luo et al., 2020a; Wang et al., 2021). However, in situ measurements of VOCs at various heights primarily rely on offline methods combined with diverse techniques, including aircraft, tethered balloons, tall buildings and towers, uncrewed aerial vehicles (UAVs or drones), and satellite observations (Klein et al., 2019; Li et al., 2022; Geng et al., 2020; Benish et al., 2020; Li et al., 2021; N. Wang et al., 2019). Owing to the low time resolution of these monitoring techniques, achieving continuous vertical coverage of VOC and $NO_x$ measurements is challenging. Consequently, the vertical distribution of VOCs remains unclear, thus largely hindering our understanding of the vertical and temporal regional $O_3$ formation mechanism.

To fill the gaps in the existing studies, we utilized a newly constructed vertical observation system based on the Shenzhen Meteorological Gradient Tower (SZMGT) (Li et al., 2023). This system measured the vertical profiles of $O_3$ and its precursors at six different heights from 5 to 335 m. To diagnose the net $O_3$ production rate, $P(O_3)_{net}$, and $O_3$ formation sensitivities across various heights, we employed an observation-based model coupled with the Master Chemical Mechanism (MCM v3.3.1), referred to as OBM-MCM in the following. Additionally, we employed a novel net photochemical $O_3$ production rate (NPOPR), $P(O_3)_{net}$, detection system to measure $P(O_3)_{net}$ at 5 m above ground level to explore potential reasons for $O_3$ pollution episodes (Hao et al., 2023), i.e. to examine the contribution of chemical and physical processes to changes in the $O_3$ concentration. Comparisons between the directly measured $P(O_3)_{net}$ results and the model-derived data enabled us to evaluate the simulation accuracy and explore potential reasons for discrepancies in the OBM-MCM model with respect to photochemical $O_3$ formation. Based on these results, we have extensively discussed the vertical and temporal variability in $P(O_3)_{net}$ and $O_3$ formation sensitivity while also acknowledging potential biases associated with the modelling. The findings of this study offer a new benchmark for understanding the vertical profile of the photochemical $O_3$ formation mechanism, aiding in the identification of the primary driver of ground-level $O_3$ pollution. This identification is crucial, as it can provide essential theoretical support for the development of short-term effective emergency and long-term control measures targeting $O_3$ in the PRD region of China.

## 2 Materials and methods

### 2.1 Sampling site

Field measurements were conducted at the Shenzhen Meteorological Gradient Tower (SZMGT; 22.65° N, 113.89° E) from 13 November to 10 December 2021. The SZMGT is 365 m high and is currently the tallest mast tower in Asia and the second tallest of its kind in the world. The main structure of the tower is made of steel, and steel stray lines are used to fix and secure the tower. It is located in the Tiegang Reservoir water reserve in the Bao'an District of Shenzhen, Pearl River Delta (PRD), China. The area is surrounded by a high density of vegetation, first-class water resource protection area, low-rise buildings, and hills and mountains (Luo et al., 2020b).

### 2.2 Instrumentation

#### 2.2.1 The vertical sampling system

A tower-based observation system for trace gases using long perfluoroalkoxy alkane (PFA) tubes (outer diameter, o.d., of 0.5 in.) was used to sample the $O_3$ and $O_3$ precursors at six heights during the campaign, including 5, 40, 70, 120, 220, and 335 m above the ground. Ambient air was continuously drawn into all six tubes using a rotary vane vacuum pump to constantly flush the tubes and reduce the tube delay of the organic compounds; the flow rates in each tube were controlled by critical orifices (orifice diameter: 0.063 in.). A Teflon solenoid valve group was used to switch the air samples at specified time intervals so that the subsamples from these six heights could be sequentially drawn by the instruments (see Fig. S1 in the Supplement). Consequently, the flow rates of the air sample streams for the six tubes varied between 12.0 and 15.0 slpm (standard litres per minute) without subsampling and were less than 20 slpm with subsampling. The residence time of the sample gas in the longest tube (∼ 400 m) was less than 180 s at a flow rate of 13 slpm. The impacts of long tubing on the measurements of various of trace gases, including $O_3$, $NO_x$, and a set of organic compounds, were systematically investigated using a combination of laboratory tests, field experiments, and modelling techniques. Field observations proved that this observation system is suitable for analysing spatiotemporal variations in atmospheric trace gases, and many trace gases could be well measured. More details about the establishment and the characterization of this observation system are described elsewhere (Li et al., 2023).

#### 2.2.2 $P(O_3)_{net}$ measurement

During the campaign, $P(O_3)_{net}$ at 5 m above ground level was measured using the home-made NPOPR detection system, which was built based on the dual-channel reaction chamber technique. The improvement, characterization, and photochemical $O_3$ formation mechanism in the reaction and reference chambers of the NPOPR detection system have been described in our previous study (Hao et al., 2023). Briefly, the NPOPR detection system consists of quartz glass reaction and reference chambers with the same geometry. The length and inner diameter of the quartz glass cylinder are 700 and 190.5 mm, respectively, resulting in an inner volume of ∼ 20 L. The outer surface of the reference chamber was covered with an Ultem film (SH2CLAR, 3 M, Japan) for ultraviolet (UV) protection, which can block sunlight with wavelengths < 390 nm, thus preventing photochemical reactions inside. During the experiment, both the reaction and reference chambers were placed outdoors and directly exposed to sunlight to simulate real ambient photochemical reactions. Ambient air was introduced into the reaction and reference chambers at the same flow rate, and a Teflon filter was mounted before the chamber inlet to remove fine particles. To correct for the effect of fresh NO titration on $O_3$, we use $O_x$ ($= O_3 + NO_2$) instead of $O_3$ to quantify the $O_3$ generated by photochemical reactions (Pan et al., 2015; Tan et al., 2018). A stream of air from the two chambers was alternately introduced into an NO-reaction chamber every 2 min to convert $O_3$ in the air to $NO_2$ in the presence of high concentrations of NO ($O_3 + NO \rightarrow NO_2$), and the $O_x$ concentrations from the outlet NO-reaction chamber, i.e. the total $NO_2$ concentrations including the inherent $NO_2$ in the ambient and that converted from $O_3$, were measured by a cavity attenuated phase shift (CAPS) $NO_2$ monitor (Aerodyne Research, Inc., Billerica MA, USA) to avoid other nitrogen oxide interferences with respect to the $NO_2$ measurement (e.g. alkyl nitrates, peroxyacyl nitrates, peroxynitric acid, and nitrogen pentoxide). $P(O_3)_{net}$ was obtained by dividing the difference between the $O_x$ concentrations in the reaction and reference chambers ($\Delta O_x$) by the mean residence time of air in the reaction chamber $\langle \tau \rangle$:

$$P(O_3)_{net} = P(O_x)_{net} = \frac{\Delta O_x}{\tau}$$
$$= \frac{[O_x]_{reaction} - [O_x]_{reference}}{\tau}. \tag{1}$$

A schematic of the NPOPR detection system is shown in Fig. S2. The pulse experiments were performed to quantify the residence time in the chambers (Hao et al., 2023).

The $[O_x]$ values plugged into Eq. (1) to derive $P(O_3)_{net}$ are measured values corrected for wall losses of $O_x$ and the light-enhanced loss of $O_3$ ($d[O_3]$) in the reaction and reference chambers during daytime (Hao et al., 2013):

$$\gamma = \frac{d[O_3] \times D}{\omega \times [O_3] \times \tau}, \tag{2}$$

where $\gamma$ is the light-enhanced loss coefficient of $O_3$, which is derived from $J(O^1D)$ according to the relationship obtained from the outdoor experiments (for more details, see Sect. S3 in the Supplement); $d[O_3]$ represents the difference between

the $O_3$ mixing ratios at the inlet and outlet of the reaction and reference chambers; $D$ is the diameter of the chambers; $\omega$ is the average velocity of $O_3$ molecules; $[O_3]$ is the injected $O_3$ mixing ratio at the inlet of the reaction and reference chambers; and $\tau$ is the average residence time of the air in the reaction and reference chambers. When quantifying the light-enhanced $O_3$ loss (d$[O_3]$) during the ambient air measurement, we first calculate $\gamma$ using the measured $J(O^1D)$ and the $\gamma - J(O^1D)$ equations listed in Fig. S8 in the reaction and reference chambers; we then use the measured $[O_3]$ and Eq. (2) to calculate d$[O_3]$. The results show that such a correction can increase the measured $P(O_3)_{net}$ by 10 % (25 % percentile) to 24 % (75 % percentile), with a median of 17 %.

The limit of detection (LOD) of the NPOPR detection system is 2.3 ppbv h$^{-1}$ at the sampling air flow rate of 5 L min$^{-1}$, which is obtained as 3 times the measurement error of $P(O_3)_{net}$ (Hao et al., 2023). The measurement error of $P(O_3)_{net}$ is determined by the estimation error of $O_x$ in the reaction and reference chambers, which includes measurement error associated with the $O_x$ of the CAPS-$NO_2$ monitor and the error due to the light-enhanced loss of $O_3$. This collective measurement error is referred to as the measurement precision of the NPOPR detection system, with further details provided in the Supplement, specifically in Sect. S4. The measurement accuracy of the NPOPR detection system is determined to be 13.9 %, representing the maximum systematic error resulting from photochemical $O_3$ production in the reference chamber. Our earlier research indicated that the modelled $P(O_3)_{net}$ in the reaction chamber is similar to that modelled in ambient air, with the modelled $P(O_3)_{net}$ in the reference chamber accounting for 0 %– 13.9 % of that in the reaction chamber (Hao et al., 2023). This is due to the UV protection (Ultem film) cover on the reference chamber, which only filtered out the sunlight with wavelengths $< 390$ nm, allowing photochemical $O_3$ production to persist at the sunlight wavelengths of between 390 and 790 nm. Here, we have utilized the same modelling approach described in Hao et al. (2023) to quantify the $P(O_3)_{net}$ in the reference chamber, and we have corrected for the bias introduced by the measurement accuracy.

### 2.2.3   VOC measurement

VOCs were measured using a high-resolution proton-transfer-reaction time-of-flight mass spectrometer (PTR-TOF-MS; IONICON Analytik, Austria) (Wang et al., 2020 ; Wu et al., 2020) and an off-line gas chromatography mass spectrometry flame ionization detector (GC-MS-FID; Wuhan Tianlong, Co. Ltd, China) (Yuan et al., 2012). The concentrations of oxygenated VOCs (OVOCs), including formaldehyde (HCHO) and acetaldehyde (CH$_3$CHO), were measured via PTR-TOF-MS, while the non-methane hydrocarbons (NMHCs) were measured via GC-MS-FID. The PTR-TOF-MS was run in both the hydronium ion ($H_3O^+$) (Yuan et al., 2017; Wu et al., 2020) and nitric oxide ion ($NO^+$) (Wang

et al., 2020) modes. The measurement error of the PTR-TOF-MS was lower than 20 %; more details on the PTR-TOF-MS technique can be found in our previous publication (Yuan et al., 2017). The $H_3O^+$ and $NO^+$ modes were automatically switched as follows: 20 min $H_3O^+$ mode and 10 min $NO^+$ mode. The background signal of each mode was measured every 30 min for at least 2 min by automatically switching the ambient measurement to a custom-built platinum catalytic converter heated to 365 °C. Operating the PTR-TOF-MS instrument in $NO^+$ mode primarily detects higher alkanes, which are known to significantly contribute to the formation of secondary organic aerosols (SOAs) but have negligible contributions to photochemical $O_3$ formation (Wang et al., 2020). Eventually, we only used VOCs measured during the $H_3O^+$ mode, which was operated at a drift tube pressure of 3.8 mbar, a temperature of 120 °C, and a voltage of 760 V, resulting in an $E/N$ (where $E$ refers to the electric field and $N$ refers to the number density of the buffer gas in the drift tube) value of $\sim 120$ Td (where Td denotes townsend). A total of 3035 ions with $m/z$ up to 510 were obtained at time resolutions of 10 s. A gas standard with 35 VOC species was used for calibrations of the PTR-TOF-MS once per day. Raw data from the PTR-TOF-MS were analysed using the Tofware software (TOFWERK AG, v3.0.3). Due to the humidity dependencies of various VOC signals of the PTR-TOF-MS observed in laboratory studies, such as formaldehyde, benzene, methanol, ethanol, and furan (Wu et al., 2020), we determined their humidity-dependence curves. During data analysis, we removed the impacts of ambient humidity change on the measured signals of the PTR-TOF-MS according to these humidity-dependence curves. For the off-line GC-MS-FID measurement, whole-air samples were collected using 3.2 L electro-polished stainless-steel canisters (Entech, USA) at 5 and 120 m with measurement time intervals of 2 h. Two automatic canister samplers connected to 12 canisters were used to collect the whole-air samples, with each of canister collecting the sample for 10 min. The canisters were analysed within 1 week (Zhu et al., 2018). The concentrations of 56 NMHC species in the canister were analysed using a GC-MS-FID that was calibrated daily using a mixture of a Photochemical Assessment Monitoring Stations (PAMS) standard gas and pure $N_2$. In addition, the mixture of PAMS standard gas and pure $N_2$ with species concentrations of 1 ppbv was injected into the analytical system every 10 samples to check the operational stability of the instrument. Pure $N_2$ was injected into the analytical system at the start and end of each day's analysis to provide reference blank measurements. A full list of all 56 NMHCs can be found in the Supplement (Table S2).

### 2.2.4   Other parameters

The photolysis frequencies of different species were measured using an actinic flux spectrometer (PFS-100, Focused Photonics Inc, China). The $O_3$, CO, and $NO_x$ concentra-

tions were measured by a 2B O$_3$ monitor based on dual-channel UV absorption (Model 205, 2B Technologies, USA), a gas filter correlation (GFC) CO analyser (Model 48i, Thermo Fisher Scientific, USA), and a chemiluminescence NO$_x$ monitor (Model 42i, Thermo Fisher Scientific, USA), respectively. According to our test (Zhou et al., 2025), a 5 % overestimation could be caused in the NO$_2$ measurement using the chemiluminescence technique compared with the CAPS technique, due to some NO$_Z$ species (i.e. HNO$_3$; peroxyacetyl nitrates – PANs; HONO; etc.) (Dunlea et al., 2007); this will result in a $< 4$ % decrease in the modelled $P(O_3)_{net}$, which is negligible compared to the bias caused by the $P(O_3)_{net}$ in the reference chamber ($\sim 14$ %) (Zhou et al., 2023). Temperature ($T$), relative humidity (RH), and pressure ($P$) were measured by a portable weather station (MetPak, Gill Instruments Ltd, UK).

## 2.3 Data analysis

### 2.3.1 Observation-based chemical box model

We investigated the detailed photochemical O$_3$ formation mechanism during the observation period based on the observed field data. The specific tropospheric O$_3$ photochemical formation process involves the photolysis of NO$_2$ at $< 420$ nm (Sadanaga et al., 2017). Simultaneously, RO$_x$ (RO$_x$ = OH + HO$_2$ + RO$_2$) radical cycles provide HO$_2$ and RO$_2$ to oxidize NO to NO$_2$, resulting in the accumulation of O$_3$ (Shen et al., 2021; Cazorla and Brune, 2010; Sadanaga et al., 2017). Therefore, the RO$_x$ radicals and the O$_3$, OH, and NO$_3$ oxidants play important roles in photochemical O$_3$ formation. A 0-D box model based on the Framework for 0-D Atmospheric Modelling (F0AM) v3.2 (Wolfe et al., 2016) coupled with the MCM v3.3.1 was used to simulate the $P(O_3)_{net}$. MCM v3.3.1 contains a total of 143 VOCs, more than 6700 species, involving more than 17 000 reactions (Jenkin et al., 2015). The $P(O_3)_{net}$ and O$_3$ concentrations were simulated by constraining $T$, RH, $P$, and organic and inorganic substances in gases, including 12 OVOCs (methanol, ethanol, formaldehyde, acetaldehyde, acrolein, acetone, hydroxyacetone, phenol, $m$-cresol, methyl vinyl ketone, methacrylaldehyde, and methyl ethyl ketone), 56 NMHCs (toluene, benzene, isoprene, styrene, etc., as listed in Table S2), inorganic gaseous pollutants (O$_3$, NO, NO$_2$, and CO), and photolysis rate values ($J(O^1D)$, $J(NO_2)$, $J(H_2O_2)$, $J(HONO)$, $J(HCHO\_M)$, $J(HCHO\_R)$, $J(NO_3\_M)$, $J(NO_3\_R)$, etc.). The VOCs, NO$_x$, $T$, RH, and $P$ were constrained throughout the modelling period, whereas O$_3$ was not constrained after providing initial concentration values. To avoid the build-up of long-lived species to unreasonable levels, we also considered the physical dilution process by setting a constant dilution factor of $1 / 43\,200$ s$^{-1}$ throughout the modelling period (J. Liu et al., 2021; Decker et al., 2019). Additionally, the dry deposition rate of O$_3$ was set to 0.42 cm s$^{-1}$, while the background val-

ues of O$_3$, CO, and CH$_4$ were set to 30, 70, and 1800 ppbv, respectively, based on the findings of Wang et al. (2011, 2022a) and WMO (2022). The model was run in a time-dependent mode with a resolution of 5 min, and it was run for a spin-up time of 72 h to establish steady-state concentrations for secondary pollutants that were not constrained during the simulation. The $P(O_3)_{net}$ can be expressed by the difference between the O$_3$ production rate ($P(O_3)$) and O$_3$ destruction rate ($D(O_3)$), where $P(O_3)$ and $D(O_3)$ can be calculated as follows:

$$P(O_3) = k_{HO_2+NO}[HO_2][NO] + \sum_i k_{RO_{2,i}+NO}[RO_{2,i}][NO]\varphi_i, \tag{3}$$

$$D(O_3) = k_{O(^1D)+H_2O}\left[O\left(^1D\right)\right][H_2O] + k_{OH+O_3}[OH][O_3] + k_{HO_2+O_3}[HO_2][O_3] + k_{O_3+alkenes}[O_3][alkenes] + k_{OH+NO_2}[OH][NO_2] + k_{RO_{2,i}+NO_2}[RO_{2,i}][NO_2]. \tag{4}$$

Here, $k_{M+N}$ represents the bimolecular reaction rate constant of $M$ and $N$, the subscript $i$ refers to different types of RO$_2$, and $\varphi_i$ is the yield of NO$_2$ of the reaction RO$_{2i}$ + NO. The relevant reaction rates of $P(O_3)$ and $D(O_3)$ and the mean measured concentrations of each VOC category at 5 m above ground level during O$_3$ episodes and non-episodes used in the model are listed in Tables S1 and S2.

### 2.3.2 Derivation of the contribution of chemical and physical processes to O$_3$ changes at ground level

It is known that chemical and physical processes jointly influence the O$_3$ concentration changes near the ground surface (Xue et al., 2014; Tan et al., 2019). The direct measurement of $P(O_3)_{net}$ gave us a chance to identify the contribution of chemical and physical processes to the variation in the observed O$_3$ concentrations using the following equation:

$$\frac{dO_x}{dt} = P(O_x)_{net} + R(O_x)_{trans}. \tag{5}$$

Here, $\frac{dO_x}{dt}$ is the change rate of the observed O$_x$ mixing ratio change (ppbv h$^{-1}$); $P(O_x)_{net}$ denotes the net photochemical O$_3$ production rate (ppbv h$^{-1}$), which was equal to $P(O_3)_{net}$ and measured directly by the NPOPR system; and $R(O_x)_{trans}$ represents the O$_3$ mixing ratio change due to physical transportation (ppbv h$^{-1}$), including the horizontal and vertical transport, dry deposition, and the atmospheric mixing (Liu et al., 2022). To correct for the effects of NO titration on O$_3$, we have replaced O$_3$ with O$_x$ ($= O_3 + NO_2$) during the calculation in this study (Pan et al., 2015).

### 2.3.3  Model performance

In order to judge the reliability of the model simulation, we calculated the index of agreement (IOA) based on the measured and modelled $P(O_3)_{net}$ and $O_3$ at 5 m above ground level using the following equation (X. Liu et al., 2021):

$$\text{IOA} = 1 - \frac{\sum_{i=1}^{n}(O_i - S_i)^2}{\sum_{i=1}^{n}\left(|O_i - \overline{O}| + |S_i - \overline{O}|\right)^2}. \quad (6)$$

Here, $S_i$ and $O_i$ represent the simulated and observed $P(O_3)_{net}$ or $O_3$ values at the same time, respectively; $\overline{O}$ is the averaged observed value; and $n$ is the number of data. Furthermore, we also judged the model simulation performance using statistical measures, including the normalized mean bias (NMB) and normalized mean error (NME), which are defined as follows:

$$\text{NMB} = \frac{\sum_{i=1}^{n}(S_i - O_i)}{\sum_{i=1}^{n}O_i} \times 100\,\%, \quad (7)$$

$$\text{NME} = \frac{\sum_{i=1}^{n}|S_i - O_i|}{\sum_{i=1}^{n}O_i} \times 100\,\%. \quad (8)$$

Here, $S_i$ and $O_i$ have the same meaning as in Eq. (6) and $n$ is the total number of such data pairs of interest. The results will be discussed in Sect. 3.2.2.

### 2.3.4  OH reactivity

In order to investigate the influence of the photochemical reactions of different VOCs on photochemical $O_3$ formation, we calculated the OH reactivities of different VOCs, which is the sum of the concentrations of OH reactants multiplied by their reaction rate coefficients, as shown below:

$$k_{OH} = k_i[\text{VOCs}]_i. \quad (9)$$

Here, $k_{OH}$ represents the total OH reactivity of a group of VOC species, $k_i$ represents the rate constants between OH radicals and different VOC species $i$, and $[\text{VOCs}]_i$ represents the concentration of species $i$. In this study, we summarized the OH reactivities of different kinds of VOC groups together to investigate their influence on the vertical gradient $P(O_3)_{net}$ in Sect. 3.2.3.

### 2.3.5  O₃ formation potential

The $O_3$ formation potential is calculated using the product of the VOC concentration and the maximum incremental reactivity (MIR) coefficient (dimensionless, grams of $O_3$ produced per gram of VOCs) (Carter and Heo, 2012):

$$\text{OFP}_i = \sum_i[\text{VOC}]_i \times \text{MIR}_i. \quad (10)$$

Here, $\text{OFP}_i$ is the $O_3$ formation potential of species $i$, $[\text{VOC}]_i$ is the mass concentration or emission of species $i$, and $\text{MIR}_i$ denotes the MIR of species $i$.

### 2.3.6  O₃ formation regime

The sensitivity of photochemical $O_3$ production to its precursors was diagnosed by calculating the relative incremental reactivity (RIR) using the OBM-MCM model. The RIR is defined as the percent change in $O_3$ photochemical production per percent change in the concentration of its single precursor/precursor group (Cardelino and Chameides, 1995). Therefore, the RIR for precursor (group) $X$ can be expressed as follows:

$$\text{RIR} = \frac{\Delta P(O_3)/P(O_3)}{\Delta X/X}. \quad (11)$$

Here, $\Delta X/X$ represents the percent change in different $O_3$ precursors or precursor groups. We classified the measured VOCs into anthropogenic organic compound (AVOC), biogenic organic compound (BVOC), and oxygenated volatile organic compound (OVOC) groups, and investigated the $O_3$ formation sensitivity to these different types of VOCs.

## 3  Results and discussions

### 3.1  Vertical and temporal profile of O₃ and its precursors

#### 3.1.1  O₃ and its precursors at 5 m above ground level

Figure 1 shows the time series of the major trace gases, photolysis rate constants, and meteorological parameters at 5 m above ground level during the observation period at SZMGT. Over the 1-month field observation period, a total of three $O_3$ pollution episodes (referred to episodes hereafter) and two non-pollution (with respect to $O_3$) episodes (referred to non-episodes hereafter) were captured. The $O_3$ pollution episodes were defined as the days during which the hourly average $O_3$ concentration above ground level (5 m) exceed the Grade-II standard (102 ppbv, GB 3095-2012, China; Ambient Air Quality Standards, 2012), while the remaining days were defined as non-episodes. Episode days (marked as grey columns in Fig. 1) included 13–18 November (episode I), 26 November (episode II), and 7–9 December (episode III), while the non-episode days included 22–25 November (non-episode I) and 26–27 and 30 November (non-episode II). The corresponding daytime mean values (06:00–18:00 LT) during all episode days and non-episode days are shown in Table 1. During the daytime on episode days (episodes I, II, and III), the mean concentrations of $O_3$ were $70.1 \pm 28.6$, $59.5 \pm 32.4$, and $71.3 \pm 31.0$, respectively. The respective mean $T$ and RH were $22.3 \pm 2.5\,°C$ and $56.2 \pm 14.5\,\%$ for episode I, $20.4 \pm 3.2\,°C$ and $52.2 \pm 16.7\,\%$ for episode II, and $20.6 \pm 3.4\,°C$ and $58.2 \pm 17.2\,\%$ for

episode III. During non-episode days, the mean concentrations of $O_3$ were $45.3 \pm 16.2$ and $63.7 \pm 21.3$ ppbv for non-episode I and II, respectively. The corresponding mean $T$ and RH were $18.4 \pm 4.3$ °C and $69.5 \pm 15.4$ % for non-episode I and $21.3 \pm 2.7$ °C and $51.8 \pm 13.7$ % for non-episode II. These observations indicate that the $T$ and RH during episode days were not significantly different from those during non-episode days. This phenomenon contrasts with previous studies in the PRD area, where $O_3$ pollution episodes have generally been associated with high $T$ and low RH (Mousavinezhad et al., 2021; Hong et al., 2022).

The mean concentrations of $O_3$ precursors, including CO, NO, $NO_2$, and the total VOCs measured by the PTR-TOF-MS (shown as TVOCs in Fig. 1 and Table 1), did not exhibit notable discrepancies between episodes and non-episodes. This suggests that their concentrations during $O_3$ pollution episodes can vary, being either higher or lower than those observed during non-episodes (as shown in Table 1). For example, although there are days with very high hourly average $O_3$ concentrations that define $O_3$ pollution episodes – where levels exceed the Grade-II standard of 102 ppbv – the overall average $O_3$ concentrations for episode II is not higher than that of non-episode II. This suggests that, despite the occurrence of peak hourly levels, the average concentration for episode II remains lower, highlighting the fluctuating pattern of $O_3$ levels during these episodes. Further comparison of the daytime mean OFP and the measured $P(O_3)_{net}$ during episodes and non-episodes showed no significant differences, ranging from $5.1 \times 10^{-4}$ to $1.0 \times 10^{-3}$ g m$^{-3}$ and from 14.3 to 21.5 ppb h$^{-1}$, respectively, during episodes, whereas they ranged from $4.1 \times 10^{-4}$ to $4.7 \times 10^{-4}$ g m$^{-3}$ and from 5.6 to 18.9 ppb h$^{-1}$, respectively, during non-episodes. Although the OFP was always higher during episodes than during non-episodes, the mean $P(O_3)_{net}$ values during episodes I and III were even lower than during non-episodes II. The higher $O_3$ concentrations during episodes I and III may be due to the more stable weather conditions (with a lower wind speed), which benefits the accumulation of $O_3$ formed by local photochemical $O_3$ formation. However, for non-episode II, although it processes higher daytime mean $P(O_3)_{net}$ than episodes I and III, outflow of $O_3$ from the observation site by physical processes is also high due to a higher wind speed. These findings indicate that the $O_3$ pollution episodes stem from either substantially elevated local photochemical $O_3$ formation (i.e. episode II) or the accumulation of $O_3$ formed by moderate local photochemical $O_3$ formation under stable weather conditions (i.e. episodes I and II). Notably, when local photochemical reactions contribute intensely to the formation of $O_3$, favourable weather conditions facilitating $O_3$ outflow diminish the likelihood of $O_3$ pollution occurrences (i.e. non-episode II). These results indicate that $O_3$ pollution episodes are jointly affected by the photochemical reactions and physical transport processes, which we will discuss in more detail in Sect. 3.2.1.

### 3.1.2 Vertical profiles of $O_3$ and its precursors at the 5–335 m level

Figure 2 shows the contour plots illustrating the vertical profiles of $O_3$, $NO_x$, $O_x$ ($= O_3 + NO_2$), and TVOCs. From Fig. 2, minimal vertical gradients were observed during daytime in the concentration of all species – $O_3$, $NO_x$, $O_x$, and TVOCs – due to the rapid vertical mixing effects. However, distinct vertical gradients were observed during nighttime owing to the stability of the nocturnal residual layer. Elevated concentrations of $O_3$ and $O_x$ were identified at higher altitudes, whereas higher $NO_x$ concentrations predominantly occurred at ground level. We further elucidated the vertical distribution patterns of different pollutants as well as the OFP of different VOC groups during local daytime (06:00–18:00 LT) and nighttime (19:00–05:00 LT) for both episodes and non-episodes, as shown in Fig. 3.

The vertical profiles of averaged concentrations of various pollutants exhibit similar trends during both episodes and non-episodes, with $O_3$ showing an increasing trend from 5 m to 355 m above ground level, aligning with findings from previous studies (Zhang et al., 2019; Wang et al., 2021). Given that $NO_x$ has a significant titration effect on $O_3$, the lower $O_3$ concentration at ground level may be attributed to the increase in the $NO_x$ concentration (Zhang et al., 2022) and also the dry deposition near the ground (Li et al., 2022). NO and $NO_x$ showed an opposite trend compared with $O_3$. These two factors jointly effected the $O_x$ changing trend with height; consequently, the gradients of $O_x$ concentrations showed a weaker increasing trend from the 5 m above ground level to 355 m height compared with $O_3$. This observation demonstrated a more pronounced NO titration effect at 5 m above ground level compared with the effect at 355 m height. However, TVOCs showed variable trends with increased height for daytime and nighttime during episodes and non-episodes. During daytime, TVOCs initially decreased from 5 to 40 m; they then continuously increased from 40 to 355 m during episodes, whereas they continuously slightly decreased from 5 to 335 m during non-episodes. During nighttime, TVOC concentrations first increased from 5 to 40 m and then continuously decreased from 40 to 335 m during both episodes and non-episodes. We further plotted the OFP of different VOC categories at various altitudes, including OVOCs, aromatics, alkynes, alkenes, and alkanes, and found that the total OFP was highest at 5 m above ground level and exhibited higher levels during episodes compared with non-episode periods. Subsequently, there was a significant decrease at 40 m height during both episodes and non-episodes. However, there was a sharp increase observed at 70, 120, and 220 m during episodes, contrasting with a gradual rise during non-episode periods, which eventually reach a peak at 220 m during non-episodes. A consistent decrease in the OFP from 220 to 335 m was observed during both episodes and non-episodes. The OFP was primarily attributed to OVOCs among different VOC categories at different altitudes throughout both

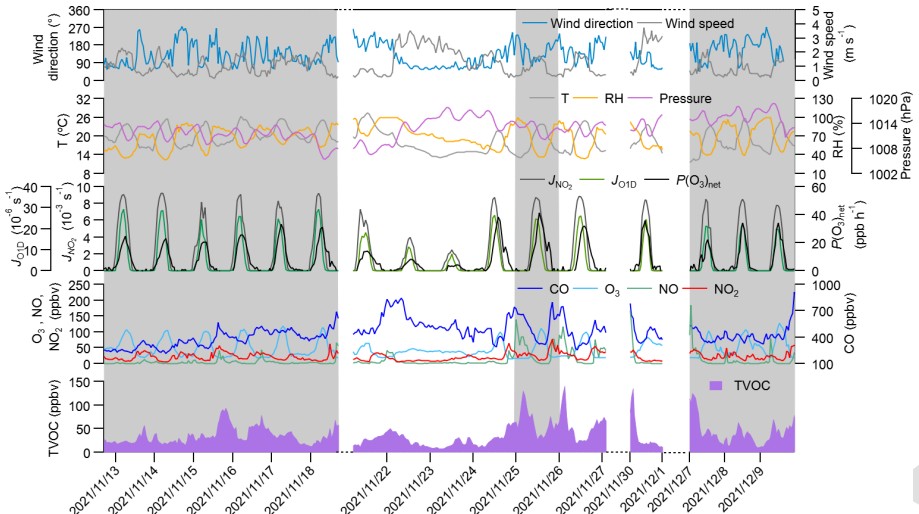

**Figure 1.** Time series of major trace gases, photolysis rate constants, $P(O_3)_{net}$, and meteorological parameters at 5 m above ground level during the observation period. The grey columns show the typical $O_3$ episodes that occurred. Dates are given in the following format: year/month/day.

**Table 1.** Daytime mean values of major trace gas concentrations (units: ppbv), OFP (units: $g\,m^{-3}$), $P(O_3)_{net}$ (units: $ppbv\,h^{-1}$), and meteorological parameters during different episodes and non-episodes in the observation period (from 13 November to 9 December 2021) at SZMGT.

| Parameters | Mean $\pm$ SD | | | | |
| --- | --- | --- | --- | --- | --- |
| | Episode I | Episode II | Episode III | Non-episode I | Non-episode II |
| $O_3$ | $70.1 \pm 28.6$ | $59.5 \pm 32.4$ | $71.3 \pm 31.0$ | $45.3 \pm 16.2$ | $63.7 \pm 21.3$ |
| TVOCs | $29.6 \pm 10.6$ | $53.8 \pm 21.7$ | $42.9 \pm 11.5$ | $23.3 \pm 8.6$ | $26.8 \pm 11.1$ |
| CO | $344.9 \pm 85.1$ | $408.8 \pm 85.4$ | $397.2 \pm 42.1$ | $508.5 \pm 117.2$ | $383.4 \pm 74.6$ |
| NO | $2.3 \pm 2.6$ | $13.1 \pm 17.4$ | $6.6 \pm 13.8$ | $2.9 \pm 2.0$ | $6.8 \pm 13.1$ |
| $NO_2$ | $15.6 \pm 7.5$ | $22.3 \pm 10.2$ | $20.0 \pm 8.3$ | $14.1 \pm 6.8$ | $15.4 \pm 8.8$ |
| OFP | $5.1 \times 10^{-4} \pm$ $7.5 \times 10^{-5}$ | $1.0 \times 10^{-3} \pm$ $2.0 \times 10^{-4}$ | $7.2 \times 10^{-4} \pm$ $8.3 \times 10^{-5}$ | $4.1 \times 10^{-4} \pm$ $5.6 \times 10^{-5}$ | $4.7 \times 10^{-4} \pm$ $7.8 \times 10^{-5}$ |
| $P(O_3)_{net}$* | $14.3 \pm 10.7$ | $21.5 \pm 14.9$ | $14.6 \pm 11.9$ | $5.6 \pm 4.6$ | $18.9 \pm 13.9$ |
| $T$ (°) | $22.3 \pm 2.5$ | $20.4 \pm 3.2$ | $20.6 \pm 3.4$ | $18.4 \pm 4.3$ | $21.3 \pm 2.7$ |
| RH (%) | $56.2 \pm 14.5$ | $52.2 \pm 16.7$ | $58.2 \pm 17.2$ | $69.5 \pm 15.4$ | $51.8 \pm 13.7$ |
| Wind speed ($m\,s^{-1}$) | $1.3 \pm 0.5$ | $1.2 \pm 0.4$ | $1.1 \pm 0.5$ | $1.8 \pm 0.9$ | $2.1 \pm 0.9$ |
| Wind direction (°) | $115.5 \pm 48.7$ | $128.6 \pm 35.3$ | $144.8 \pm 57.1$ | $115.0 \pm 57.6$ | $115.3 \pm 36.2$ |

* All values here were calculated as the mean average values during daytime (06:00–18:00 LT).

episodes and non-episodes, followed by aromatics and alkanes during episodes and non-episodes, respectively.

In conclusion, our daytime observations revealed minimal vertical gradients in the concentrations of $O_3$, $NO_x$, $O_x$, and TVOCs, attributed to the rapid vertical mixing effects driven by surface heating effects (Tang et al., 2017). This suggests that ground-level $O_3$ concentrations during daytime would be representative of the entire vertical column. Nonetheless, the OFP varies for different VOC profiles at various heights, and the vertical mixing effect facilitates the downward transport of $O_3$ photochemically formed from higher altitudes to the near-ground layer. Consequently, a box model that was constrained by all measured ground-level $NO_x$ and VOC con-

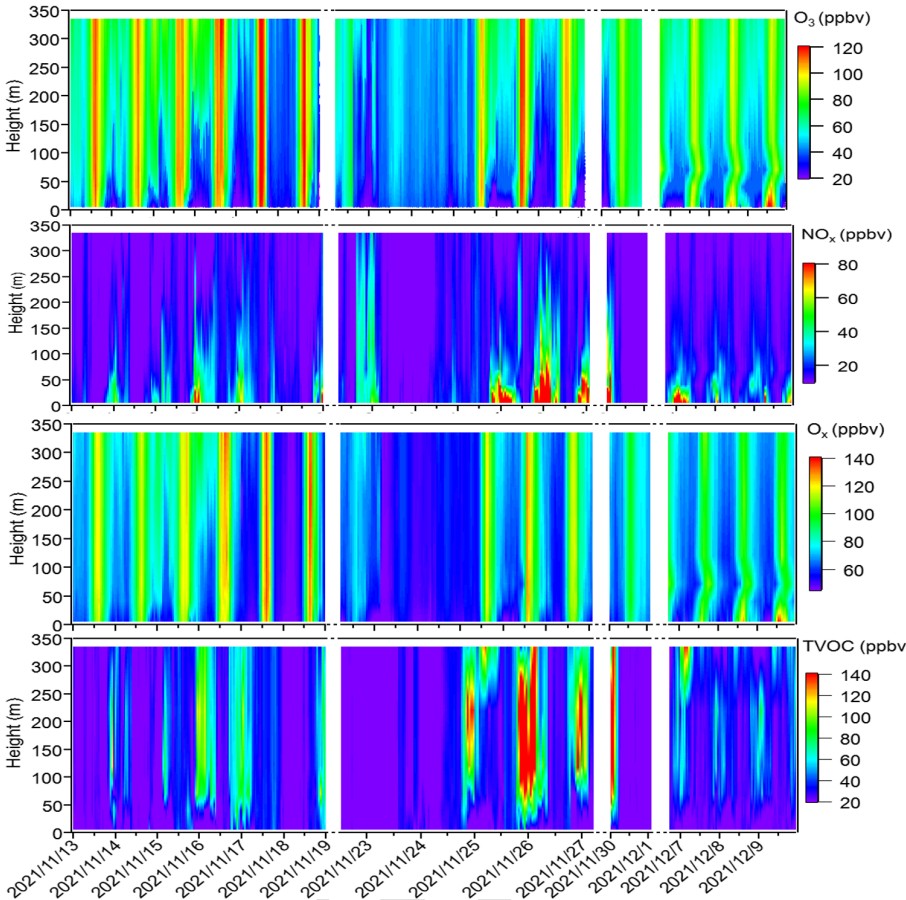

**Figure 2.** Time series of vertical profiles for $O_3$, $NO_x$, $O_x$, and TVOCs during the observation period. The contour plots are made using the measured values from six heights. Dates are given in the following format: year/month/day.

centrations may not accurately reflect the in situ $O_3$ production in the vertical atmospheric column.

## 3.2 The $O_3$ pollution episode formation mechanism at the near-ground surface

In this section, we first explore the possible reason for $O_3$ pollution episodes at 5 m above ground level, aiming to identify the contribution of chemical and physical processes to change in $O_3$ concentrations (Sect. 3.2.1). Subsequently, we assessed the modelling performance and investigated the potential reasons for the modelling bias in photochemical $O_3$ formation by comparing the measured $P(O_3)_{net}$ with the modelled $P(O_3)_{net}$ (Sect. 3.2.2). To gain insights into the photochemical $O_3$ formation mechanism at different heights and understand their impacts on overall $O_3$ pollution, we further discuss the chemical budget of $O_3$ at different heights (Sect. 3.2.3), the vertical and temporal variability in the $P(O_3)_{net}$ and $O_3$ formation regime (Sect. 3.2.4).

### 3.2.1 Contribution of the chemical and physical processes to $O_3$ changes at ground level

As concluded in Sect. 3.1.1, $O_3$ pollution episodes may be jointly affected by photochemical reactions and physical transport. In order to identify the main reasons for $O_3$ pollution at ground level, we calculated the contribution of chemical and physical processes to $O_x$ concentration changes at 5 m above ground level separately for all three pollution episodes and two non-episodes. Typically, as dry deposition contributes a relatively small portion and can often be considered negligible, vertical and horizontal transport are the main contributors to physical processes (Tan et al., 2019).

The $R(O_x)_{trans}$ at 5 m above ground level was derived from $\frac{dO_x}{dt}$ minus $P(O_x)_{net}$, according to Eq. (5) (shown Sect. 2.3.2), and the hourly averages and diurnal variations are shown in Figs. 4 and 5, respectively. From these figures, it is evident that the fluctuation in the $O_x$ concentration change rate $(d(O_x)/dt)$ at 5 m above ground level is typically small and primarily dominated by the physical processes during nighttime. During nighttime, $P(O_x)_{net}$ should be zero without solar radiation; the significant $P(O_x)_{net}$ shown in Fig. 5

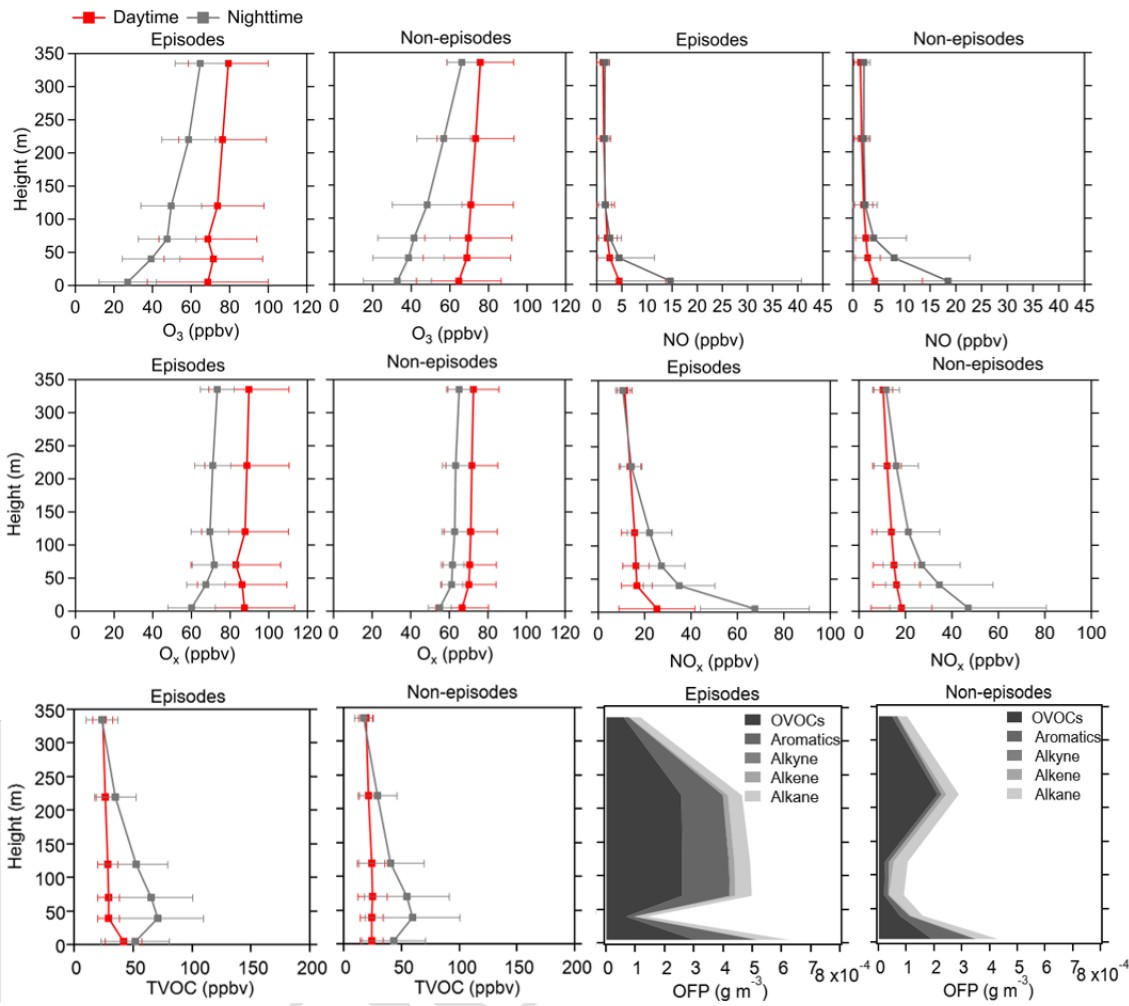

**Figure 3.** Average vertical profiles of $O_3$, NO, $O_x$, $NO_x$, and TVOCs during both daytime and nighttime as well as the OFP of different VOC types during daytime at six heights (5, 40, 70, 120, 220, and 335 m) for episodes and non-episodes throughout the observation period. The error bars indicate the standard deviation calculated from the measured values during these periods.

may be due to the measurement uncertainty in $P(O_x)_{net}$, which is determined by the measurement error of $O_x$ for the CAPS-$NO_2$ monitor in the reaction and reference chambers (as discussed in Sect. S4). The measurement uncertainty in $P(O_x)_{net}$ is higher at lower $P(O_x)_{net}$ values (as shown in Fig. 4), which was mainly determined by the instrumental error of the $O_x$ measurement and the ambient $O_x$ concentrations during nighttime. It was estimated to be $\sim 38\%$ and can be considered to be the measurement precision. Around 06:00–07:00 LT, $O_3$ concentrations increase for all episodes and non-episodes, mainly due to physical transport during episodes I and II and non-episode I, while photochemical reactions and physical processes are equally important for episode III and non-episode II. This could be due to short-term strong vertical turbulence in the early morning, which leads to an expansion of the boundary layer height and makes the residual layer "leaky", allowing vertical transport. At the same time, $O_3$ precursors were also transported

down from the residual layer; with increasing sunlight, these $O_3$ precursors underwent rapid photochemical reactions that competed with the physical processes between 06:00 and 7:00 LT, leading to a sharp increase in $P(O_x)_{net}$ between 08:00 and 12:00 LT. The $P(O_x)_{net}$ peaked around 11:00–14:00 LT and started to decrease around 15:00 LT, eventually approaching zero by around 19:00–20:00 LT. Between 07:00 and 08:00 LT, $R(O_x)_{tran} > 0$ for all episodes and non-episodes, indicating inflow of $O_3$ from physical transport, increasing the surface $O_3$ concentration by averages of 4.7, 3.9, 2.3, 3.5, and 4.5 ppbv h$^{-1}$ for episodes I, II, and III and non-episodes I and II, respectively. From 09:00 to 10:00 LT, $R(O_x)_{tran} > 0$ only for episode I, increasing the $O_3$ concentration by 1.5 ppbv h$^{-1}$, indicating inflow of $O_3$ from physical transport; on the contrary, $R(O_x)_{tran} < 0$ for episodes II and III and for non-episodes I and II, indicating outflow of $O_3$ from physical transport, decreasing the $O_3$ concentration by 3.1, 0.1, 3.0, and 16.9 ppbv h$^{-1}$, respectively. After

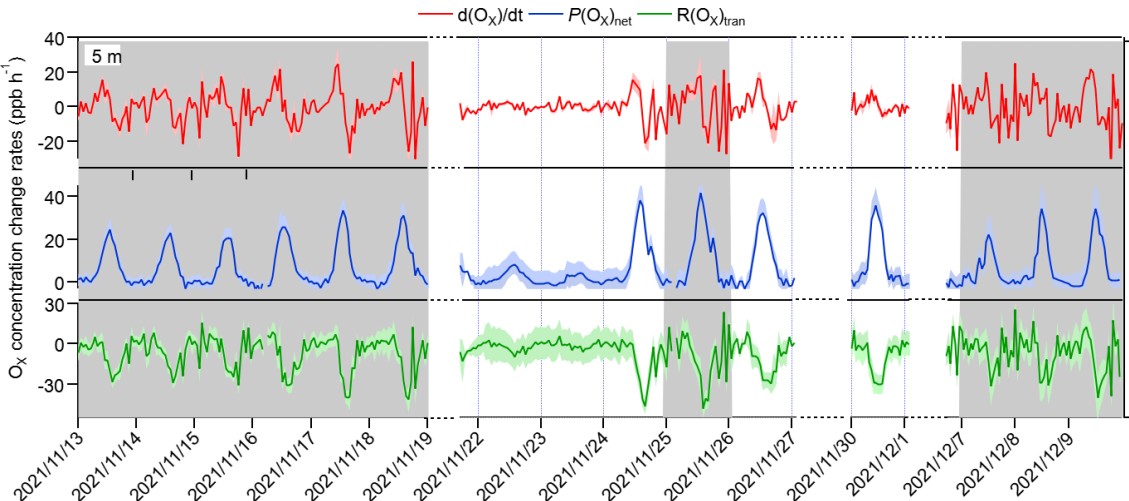

**Figure 4.** Time series of $O_x$ concentration changes ($d(O_x)/dt$) and contributions from local photochemical production ($P(O_x)_{net}$) and physical transport ($R(O_x)_{tran}$). The grey columns show the typical $O_3$ episodes that occurred. The shaded areas of $d(O_x)/dt$, $P(O_x)_{net}$, and $R(O_x)_{tran}$ represent 1 standard deviation (denoted by $\sigma$) of the mean $d(O_x)/dt$, the uncertainty in the measured $P(O_x)_{net}$, and the propagated error of $R(O_x)_{tran}$, respectively. Dates are given in the following format: year/month/day.

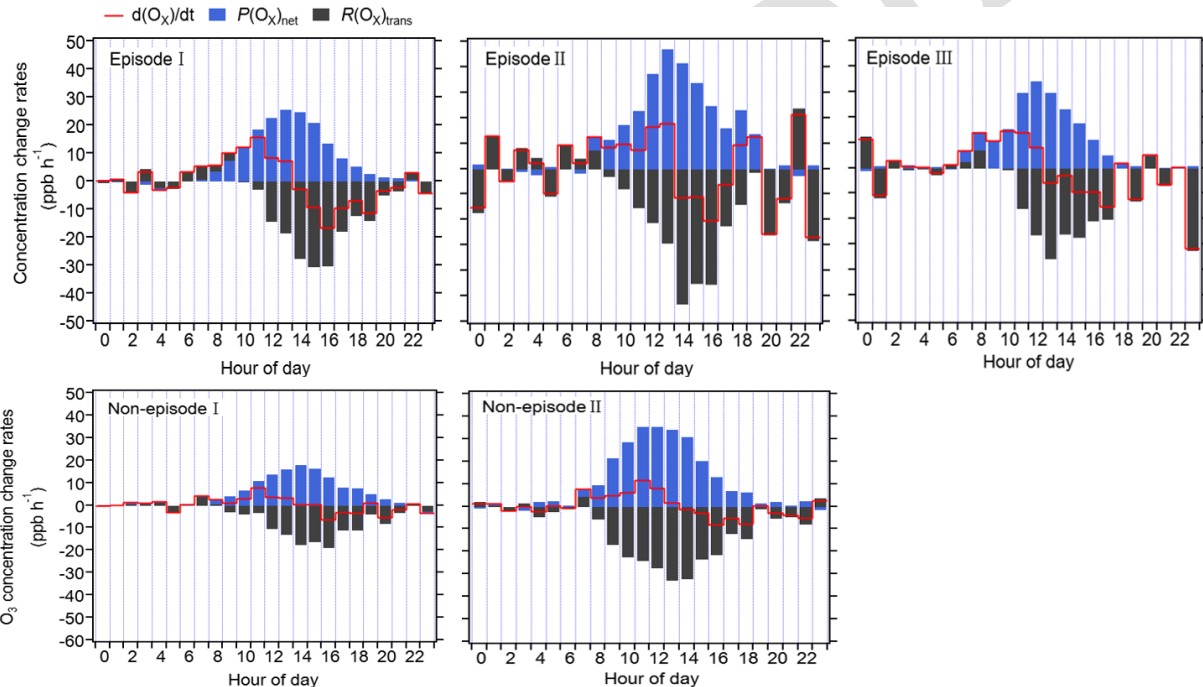

**Figure 5.** Diurnal variation in the contribution of chemical and physical processes to $O_3$ changes at 5 m above ground level.

10:00 LT, $R(O_x)_{tran} < 0$ for all episodes and non-episodes, indicating outflow of $O_3$ from the observation sites. This outflow may be possibly due to the accumulation of photochemically formed $O_3$, which increases the concentration at the observation site and causes it to diffuse upward or to surrounding areas.

In conclusion, the observed daytime $O_3$ concentration changes during all episodes and non-episodes were influenced by both photochemical production and physical transport. In the early morning, the increase in $O_3$ concentrations can be attributed to photochemical reactions, physical processes, and possibly reduced NO titration effects as the boundary layer height increases. Around noon, $O_3$ concentrations stabilize, suggesting a balance between photochemical reactions and physical transport affecting $O_3$ concentration changes. In the afternoon, $O_3$ concentrations decrease

due to the transport of photochemically formed $O_3$ from the observation site in the vertical direction or to surrounding areas. Our findings indicate that local photochemical reactions dominate $O_3$ pollution. For example, $O_3$ pollution episodes recorded during the observation period manifest under specific conditions: (1) high photochemical $O_3$ production (i.e. episode II) or (2) moderate photochemical $O_3$ production coupled with $O_3$ accumulation under stable weather conditions (i.e. episodes I and III). In contrast, non-episodes observed during the observation period occur under different conditions: (1) low levels of photochemical $O_3$ production (i.e. non-episode I) or (2) elevated photochemical $O_3$ production, with $O_3$ transport to surrounding areas under favourable diffusion conditions (i.e. non-episode II).

### 3.2.2 The model performance

In order to test the simulation ability of the OBM-MCM model with respect to $P(O_3)_{net}$, we compared the measured and modelled $P(O_3)_{net}$ at 5 m above ground level, as depicted in Fig. S3a. The measured and modelled $P(O_3)_{net}$ revealed close alignment during episodes I and III, but it displayed discernible variations during episode II, non-episode I, and non-episode II. Assessment metrics, including the IOA, NMB, and NME, were computed based on the observed and modelled $P(O_3)_{net}$ over the entire measurement period (as described in Sect. 2.3.3). The IOA was 0.90 for the measured and modelled daytime $P(O_3)_{net}$ across the measurement period, indicating the acceptable performance of the OBM-MCM model simulation (a higher IOA value signifies a stronger agreement between simulated and observed values). Additionally, comparison of measured and modelled $O_3$ concentrations at different heights (as shown in Fig. S4) revealed generally higher modelled values during daytime and closer alignment during nighttime at lower heights (i.e. 5, 40, and 70 m), while discrepancies were observed at higher heights (i.e. 120, 220, and 335 m). These phenomena may be primarily attributed to uncertainties in assumed physical processes in the modelling, such as vertical and horizontal transport. Previous work has utilized the comparison of measured and modelled $O_3$ concentrations to determine the dilution factor in modelling studies, discovering that suitable dilution factors vary by location (Yang et al., 2022). To achieve the best agreement between the modelled $O_3$ concentrations and the observed values, we applied different dilution factors (the lifetime of the species) in the model, varying from 6 to 24 h. We found that the simulated $O_3$ is closest to the measured $O_3$ concentrations when the lifetime of the species is set to 12 h. However, given that $O_3$ concentrations are affected by physical transport processes, the dilution factor might only represent the outflow of $O_3$ from the observation site. Therefore, there may be limitations to using this method for precise comparisons. We further compared the measured and modelled $P(O_3)_{net}$ using different dilution factors. The modelled $P(O_3)_{net}$ initially increases and then

decreases as the dilution factor decreases (equivalent to an increase in the species lifetime). However, the influence of varying dilution rates on the modelled $P(O_3)_{net}$ is minimal, constituting less than 30 %, due to the short lifetimes of the $HO_2$ and $RO_2$ radicals, which determine the $P(O_3)_{net}$ values (Wang et al., 2021). Notably, the modelled $P(O_3)_{net}$ closely matched the measured values when the species lifetime was set to 12 h, as illustrated in Fig. S3b. Consequently, a constant dilution factor of $1 / 43\,200\,s^{-1}$ was applied throughout the observation period. Further investigations revealed an IOA of 0.82 for measured and modelled daytime $O_3$ concentrations at 5 m above ground level, which lies in between the IOA results for the modelled and observed $O_3$ concentrations in previous studies, which ranged between 0.68 and 0.89 (Wang et al., 2018), signifying that the modelling results for $O_3$ concentrations here are acceptable. The calculated NMB and NME using the modelled and observed daytime $P(O_3)_{net}$ at 5 m above ground level during the whole measurement period ranged from $-0.42$ (25th percentile) to $-0.31$ (75th percentile) and from $-0.42$ (25th percentile) to 0.54 (75th percentile), respectively **TS1**. These analysis results indicate that the model underestimates the measured $P(O_3)_{net}$ by a factor ranging from 1.42 (25th percentile) to 1.31 (75th percentile), calculated as $(1 + |NMB|)$, and that the simulation results are reliable (with $-1 < NME < 1$) **TS2**.

The mean diel variation in the measured and modelled $P(O_3)_{net}$ during different episodes and non-episodes is shown in Fig. 6a–e. The maximum daily $P(O_3)_{net}$ values were 29.3, 47.2, and 34.2 $ppbv\,h^{-1}$ for episodes I, II, and III, respectively, while they were 17.9 and 35.5 $ppbv\,h^{-1}$ for non-episodes I and II, respectively. These values were comparable to or lower than those measured in urban areas of Houston (USA; 40–50 and 100 $ppbv\,h^{-1}$ in autumn and spring, respectively; Baier et al., 2015; Ren et al., 2013) but higher than those measured in a remote area of Japan (10.5 $ppbv\,h^{-1}$ in summer) and an urban area of Pennsylvania (USA; $\sim 8\,ppbv\,h^{-1}$ in summer) (Sadanaga et al., 2017; Cazorla and Brune, 2010). The averaged diel profiles of measured and simulated $P(O_3)_{net}$ exhibited large standard deviations (as depicted in Table 1), representing their day-to-day variation throughout the campaign. The measured $P(O_3)_{net}$ values were mostly higher than the modelled $P(O_3)_{net}$, which could be attributed to the underestimation of $RO_2$ under high-NO conditions, leading to substantial disparities between calculated $P(O_3)_{net}$ derived from measured and modelled $RO_2$ concentrations, as highlighted in previous studies (Whalley et al., 2018, 2021; Tan et al., 2017, 2018). The median value of [measured $P(O_3)_{net}$ − modelled $P(O_3)_{net}$] / measured $P(O_3)_{net}$ ranged from 22 % to 45 % for different episodes and non-episodes. To delve deeper, we further investigated the relationship between the daily disparities in measured and modelled $P(O_3)_{net}$ ($\Delta P(O_3)_{net}$ = measured $P(O_3)_{net}$ − modelled $P(O_3)_{net}$) and average daytime NO concentrations during different episodes and non-episodes, as depicted in Fig. 6f.

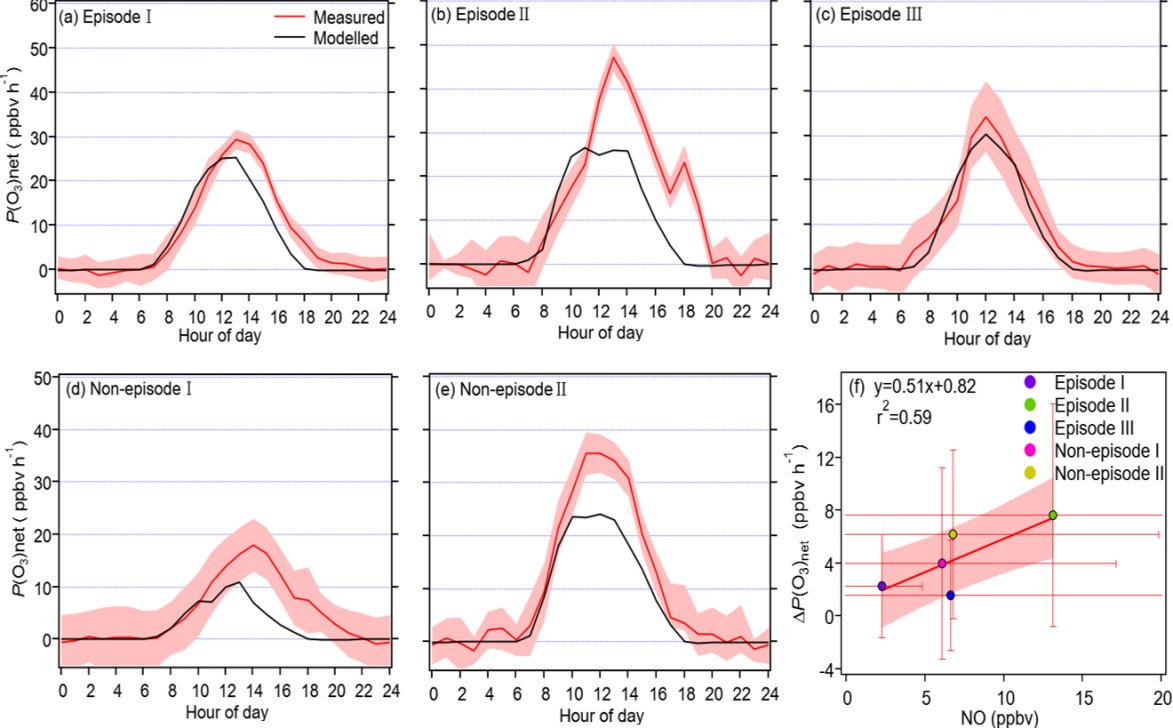

**Figure 6.** Panels **(a)**–**(e)** show the diurnal variation in the measured and modelled $P(O_3)_{net}$ during the observation period for different episodes and non-episodes, while panel **(f)** presents the relationship between the average daily differences in the measured and modelled $P(O_3)_{net}$ ($\Delta P(O_3)_{net}$) and the average daytime NO concentrations during different episodes and non-episodes.

The observed elevated $\Delta P(O_3)_{net}$ at higher NO concentrations aligns with findings from previous studies, which suggest that multiple factors could contribute to these outcomes. For example, the reaction of OH with unknown VOCs (Tan et al., 2017), the missing RO$_2$ production from photolysis ClNO$_2$ (Whalley et al., 2018; Tan et al., 2017), and the underestimation of OVOC photolysis (Wang et al., 2022b) in modelling approaches may lead to the underestimation of RO$_2$, thus underestimating the modelled $P(O_3)_{net}$. Further analysis showed that the underestimation of $P(O_3)_{net}$ can lead to the NO$_x$-limited regime being shifted to the VOC-limited regime, thus underestimating the NO$_x$-limited regime (Wang et al., 2022b, 2024). However, the derived IOA, NMB, and NME values from the modelled and observed $P(O_3)_{net}$ (and O$_3$) at 5 m above ground level during different episodes and non-episodes indicate that the model proficiently reproduces the genuine $P(O_3)_{net}$ at the observation site (as shown in Table S3). Consequently, these results provide confidence in exploring the vertical and temporal variations in $P(O_3)_{net}$ and O$_3$ formation sensitivities utilizing the outcomes from the modelling approach. Nonetheless, it is important to acknowledge and discuss the potential biases caused by the modelling methodology in this study.

### 3.2.3 Vertical and temporal variability in the $P(O_3)_{net}$ budget

The detailed $P(O_3)_{net}$ budget values at different heights during the observation period from the modelling results are shown in Fig. 7. Across various heights and different episodes and non-episodes, the contributions of different reaction pathways to $P(O_3)$ were almost the same, with HO$_2$ + NO as the major O$_3$ production pathway, followed by CH$_3$O$_2$ + NO and other RO$_2$ + NO, where other RO$_2$ + NO encompasses all RO$_2$ species except CH$_3$O$_2$. This result aligns with previous studies (X. Liu et al., 2021, 2022). The major O$_3$ destruction pathway was OH + NO$_2$ (loss of OH radicals), followed by net RO$_2$ + NO$_2$ (form peroxyacetyl nitrate, commonly called PAN species) and O$_3$ photolysis, while other O$_3$ destruction pathways, including O$_3$ + OH, O$_3$ + HO$_2$, C$_5$H$_8$ + O$_3$, C$_3$H$_6$ + O$_3$, and C$_2$H$_4$ + O$_3$, together contributed negligibly to O$_3$ destruction. These $P(O_3)$ and $D(O_3)$ reaction pathways occurred between 06:00 and 18:00 LT, exhibiting strong diurnal variation characterized by a sharp increase between 06:00 and 11:00 LT in the morning, peaking between 11:00 and 14:00 LT, and decreasing rapidly after 14:00 LT. These phenomena were in accordance with the concentration changes in the major oxidants (i.e. OH, O$_3$, and NO$_3$), as shown in Fig. S5, where OH radicals and O$_3$ concentrations increased significantly in the morning and

reached a peak around noon, followed by sharp afternoon decreases.

The diurnal changes in the concentrations of different reaction pathways to $P(O_3)$ and $D(O_3)$ at 5 m above ground level during different episodes and non-episodes are depicted in Fig. S6. We note that the maximum total $P(O_3)$ resulting from diel variations at 5 m above ground level for episode I, II, and III were 32.0, 34.9, and 38.3 ppbv h$^{-1}$, respectively. These values were consistently higher than the maximum total $P(O_3)$ observed for non-episodes I and II, which were 15.6 and 30.7 ppbv h$^{-1}$, respectively. However, as $P(O_3)_{net}$ was determined by both $P(O_3)$ and $D(O_3)$, the maximum total $D(O_3)$ values resulting from diel variations during episodes I, II, and III and non-episodes I and II were 5.0, 5.7, 5.1, 2.4, and 5.3 ppbv h$^{-1}$, respectively. Consequently, the modelled $P(O_3)_{net}$ during episodes does not exhibit a statistically significant difference from that during non-episodes (Mann–Whitney $p$ value $= 0.12$), as shown in Fig. S6, which is in agreement with the measured $P(O_3)_{net}$ (Mann–Whitney $p$ value $= 0.28$), as depicted in Sect. 3.1.1.

The diurnal variation in $P(O_3)_{net}$ during different episodes and non-episodes obtained by the OBM-MCM model at different heights is shown in Fig. 8. We saw that the $P(O_3)_{net}$ values all showed a decreasing trend with an increase in the measurement height during different episodes and non-episodes, but the variation in $P(O_3)_{net}$ along with the measurement height differed for different episodes and non-episodes. For example, the decrement in the averaged $P(O_3)_{net}$ during 06:00–18:00 LT from 5 to 335 m was 1.5 and 0.6 ppbv h$^{-1}$ for episode I and non-episode I, respectively, which was relatively smaller than that during episode II, episode III, and non-episode II (5.3, 5.4, and 4.0 ppbv h$^{-1}$, respectively). To explore the reason for this, we plotted the differences in the calculated OH reactivities at 5 and 335 m of different VOC groups (marked as $\Delta$OH reactivity) as a function of the $P(O_3)_{net}$ change at 5 and 335 m (marked as $\Delta P(O_3)_{net}$), including non-methane hydrocarbons (NMHCs), anthropogenic volatile organic compounds (AVOCs), biogenic volatile organic compounds (BVOCs), and oxygenated volatile organic compounds (OVOCs) (as shown in Fig. 8f). The VOC species included in each category are listed in Table S2. We found that the OH reactivities of AVOCs and OVOCs had the highest correlation coefficients ($R^2$) with the $\Delta P(O_3)_{net}$, which are 0.85 and 0.67, respectively, indicating their predominant influence on the decrement in $P(O_3)_{net}$ from 5 to 335 m. However, the OH reactivity change from 5 to 335 m of different groups was quite different. Therefore, we further explored $O_3$ formation sensitivity to its different VOC precursors and precursor groups.

### 3.2.4   Vertical distributions of $O_3$ formation regime

To investigate the reasons behind the variable distribution of $P(O_3)_{net}$ at varying heights, we clarified the sensitivity of $O_3$ formation to different $O_3$ precursors or precursor groups, including NMHCs, AVOCs, BVOCs, OVOCs, CO, and NO$_x$, by calculating their RIRs during different episodes and non-episodes, as shown in Fig. 9. The VOC species, categorized into different precursor groups as listed in Table S2, indicate that some species depicted in Fig. 9 may appear in multiple categories and, hence, could be repeated. We note that AVOCs include both NMHCs and OVOCs. Figure 9 demonstrates that the aggregate RIR of OVOCs and NMHCs is nearly identical to that of AVOCs alone. Recognizing that VOC species within the OVOC category primarily originate from anthropogenic sources but can also originate from biogenic precursors (Wu et al., 2020; Park et al., 2013), we acknowledge the possibility of an overestimated RIR for AVOCs due to this overlap. As illustrated in Fig. 9, the RIR values for different $O_3$ precursors or precursor groups do not exhibit significant variation at different heights during specific episodes or non-episodes, indicating a similar photochemical $O_3$ formation regime. However, the $O_3$ formation regimes differ between different episodes or non-episodes. During $O_3$ pollution episode I, $O_3$ formation is located in a transition regime and is more sensitive to VOC emissions. Conversely, during $O_3$ pollution episodes II and III and non-episodes I and II, $O_3$ formation is located in the VOC-sensitive regime. This finding aligns with previous studies suggesting that photochemical $O_3$ formation in the PRD region is likely in a VOC-limited or mixed-sensitivity regime (Hong et al., 2022; Lu et al., 2018). The results highlight the possible complexity of $O_3$ mitigation at the observation site. For example, during $O_3$ pollution episode I, reducing both VOCs and NO$_x$ can mitigate photochemical $O_3$ formation. However, during other $O_3$ pollution episodes and non-episodes, reducing VOCs can effectively alleviate photochemical $O_3$ formation, whereas reducing NO$_x$ might aggravate it. Nevertheless, during all episodes and non-episodes, $O_3$ formation is most sensitive to AVOCs (RIR: 0.83–1.12), followed by OVOCs (RIR: 0.59–0.79) at different heights; given that the AVOCs include NMHCs and OVOCs, there is urgent need to reduce NMHC and OVOC emissions to mitigate $O_3$ pollution in this area. Additionally, it is evident that OVOCs have a substantially higher RIRs than NMHCs; therefore, it is more accurate to conclude that the $O_3$ formation is most sensitive to OVOCs, rather than AVOCs.

The RIR tests for different episodes and non-episodes at various hours of the local daytime are illustrated in the Supplement (Fig. S7). The results indicate that the diurnal changes in the RIR values for different episodes and non-episodes exhibit remarkable similarities. In the morning, the RIR values for various VOC groups, including AVOCs, BVOCs, NMHCs, OVOCs, and CO, are typically higher than those for NO$_x$. However, the RIR values for AVOCs, BVOCs, NMHC, and OVOCs first gradually decrease throughout the day until 14:00 LT, then increase and reach a peak at 17:00 LT, and subsequently decrease sharply between 17:00 and 18:00 LT. Conversely, the RIR values for NO$_x$ are usually around zero or below zero during most of the day,

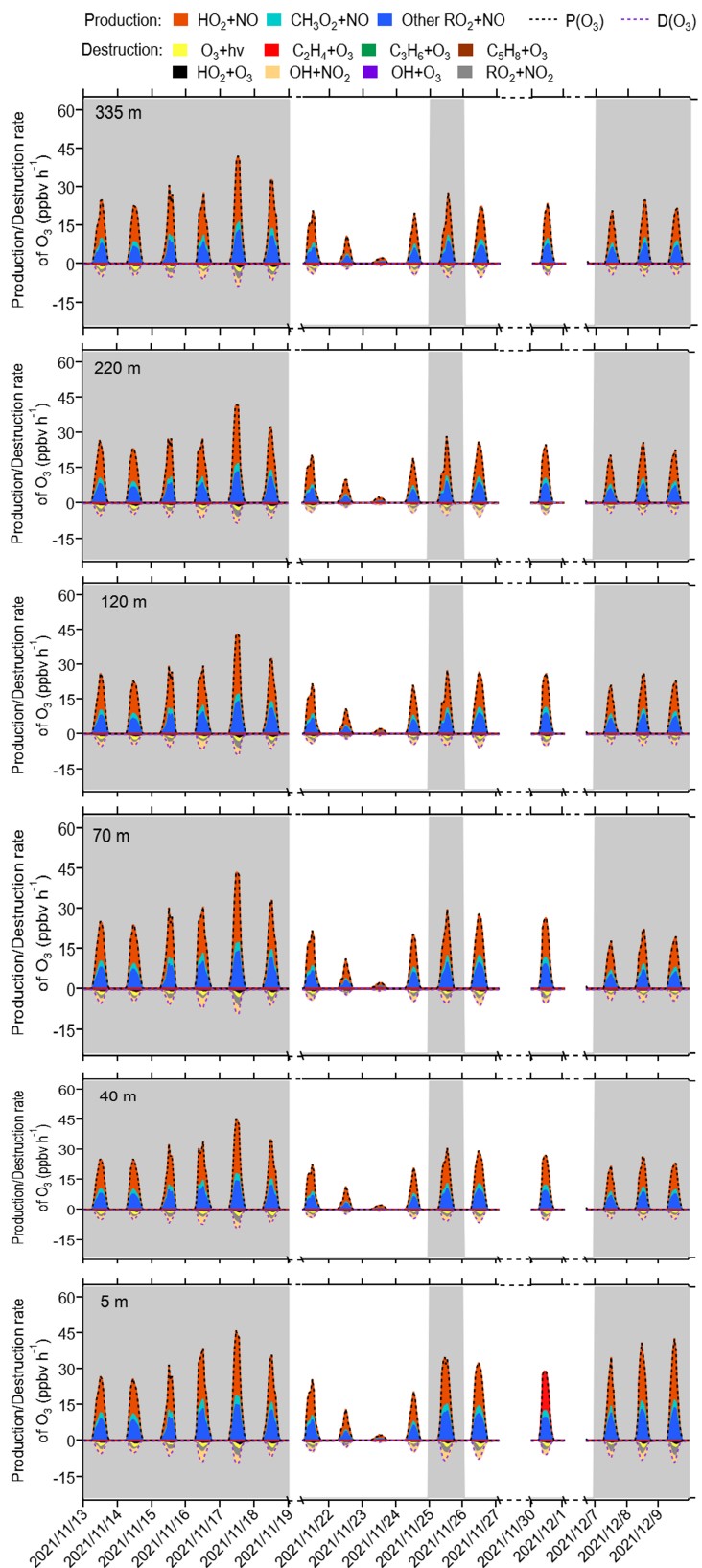

**Figure 7.** Time series of model-simulated O₃ production and destruction rates during 13 November and 9 December 2021 at different heights at SZMGT. The grey columns show the typical O₃ episodes that occurred. Dates are given in the following format: year/month/day.

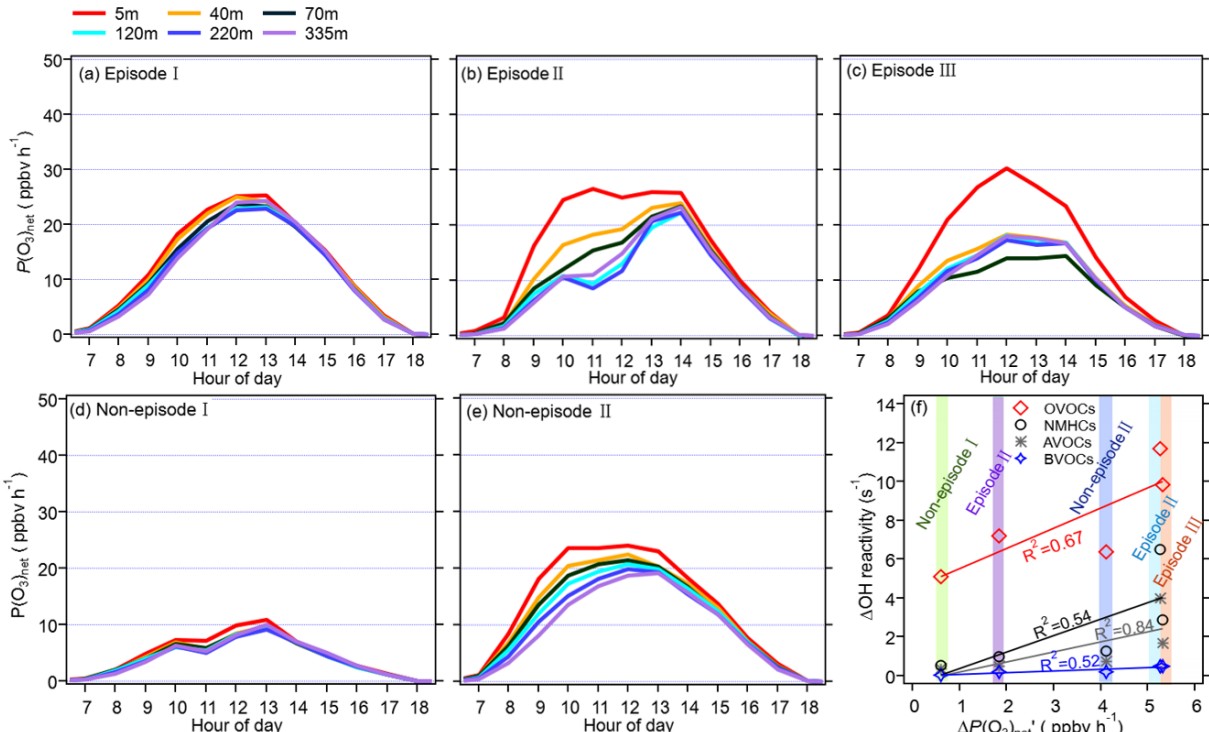

**Figure 8.** Panels **(a)**–**(e)** show the diurnal variation in the vertical profile of the model-simulated $P(O_3)_{net}$ during different episodes and non-episodes from 13 November to 9 December 2021, while panel **(f)** presents the relationship between the average daytime differences in the modelled $P(O_3)_{net}$ (denoted as $\Delta P(O_3)_{net}$) and the OH reactivity of different precursor groups at 5 and 335 m (denoted as $\Delta$OH reactivity).

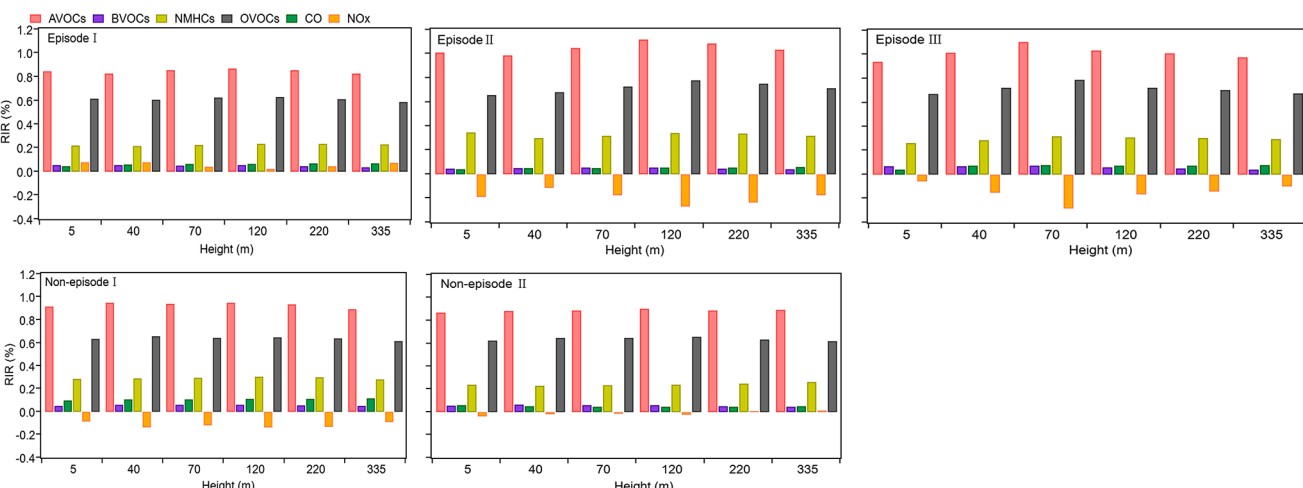

**Figure 9.** RIR values for $O_3$ precursors or precursor groups at different heights during different classified episodes and non-episodes.

gradually increase at around 16:00 LT, and peak at 18:00 LT. At 18:00 LT, the RIR values for the VOC groups are lower than those of $NO_x$. This suggests a transition in the photochemical $O_3$ formation regime throughout the day, shifting from a VOC-limited regime in the morning to a transition regime and a regime more sensitive to $NO_x$ in the afternoon

at around 16:00 LT. The diurnal variations in the RIRs of different $O_3$ precursors or precursor groups offer detailed insights into the dominant factors influencing the photochemical formation of $O_3$ at different times of a day.

Through the sensitivity study, $NO_x$ is not found to be the limiting factor affecting $P(O_3)_{net}$; therefore, reac-

tions involving NO$_x$ in the RO$_x$ radical cycle, such as RO$_2$ + NO → HO$_2$ and HO$_2$ + NO → OH, should occur efficiently. Conversely, reactions not involving NO$_x$, such as OH + VOCs → RO$_2$, are the limiting steps of the RO$_x$ radical cycle. We further identified and presented the three VOC species with the highest OFP values in the NMHC and OVOC groups during different episodes and non-episodes (Table S4). Results show that compounds such as toluene, $m/p$-xylene, and $n$-butane in the NMHC group and formaldehyde, hydroxyacetone, acetaldehyde, and ethanol in the OVOC group have been identified as the most significant contributors to the total OFP in all episodes and non-episodes. Toluene, $m/p$-xylene, and $n$-butane are often associated with specific industrial processes (Shi et al., 2022; Liang et al., 2017), whereas formaldehyde, hydroxyacetone, acetaldehyde, and acetaldehyde can originate from both industrial processes and natural sources (Parrish et al., 2012; Fan et al., 2021; Spaulding et al., 2003; Salthammer, 2023). Priority should be given to reducing these emission sources in order to mitigate O$_3$ pollution in the PRD area of China.

## 4   Conclusions

We carried out a field observation campaign in an urban area of the Pearl River Delta (PRD) in China, focusing on investigating the vertical temporal variability in the near-surface O$_3$ production mechanisms using a newly built vertical observation system and an observation-based model coupled to the Master Chemical Mechanism (OBM-MCM) v3.3.1. In total, three O$_3$ pollution episodes and two non-episodes occurred during the observation period. To assess the modelling performance with respect to the O$_3$ production rates and sensitivity and to investigate the potential reasons for O$_3$ pollution episodes at 5 m above ground level, a net photochemical O$_3$ production rate (NPOPR, $P(O_3)_{net}$) detection system based on the current dual-channel reaction chamber technique was employed to directly measure $P(O_3)_{net}$ at 5 m above ground level.

The vertical profiles of the averaged concentrations of various pollutants exhibit similar trends during both episodes and non-episodes. The O$_3$, NO$_x$, and O$_x$ concentrations show a minimal vertical gradient during the daytime due to rapid vertical mixing effects, but distinct vertical gradients emerge during nighttime owing to the stability of the nocturnal residual layer. Higher concentrations of O$_3$ and O$_x$ were observed at higher heights, whereas elevated NO and NO$_x$ concentrations were mainly detected at ground level. Given that NO has a significant titration effect on O$_3$, the lower O$_3$ concentration at ground level may be attributed to an increase in the NO$_x$ concentration due to a more pronounced NO titration effect, as well as dry deposition near the ground. However, the TVOCs and their OFP values exhibited variable trends with increased height during both daytime and nighttime, observed during both episodes and non-episodes, which indi-

cates the complexities involved with the O$_3$ formation mechanisms at different heights throughout the atmospheric column. The total OFP was highest at 5 m above ground level and exhibited higher levels during episodes compared with non-episode periods. The OFP was primarily attributed to OVOCs at different altitudes throughout both episodes and non-episodes.

The mean concentrations of O$_3$ precursors, including CO, NO, NO$_2$, and TVOCs, were not consistently elevated during episodes compared to their levels during non-episodes. By considering the observed O$_3$ concentration changes and the measured $P(O_3)_{net}$ at 5 m above ground level, we found that the O$_3$ pollution episodes were influenced by both photochemical production and physical transport, with local photochemical reactions playing a key role. The O$_3$ pollution episodes recorded during the observation period occurred under specific conditions: (1) high photochemical O$_3$ production and (2) moderate photochemical O$_3$ production coupled with O$_3$ accumulation under stable weather conditions. The index of agreement (IOA) was 0.90 for the measured and modelled $P(O_3)_{net}$ across the measurement period, indicating the rationality of investigating the vertical and temporal variability in the O$_3$ formation mechanisms using modelling results. However, the measured $P(O_3)_{net}$ generally exceeded the modelled $P(O_3)_{net}$, and the differences between measured and modelled $P(O_3)_{net}$ ($\Delta P(O_3)_{net}$) were found to be correlated with NO concentrations. Based on previous studies, this phenomenon could potentially be attributed to the underestimation of RO$_2$ under high-NO conditions, arising from inadequate knowledge concerning photochemical reaction mechanisms. Therefore, the potential biases caused by the modelling methodology have been acknowledged and discussed.

From the modelling results, the contribution of different reaction pathways to $P(O_3)$ was almost the same at varying heights during both episodes and non-episodes, with HO$_2$ + NO as the major O$_3$ production pathway, followed by other RO$_2$ + NO (comprising all RO$_2$ species except CH$_3$O$_2$) and CH$_3$O$_2$ + NO. The major O$_3$ destruction pathway was OH+NO$_2$ (loss of OH radicals), followed by net RO$_2$ + NO$_2$ (forming peroxyacetyl nitrate) and O$_3$ photolysis. However, other O$_3$ destruction pathways, including O$_3$ + OH, O$_3$ + HO$_2$, C$_5$H$_8$ + O$_3$, C$_3$H$_6$ + O$_3$, and C$_2$H$_4$ + O$_3$, collectively contributed negligibly to O$_3$ destruction. Nevertheless, $P(O_3)_{net}$ showed a decreasing trend with the increase in the height during different episodes and non-episodes, which was found to be mainly attributed to the decline in O$_3$ precursor concentrations, specifically the oxygenated volatile organic compounds (OVOCs) and non-methane hydrocarbons (NMHCs). We observed that modelling biases were correlated with NO concentrations and VOC categories, impacting $P(O_3)_{net}$ through the regulation of the RO$_2$ radicals' budget. The median relative difference between measured and modelled $P(O_3)_{net}$ ranged from 22 % to 45 % during different episodes and non-episodes.

Similar photochemical $O_3$ formation regimes were observed at different heights during specific episodes or non-episodes, but they varied between different episodes or non-episodes. The $O_3$ formation was predominantly located in a transition regime and was more sensitive to VOC emissions during $O_3$ pollution episode I, whereas it shifted to a VOC-sensitive regime during $O_3$ pollution episodes II and III as well as during non-episodes I and II. Further analysis revealed a daytime shift in the photochemical $O_3$ formation regime, transitioning from a VOC-limited regime in the morning to a transition regime that was more sensitive to $NO_x$ at around 16:00 LT in the afternoon. However, the underestimation of $RO_2$ radicals in the model, especially at lower heights with higher NO concentrations, could result in an overestimation of the VOC-limited regime. This study highlights the need for more precise analysis using direct measurement techniques in future studies. Nonetheless, throughout all episodes and non-episodes, $O_3$ formation is most sensitive to OVOCs at various heights, emphasizing the urgent need to reduce the emissions of these compounds and their precursors to mitigate $O_3$ pollution in this area.

This is the first measurement report of the vertical and temporal $O_3$ formation mechanisms near the ground surface. Together with deliberation regarding the possible bias in the vertical and temporal profile of the $O_3$ formation rate and sensitivity using modelling studies, this research provides critical foundational insights. The findings provide us with an in-depth understanding of near-ground vertical variability in the $O_3$ formation mechanisms, which are influenced by the concentrations of VOCs and $NO_x$, and the distinct OFP values associated with different VOC profiles. During daytime, the vertical mixing of air masses is substantially enhanced due to the effect of surface heating. Consequently, photochemically formed $O_3$ at higher altitudes can be vertically transported downward to the near-ground layer. Under these conditions, control strategies for $O_3$ precursors based on the $O_3$ formation mechanisms at ground level are insufficient. Thus, a more comprehensive approach is necessary to effectively address the complexities in $O_3$ production throughout the atmospheric column. The vertical variability in the $O_3$ formation mechanisms should be taken into account when making effective $O_3$ control strategies in the PRD area of China.

**Data availability.** Data related to this article are available online at https://doi.org/10.5281/zenodo.10473104 (Zhou and Yuan, 2024).

**Supplement.** The supplement related to this article is available online at: https://doi.org/10.5194/acp-24-1-2024-supplement.

**Author contributions.** BY, JZ, XBL, and MS designed the experiment; YHa and JZ performed the $P(O_3)_{net}$ measurements; BY and XBL built the vertical observation system based on the Shenzhen Meteorological Gradient Tower; JZ, CZ, AL, BY, JPZ, YHa, YWa, XBL, XH, XS, YC, SuY, ShY, YWu, and JQ collected and analysed the data; JZ wrote the manuscript; and all authors revised the manuscript.

**Competing interests.** The contact author has declared that none of the authors has any competing interests.

**Disclaimer.** Publisher's note: Copernicus Publications remains neutral with regard to jurisdictional claims made in the text, published maps, institutional affiliations, or any other geographical representation in this paper. While Copernicus Publications makes every effort to include appropriate place names, the final responsibility lies with the authors.

**Acknowledgements.** The authors wish to thank Shenzhen National Climate Observatory for their assistance during the observation campaign.

**Financial support.** This research has been supported by the Key Area Research and Development Program of Guangdong Province (grant no. 2020B1111360003), the Natural Science Foundation of Guangdong Province (grant no. 2024A1515011494), and the National Natural Science Foundation of China (grant no. 42305096). Bin Yuan and Xiao-Bing Li are supported by the National Natural Science Foundation of China (grant nos. 42275103 and 42121004).

**Review statement.** This paper was edited by Lisa Whalley and reviewed by three anonymous referees.

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

## Remarks from the typesetter