# Peer review of "Measurement report: Vertical and temporal variability of near surface ozone production rate and sensitivity in an urban area in Pearl River Delta (PRD) region, China"

_EGUsphere, 2023_

## Referee Comment (RC2)

**Review for: Measurement report: Vertical and temporal variability of near-surface ozone production rate and sensitivity in an urban area in Pearl River Delta (PRD) region, China.**

**Summary:**

This manuscript presents a detailed study of the vertical and temporal profile of ozone production, using both measurement data and chemical box modelling. Overall, the manuscript is well written with a good flow. However, there are some grammatical issues that should be addressed before publication. I really enjoyed reading this manuscript, and feel that it would be of great interest to the urban ozone community, therefore I recommend it for publication to ACP provided some important changes to the text have been made.

Of particular concern is the use of the phrase "statistically different/difference" throughout the text. When this phrase is used, I would expect it to be backed up by a statistical test(s). I don't think any further analysis is required, but the authors should re-think how they describe their findings and avoid this phrase when there is no statistical evidence to back up their claims. I would also recommend another detailed read-through of the document to check for grammatical errors that occur periodically. Please see below for more detailed comments.

**Detailed comments:**

56      Replace "have" with "has".

65      Check grammar.

71      Check grammar.

75      Please explain why using only one height of measurements is of great limitation. Why is it important that this is done at multiple altitudes, when $O_3$ exposure occurs at ground level?

81      Check grammar.

85      This sentence is very long – check grammar and split into two after "VOC and NOx measurements".

113     Misspelled "Asia".

116     Check grammar.

224     Please state in the text that a full list all 47 NMHCs s available in the supplementary (Table S2).

230     Please include in this section the dilution / ventilation approach you have included in the model.

333     I advise that you should avoid using the word "statistically", unless you have performed a statistical test. Please rephrase.

346     Again, if you use the phrase "statistically different", I would expect to see evidence of a statistical test showing this.

351     I would rephrase this. Are you saying that event and non-event OFP and $P(O_3)net$ are not statistically different because they are within one standard deviation?

352     It is not clear to me how you have come to this conclusion (These findings indicate that…). Please clarify in the text.

358     It is well known that $O_3$ pollution episodes are jointly affected by photochemical reactions and physical transport processes. It's unclear to be how what you have said previously has led you to this statement.

438     Remove word "besides" – not needed.

476     Instead of saying "the results show", be specific about which part of the figure you are referring to in the text. What is it in the figure that has led you to this conclusion?

478     How do you know this is attributed to physical transport? Is this just a suggestion based on knowledge of the atmosphere (please supply a reference), or is there something in the figure that provides evidence that increasing $O_3$ is due to vertical transport, and not early morning photochemistry? If it's because you have not measured any $P(O_3)net$ at this time, please state this in the text.

483     Change "residue" to "residual".

644     "statistically difference" – see comments for lines 333 and 346

659     In the text, please direct the reader to Table S2 which defines which VOCs are in each category (NMHC, AVOC, BVOC and OVOC).

698     Check grammar in this section.

816     Remove "proper" – not needed.

---

## Author Comment (AC1)

In this paper, the authors measure vertical profiles of ozone and its precursors concentrations. The authors measure directly net ozone production rate P(O3)net at ground level, and discuss ozone concentration variations in terms of both photochemical ozone production and physical transportation using measured P(O3)net and ozone concentrations. In addition, they compare observed and modeled values for P(O3)net and discuss the vertical distribution of P(O3)net and ozone production regimes calculated from the model. The discussion on the ozone budget and its vertical distribution is very important to mitigate ozone pollution problems, so that I recommend this paper to be published in ACP. However, I found several concerns to be published in the present form, so the authors should perform appropriate revisions sufficiently.

We thank the reviewer for the useful comments and suggestions, which will help us to provide a more accurate description of our work. Our responses are given below in red, after the reviewer's comments, which are in black. The changes in the text are marked in yellow.

Major comments:

Line 225: $NO_2$ and NOx concentrations measured by commercially available NOx analyzer include NOz species such as PAN and HNO3. I think this is a large problem because NO2 and NOx are important ozone precursors. If this is no problem for the authors, they should prove that there is no problem. For example, an intercomparison of NO2 concentrations measured by the CAPS and chemiluminescence methods should be performed.

Yes, we have used a commercially available chemiluminescent NOx monitor with the interference of $HNO_3$ and PANs on $NO_2$ measurement. However, we compared the $NO_2$ measured by the chemiluminescence NOx monitor with that measured by the Cavity Attenuated Phase Shift (CAPS, which is considered to be the more reliable $NO_2$ measurement technique without chemical interference) and found that a 5% bias could be caused by the chemiluminescence NOx monitor as shown in Zhou et al (2025). Therefore, we simulated $P(O_3)_{net}$ by reducing and increasing the mixing ratios of $NO_2$ by 5% to check the interference caused by using the chemiluminescence NOx monitor to modelled $P(O_3)_{net}$. The results show that increasing and decreasing $NO_2$ by 5% resulted in a decrease in $P(O_3)_{net}$ of 1.64% and 3.68%, respectively, which is much smaller than the bias caused by $P(O_3)_{net}$ in the reference chamber (~ 13.9%), these tests are shown in Hao et al. (2023). However, this won't affect the measured $P(O_3)_{net}$ values, as we used a CAPS $NO_2$ monitor (Aerodyne research, Inc., Billerica MA, USA) in the net photochemical ozone production rate (NPOPR) detection system to avoid such interference, and quantified Ox (=$O_3$+$NO_2$) differences in the reaction and reference chambers to correct the effects of fresh NO titration to $O_3$. We have specified the interference of $NO_2$ measurements using the chemiluminescence technique on pages 9, lines 263-269 in the amended manuscript:

"According to our test (Zhou et al., 2025), a 5% overestimation could be caused in the $NO_2$

measurement using the chemiluminescence technique compared to the CAPS technique, due to some $NO_Z$ species (i.e., $HNO_3$, peroxyacetyl nitrate (PANs), HONO, etc.)(Dunlea et al., 2007), this will result in a decrease of the modelled $P(O_3)_{net}$ by < 4%, which is negligible compared to the bias caused by the $P(O_3)_{net}$ in the reference chamber (~ 14%) (Zhou et al., 2023)."

Fig. 5: Why are there significant P(O3)net (not zero) in the nighttime? What is the precision of P(O3)net measured by this instrument? This should be discussed. Since Ox concentrations derived from the reaction chamber and reference chamber are measured alternately by solenoid valves, large fluctuations in ambient Ox concentrations are expected to cause poor precision.

According to our measurement error description added in "S4. The measurement error of $P(O_3)_{net}$ and the LOD of the NPOPR detection system", the uncertainty of the measured $P(O_3)_{net}$ is determined by the measurement error of $O_X$ of the CAPS-$NO_2$ monitor and the error caused by the light-enhanced loss of $O_3$ in the reaction and reference chambers, which is higher at lower $P(O_3)_{net}$ values (as shown in the updated Fig. 4). During the night, $P(O_3)_{net}$ is close to zero, but with a high uncertainty due to the instrument measurement error. As there is no light-enhanced $O_3$ loss in the reaction and reference chambers during the night, the uncertainty of the measured $P(O_3)_{net}$ is mainly determined by the ambient Ox concentrations, which can be considered as the measurement precision, and is estimated as ~38%. We have added the corresponding discussion on page 20, lines 535-543 in the revised manuscript:

"During nighttime, $P(O_3)_{net}$ should be zero without sun radiation. The significant $P(O_3)_{net}$ shown in Fig. 5 may be due to the measurement uncertainty of $P(O_3)_{net}$, which is determined by the measurement error of $O_X$ of CAPS-$NO_2$ monitor and the error caused by the light-enhanced loss of $O_3$ in the reaction and reference chambers (as discussed in Sect. S4). The measurement uncertainty of $P(O_3)_{net}$ is higher at lower $P(O_3)_{net}$ values (as shown in Fig. 4), which was mainly determined by the instrumental error of $O_X$ measurement and the ambient $O_X$ concentrations during nighttime. It was estimated to be ~ 38% and can be considered as the measurement precision."

Figs. S2 and 6: For P(O3)net, there are cases where the model agrees with the observation and cases where it does not. Why? The authors should discuss in depth? For IOA, NMB, and NME, the authors state their values during the whole measurement period only. What about the values of these parameters for each episode? Episode I and III may be good, but are the other episodes adequately reproduced, as described in Lines 571-573? Also, I think the discussion on the accuracy also concern the accuracy of the discussion on the vertical profiles of ozone budgets

and ozone production regime described in Figs. 7 and 9.

We discussed the relationship between the average daily disparities of the measured and modelled $P(O_3)_{net}$ ($\Delta P(O_3)_{net}$) with the various average daily NO concentrations during different episodes and non-episodes, which is depicted in Fig. 6f. The related discussion can be found on page 23, lines 637-644 in the main text:

"The observed elevated $\Delta P(O_3)_{net}$ at higher NO concentrations aligns with findings from previous studies, which suggest that multiple factors could contribute to these outcomes. For example, the reaction of OH with unknown VOCs (Tan et al., 2017), the lack of correction for the decomposition of $CH_3O_2NO_2$, the missing $RO_2$ production from photolysis $ClNO_2$ (Whalley et al., 2018; Tan et al., 2017), and the underestimation of OVOCs photolysis (Wang et al., 2022) in modelling approaches may lead to the underestimation of $RO_2$, thus underestimating the modelled $P(O_3)_{net}$."

| | Parameters | P1 | P2 | P3 | C1 | C2 |
|---|---|---|---|---|---|---|
| IOA | $P(O_3)_{net}$ | 0.89 [0.88,0.90] | 0.89 [0.88,0.90] | 0.96 [0.95,0.96] | 0.86 [0.86,0.87] | 0.91 [0.90,0.91] |
| | $O_3$ | 0.81 [0.80,0.82] | 0.79 [0.78,0.73] | 0.83 [0.81,0.84] | 0.81 [0.80,0.82] | 0.80 [0.79,0.80] |
| NMB | $P(O_3)_{net}$ | -0.33 [-0.36,-0.32] | -0.31 [-0.32,-0.31] | -0.12 [-0.14,-0.11] | -0.45 [-0.45,-0.44] | -0.26 [-0.27,-0.26] |
| | $O_3$ | 0.23 [0.23,0.24] | 0.28 [0.28,0.28] | 0.29 [0.27,0.32] | 0.24 [0.23,0.24] | 0.27 [0.78,0.80] |
| NME | $P(O_3)_{net}$ | 0.44 [0.43,0.47] | 0.43 [0.43,0.44] | 0.25 [0.24,0.26] | 0.56 [0.55,0.56] | 0.4 [0.38,0.40] |
| | $O_3$ | 0.30 [0.29,0.30] | 0.34 [0.34,0.34] | 0.31 [0.29,0.34] | 0.32 [0.31,0.33] | 0.33 [0.33,0.33] |

To better describe the IOA, NMB, and NME of the measured and modelled $P(O_3)_{net}$ (or $O_3$) values, we added the IOA, NMB, and NME values during different episodes and non-episodes, as shown in Table S3 in the modified supplementary material:

"**Table S3. The median values of IOA, NMB, and NME between measured and modelled $P(O_3)_{net}$ (or $O_3$) for different episodes and non-episodes.**
[x, y]: x, y represent 25% and 75% percentile values of IOA during different episodes and non-episodes, respectively.

The relevant discussion is added on page 23, lines 647-653 in the revised

manuscript:

"However, the derived IOA, NMB, and NME values from the modelled and observed $P(O_3)_{net}$ (and $O_3$) at 5 m ground level during different episodes and non-episodes indicate that the model proficiently reproduces the genuine $P(O_3)_{net}$ at the observation site well (as shown in Table S3). Consequently, these results provide confidence in exploring the vertical and temporal variations of the $P(O_3)_{net}$ and $O_3$ formation sensitivities utilizing the outcomes from the modelling approach. Nonetheless, it is important to acknowledge and discuss the potential biases induced by the modelling methodology in this study."

Other minor comments:

Line 61-63: The authors should explain ozone production regime in more detail.

Ok, we have added the definition of OBM-MCM on page 2-3, lines 63-67:

"A "NOx-limited" regime has higher VOCs/NOx ratios and the $O_3$ formation is sensitive to NOx concentration changes, while a "VOCs-limited" regime has lower VOCs/NOx ratios and the $O_3$ formation is sensitive to NOx concentration changes. In a "mixed-sensitive" regime, $O_3$ formation responds positively to changes in both NOx and VOC emissions (Wang et al., 2019)."

Line 100: The authors should define OBM-MCM.

We have added the definition of OBM-MCM on page 4, lines 104-107:

"To diagnose the net ozone production rate, $P(O_3)_{net}$, and $O_3$ formation sensitivities across various heights, we employed an observation-based model coupled with the Master Chemical Mechanism (MCM v3.3.1), hereafter referred to as OBM-MCM."

Sections 2.1 and 2.2.1: I think it would be easier for the readers to understand if the authors explain the details of the SZMGT and sampling method at SZMGT, using schematic diagrams in supplement.

We agree with the reviewer. Further details of the SZMGT have been added on pages

4, lines 125-128 in the revised manuscript:

"The SZMGT is 365 m high and is currently the tallest mast tower in Asia and the second tallest of this kind in the world. The main structure of the tower is made of steel, steel stray lines are used for fixing and securing the tower."

More details on the sampling method at SZMGT are added on page 5, lines 134-145 in the amended manuscript:

"A tower-based observation system for traces gases using long perfluoroalkoxy alkane (PFA) tubing (OD: 1/2") was used to sample the $O_3$ and $O_3$ precursors at six heights during the campaign, including 5, 40, 70, 120, 220, and 335 m above the ground. All six tubes were continuously drawn using a rotary vane vacuum pump to keep flushing with ambient air to reduce tube delay of the organic compounds, with the flow rate controlled by critical orifices (orifice diameter: 0.063"). A Teflon solenoid valve group was used to switch the air samples at specified time intervals so that the subsamples from these six heights could be sequentially drawn by instruments (see Fig. S1). Consequently, the flow rates of the air sample streams for the six tubes varied between 12.0 and 15.0 SLPM without subsampling and were less than 20 SLPM with subsampling. The residence time of the sample gas in the longest tube (~ 400 m) is less than 180 s at a flow rate of 13 SLPM."

And added the sampling schematic scheme diagram at SZMGT in the supplementary material:

[Figure]

**Figure S1. A simple schematic illustration of the vertical observation system on the SMT and locations of the six sampling inlets for measuring atmospheric gaseous species (Li et al., 2023).**

Line 149: O3 + NO = NO2 → O3 + NO → NO2 (This is a chemical reaction, not an equation)

We have changed "$O_3$ + NO = $NO_2$" to "$O_3$ + NO → $NO_2$" in the modified manuscript on page 6, line 169-172:

 "A stream of air from the two chambers was alternately introduced into an NO-reaction chamber every 2 min to convert $O_3$ in the air to $NO_2$ in the presence of high concentrations of NO ($O_3$+NO→$NO_2$), …"

Section 2.2.3: What kinds of VOCs did the authors measure? Listed in Table S2? If so, the authors should refer to Table S2 in the text.

We have added the description of Table S2 on page 8, lines 254-256:

". A full list of all 56 non-methane hydrocarbons (NMHCs) can be found in the supplementary material (Table S2)."

Line 233: at 424 nm → less than 424 nm?

We have changed the description on page 9, lines 274-276:

"The specific tropospheric $O_3$ photochemical formation process involves the photolysis of $NO_2$ at < 420 nm (Sadanaga et al., 2017)."

Line 289: In order to investigated → In order to investigate

We modified the description on page 11, lines 340:

"In order to investigate the influence of the photochemical reactions of different VOCs to photochemical $O_3$ formation, …"

Fig. 1 and Table 1: How did the authors measure CO and TVOCs? And the authors should define TVOCs.

Ok. We have added the measurement method for CO on page 9, line 259-263 in the revised manuscript:

"$O_3$, CO, and $NO_X$ concentrations were measured by a 2B $O_3$ monitor based on dual-channel UV-absorption (Model 205, 2B Technologies, USA), a gas filter correlation (GFC) CO analyzer (Model 48i, Thermo Fisher Scientific, USA), and a chemiluminescence NOx monitor (Model 42i, Thermo Fisher Scientific, USA), respectively."

And defined TVOC on page 14-15, line 404-406:

"The mean concentrations of $O_3$ precursors, including CO, NO, $NO_2$, and the total VOCs measured by PTR-TOF-MS (shown as TVOC in Fig. 1 and Table 1),…"

Line 363-365: Is this sentence made during the daytime?

Yes, we added "during daytime" to the sentence on page 15, line 430-432:

"From Fig. 2, minimal vertical gradients were observed during daytime in the concentration of all species–$O_3$, NOx, Ox, and TVOC–due to the rapid vertical mixing effects."

Line 454: Sect. 3.3.1 → Sect. 3.1.1?

Yes, it should read "Sect. 3.1.1". We have changed "Sect. 3.3.1" to "Sect. 3.1.1" on page 18, line 508-509:

"As concluded in Sect. 3.1.1, $O_3$ pollution episodes may be jointly affected by the

photochemical reactions and physical transport."

---

## Author Comment (AC2)

Review for: Measurement report: Vertical and temporal variability of near-surface ozone production rate and sensitivity in an urban area in Pearl River Delta (PRD) region, China.

We are grateful for the reviewer's comments and suggestions, which will help us to provide a more accurate description of our work. Our responses are given below in red, after the reviewer's comments, which are in black. The changes in the text are marked in yellow.

Summary:

This manuscript presents a detailed study of the vertical and temporal profile of ozone production, using both measurement data and chemical box modelling. Overall, the manuscript is well written with a good flow. However, there are some grammatical issues that should be addressed before publication. I really enjoyed reading this manuscript, and feel that it would be of great interest to the urban ozone community, therefore I recommend it for publication to ACP provided some important changes to the text have been made.

Of particular concern is the use of the phrase "statistically different/difference" throughout the text. When this phrase is used, I would expect it to be backed up by a statistical test(s). I don't think any further analysis is required, but the authors should re-think how they describe their findings and avoid this phrase when there is no statistical evidence to back up their claims. I would also recommend another detailed read-through of the document to check for grammatical errors that occur periodically. Please see below for more detailed comments.

Thank you for your careful review and useful suggestions. We have checked the grammar throughout the manuscript and addressed the grammatical issues. We have also improved the presentation of the results of the statistical analyses, avoiding the overuse of the phrase "statistically different/difference" throughout the text.

Detailed comments:

56 Replace "have" with "has".

We have replaced "have" with "has" on page 2, line 56.

65 Check grammar.

We have checked the grammar and modified the sentence on page 3, lines 67-72 in the modified manuscript:

"Local $O_3$ concentrations can be further influenced by meteorological conditions and the regional transport of $O_3$ and its precursors (Gong and Liao, 2019; Chang et al., 2019). The Pearl River Delta (PRD) stands out as one of the most rapidly developing economic and urbanized regions in China, which currently is suffering from severe ground-level $O_3$ pollution (Lu et al., 2018; Yang et al., 2019)."

71 Check grammar.

We have checked the grammar and changed the sentence on page 3, lines 72-80 in the revised manuscript:

"Currently, many scholars have analyzed the relationship between tropospheric ozone pollution and its precursors and meteorological elements in the PRD region (Mao et al., 2022; Li et al., 2022a), which has greatly improved our understanding of the sources and formation processes of $O_3$ in the PRD region. However, the distribution of $O_3$ is highly variable at different altitudes (Wang et al., 2021), due to vertical differences in VOCs concentrations and sources, as well as the sensitivity of $O_3$ formation (Liu et al., 2023; Tang et al., 2017)."

75 Please explain why using only one height of measurements is of great limitation. Why is it important that this is done at multiple altitudes, when $O_3$ exposure occurs at ground level?

In the boundary layer, the surface heating leads to strong vertical mixing during daytime, so ozone formation at higher altitudes may also influence the $O_3$ budgets and its exposure at ground level. Additionally, the vertical gradients of $O_3$ precursors may drive the change in the photochemical formation regimes of ozone in vertical directions (Zhao et al., 2019). We have added the corresponding description on page 3, lines 80-85 in the revised manuscript:

"Due to the presence of strong vertical mixing driven by the surface heating effect in the daytime boundary layer, the budget of the ozone at the ground level and also at an arbitrary height in the daytime boundary layer is closely related to the formation and removal of ozone at other heights (Tang et al., 2017). In addition, the difference in vertical gradients of precursors may drive the vertical change in the photochemical formation regimes of ozone (Zhao et al., 2019)."

81 Check grammar.

We have checked the grammar and changed the sentences on pages 3, lines 89-94 in the revised manuscript:

"Currently, remote sensing techniques with high time resolution and real-time response, such as lidar and optical absorption spectroscopy, have been utilized to measure the vertical distribution of $O_3$ (Luo et al., 2020a; Wang et al., 2021). However, in situ measurements of VOCs at various heights primarily rely on offline methods combined with diverse techniques, including aircraft, tethered balloons, tall buildings and towers, unmanned aerial vehicles (UAVs or drones), and satellite observations"

85 This sentence is very long – check grammar and split into two after "VOC and NOx measurements".

We have checked the grammar and changed the sentence on page 3-4, lines 96-100 in the modified manuscript:

"Owing to the low time resolution of these monitoring techniques, achieving continuous vertical coverage of VOCs and NOx measurements is challenging. Consequently, the vertical distribution structure of VOCs remains unclear, thus largely hindering our understanding of the vertical and temporal regional ozone formation mechanism."

113 Misspelled "Asia".

We modified the word of "Asia" on page 5, lines 123.

116 Check grammar.

We have checked the grammar and changed the sentence on page 4, lines 127-128 in the revised manuscript:

"The area is surrounded by a high density of vegetation, reservoir features, lowrise buildings, and hills/mountains (Luo et al., 2020b)."

224 Please state in the text that a full list all 47 NMHCs s available in the supplementary (Table S2).

Thank you for the careful checking. We have found an error in this description, we actually measured 56 non-methane hydrocarbons (NMHCs), instead of 47 NMHCs. We have added this statement on page 8, lines 254-256 in the modified manuscript:

"A full list of all 56 non-methane hydrocarbons (NMHCs) can be found in the supplementary materials (Table S2)."

As well as page 9, lines 287-288:

"56 NMHCs (toluene, benzene, isoprene, styrene, etc., as listed in Table S2),…"

230 Please include in this section the dilution / ventilation approach you have included in the model.

Ok. We have added the description of the dilution factor throughout the modelling period on page 10, lines 292-296 in the revised manuscript:

"The effect of physical processes (such as vertical and horizontal transport) was considered by setting a constant dilution factor of $1/43200$ s$^{-1}$ throughout the modelling period. Additionally, the dry deposition rate of $O_3$ was set to 0.42 cm s$^{-1}$ and the background concentrations of $O_3$, $CO$, and $CH_4$ were set to 30, 70, and 1800 ppbv, respectively."

333 I advise that you should avoid using the word "statistically", unless you have performed a statistical test. Please rephrase.

Thank you for your advice. We have changed "statistically" to "significantly" on page 13, line 393 in the revised manuscript.

346 Again, if you use the phrase "statistically different", I would expect to see evidence of a statistical test showing this.

Ok, we changed the description on page 14-15, lines 404-406:

"The mean concentrations of $O_3$ precursors, including CO, NO, $NO_2$, and the total VOCs measured by PTR-TOF-MS (shown as TVOC in Fig. 1 and Table 1), did not exhibit notable discrepancies between episodes and non-episodes."

351 I would rephrase this. Are you saying that event and non-event OFP and P(O3)net are not statistically different because they are within one standard deviation?

We apologize for the confusing description. We meant that the averaged OFP and $P(O_3)_{net}$ during $O_3$ pollution events and non-events showed no difference when considering $\pm 1\sigma$ obtained from their average calculation. In other words, the averaged OFP$\pm 1\sigma$ (or averaged $P(O_3)_{net}\pm 1\sigma$) obtained during episodes are not significantly different from those obtained during non-episodes, they fall in the same range within $1\sigma$. We have changed this statement on page 15, lines 408-413, in the revised manuscript:

"Further comparison of the daytime mean $O_3$ formation potential (OFP) and the measured $P(O_3)_{net}$ during episodes and non-episodes showed no significant differences, ranging from 5.1E-4 to 1.0E-3 g m$^{-3}$ and 14.3 to 21.5 ppb h$^{-1}$, respectively, during non-episodes, whereas they are ranged from 4.1E-4 to 4.7E-4 g m$^{-3}$ and 5.6 to 18.9 ppb h$^{-1}$ respectively, during episodes."

Further explanations are provided in the response to comments 352 and 644 below.

352 It is not clear to me how you have come to this conclusion (These findings indicate that…). Please clarify in the text.

We apologize for the confusing description. As shown in the response to comments 351 and 644, we meant to say that the daytime averaged $O_3$ formation potential (OFP) and the measured $P(O_3)_{net}$) ranged from 5.1E-4 to 1.0E-3 g m$^{-3}$ and 14.3 to 21.5 ppb h$^{-1}$, respectively, during non-episodes, whereas they are ranged from 4.1E-4 to 4.7E-4 g m$^{-3}$ and 5.6 to 18.9 ppb h$^{-1}$, respectively, during episodes. Although OFP during episodes was always higher during episodes than that during non-episodes, $P(O_3)_{net}$ during episodes can be higher or lower than that during non-episodes, as shown in Table 1. This demonstrates that the $O_3$ pollution episodes are not always due to local photochemical $O_3$ formation (represented as $P(O_3)_{net}$). For example, $P(O_3)_{net}$ is lower during episodes I and III than during non-episode II, which may be due to the much less stable weather conditions during episodes III (with lower wind

speed), favoring the accumulation of $O_3$ formed by local photochemical $O_3$ formation. While for non-episode II even higher $P(O_3)_{net}$ is processed, the average $O_3$ concentration is still lower than that during episodes I and III, which may be due to the outflow of $O_3$ from the observation site by physical processes. Therefore, we conclude that the $O_3$ pollution episodes are either due to significantly increased local photochemical $O_3$ formation (i.e., episode II), or to the accumulation of $O_3$ formed by moderate local photochemical $O_3$ formation under stable weather conditions (i.e., episodes I and II). To make the sentences clearer, we have added the following explanations on pages 15, lines 414-419 in the revised manuscript:

"Although OFP was always higher during episodes than during non-episodes, the mean $P(O_3)_{net}$ values during episodes I and III were even lower than during non-episodes II. The higher $O_3$ concentrations may be due to the more stable weather conditions during episodes I and III (with lower wind speed), which benefits the accumulation of $O_3$ formed by local photochemical $O_3$ formation. While for non-episode II, even it processes higher $P(O_3)_{net,}$ the outflow of $O_3$ from the observation site by physical processes may be higher due to the higher wind speed. These findings indicate that the $O_3$ pollution episodes stem from either substantially elevated local photochemical $O_3$ formation (i.e., episode II), or the accumulation of $O_3$ formed by moderate local photochemical $O_3$ formation under stable weather conditions (i.e., episodes I and II)."

358 It is well known that O3 pollution episodes are jointly affected by photochemical reactions and physical transport processes. It's unclear to be how what you have said previously has led you to this statement.

According to the similar $P(O_3)_{net}$ average values obtained on episode and non-episode days (as described above, they were not statistically different within one standard deviation), we concluded that the $O_3$ pollution episodes stem from either substantially elevated local photochemical $O_3$ formation (i.e., episode II ), or the accumulation of $O_3$ formed by moderate local photochemical $O_3$

formation under stable weather conditions (i.e., episodes I and Ⅱ). On the other hand, if there is an outflow of $O_3$ from the observation site (which can be considered as physical transport) due to favorable weather conditions, the intense local photochemical reactions may not lead to the $O_3$ pollution (i.e., non-episode Ⅱ). Therefore, we have left to the statement that the $O_3$ pollution episodes in this study are jointly affected by the photochemical reactions and physical transport processes. We have changed this statement on page 15, lines 425-427 in the revised manuscript:

"These results indicate that $O_3$ pollution episodes are jointly affected by the photochemical reactions and physical transport processes, which we will discuss in more detail in Sect. 3.2.1."

438 Remove word "besides" – not needed.

Ok, we removed the word "besides" accordingly.

476 Instead of saying "the results show", be specific about which part of the figure you are referring to in the text. What is it in the figure that has led you to this conclusion?

Ok, thank you for your suggestion. We have changed this sentence on page 19-20, lines 531-535:

"$R(O_X)_{trans}$ at 5 m ground level was derived from $\frac{dO_X}{dt}$ manus $P(O_X)_{net}$, according to Eq. (5) shown Sect. 2.3.2, their hourly averages and diurnal variations are shown in Figs. 4 and 5, respectively. From these figures, it is evident that the fluctuation of the $O_3$ concentration change rate ($d(O_3)/dt$) at ground level is typically small and primarily dominated by the physical processes during nighttime."

478 How do you know this is attributed to physical transport? Is this just a suggestion based on knowledge of the atmosphere (please supply a reference), or is there something in the figure that provides evidence that increasing O3 is due to vertical transport, and not early morning photochemistry? If it's because you have not measured any P(O3)net at this time, please state this in the text.

Sorry for the confusing description. We reached this conclusion because of the

diurnal variation of the contribution of chemical and physical transport to the $O_3$ changes at the ground level, as shown in Fig. 5. After a careful check, we realize that around 6:00-7:00 LT, $O_3$ concentrations increase for all episodes and non-episodes, mainly due to physical transport during episodes I and II and non-episodes I, while photochemical reactions and physical processes are equally important for episodes III and non-episode II. We have changed the description on page 20, lines 543-553 to make the description more accurate:

"Around 6:00-7:00 LT, $O_3$ concentrations increase for all episodes and non-episodes, mainly due to physical transport during episodes I and II and non-episodes I, while photochemical reactions and physical processes are equally important for episodes III and non-episode II. This could be due to short-term strong vertical turbulence in the early morning, which leads to an expansion of the boundary layer height and makes the residual layer "leaky", allowing vertical transport. At the same time, $O_3$ precursors were also transported down from the residual layer, and with increasing sunlight, these $O_3$ precursors underwent rapid photochemical reactions that competed with the physical processes between 6:00-7:00 LT, leading to a sharp increase in $P(O_3)_{net}$ between 8:00 to 12:00 LT."

483 Change "residue" to "residual".

We have changed "residue" to "residual" on page 20, line 548.

644 "statistically difference" – see comments for lines 333 and 346

Thank you for the suggestions. We have added the Mann-Whitney tests, and found that the differences in the measured/modelled $P(O_3)_{net}$ during episodes and non-episodes are not statistically different, with the Mann-Whitney $p$-value=0.12 and 0.28 for measured and modelled $P(O_3)_{net}$, respectively. We have added such explanations on page 24-25, lines 692-695 in the modified manuscript:

"Consequently, the modelled $P(O_3)_{net}$ during episodes does not exhibiting a statistically significant difference from that during non-episodes (Mann-Whitney $p$ value=0.12), as shown in Fig. S5, which is in agreement with the measured $P(O_3)_{net}$ (Mann-Whitney $p$-value=0.28), as depicted in Sect. 3.1.1."

659 In the text, please direct the reader to Table S2 which defines which VOCs are in each category (NMHC, AVOC, BVOC and OVOC).

Thank you for the suggestion. We have added a sentence on page 26, lines 734-737 to refer the reader to Table S2:

"including nonmethane hydrocarbons (NMHC), anthropogenic volatile organic compounds (AVOC), biogenic volatile organic compounds (BVOC), and oxygenated volatile organic compounds (OVOC) (as shown in Fig.8f). The VOCs species included in each category are listed in Table S2."

698 Check grammar in this section.

We have checked the grammar in this section accordingly and corrected any errors.

816 Remove "proper" – not needed.

Ok, we have removed "proper" accordingly.

References:

Zhao, W., Tang, G., Yu, H., Yang, Y., Wang, Y., Wang, L., An, J., Gao, W., Hu, B., Cheng, M., An, X., Li, X., and Wang, Y.: Evolution of boundary layer ozone in Shijiazhuang, a suburban site on the North China Plain, J. Environ. Sci., 83, 152-160, 10.1016/j.jes.2019.02.016, 2019.

---

## Author Comment (AC3)

The manuscript of Zhou et al. reports novel information on the $O_3$ budget and chemistry at a monitoring site in the PRD region (China). This work takes advantage of ozone production rates measurements performed at the ground level during one month in Nov.-Dec. 2021 to infer the contributions of both photochemical and transport processes to the local ozone budget. In addition, the authors use a rich dataset of trace gas measurements performed at multiple heights (up to 335m) and 0-D box modeling to investigate the vertical distribution of $P(O_3)$. The ozone formation chemistry is investigated at the different heights, highlighting that ozone production occurs in the VOC-limited and transition regimes most of the time. The authors conclude that emission regulations focusing on the reduction of AVOCs and OVOCs should help reducing $O_3$ at this location.

We thank the reviewer's comments and suggestions, which will assist us in providing a more accurate description of our work. Our answers are listed in the following in red, after the reviewer's comments, which are in black. The modifications in the text are marked in yellow.

While the manuscript is well structured, the writing needs to be revised before publication. Some suggestions are provided below but there are more instances in the manuscript where improvements are needed. The methodology and the results seem scientifically sound and the authors provide novel information that will be of interest for the atmospheric community. I recommend publication after the authors have addressed the writing issues and the following comments:

Thanks to the reviewer's careful checking, we have revised the writing throughout the manuscript and changed the incorrect descriptions and grammar.

**Major comments:**

- L172-178: This section is confusing. The authors indicate that a light-enhanced loss of $O_3$ is corrected for but present an equation to compute an uptake coefficient for $O_3$. It's not clear how the correction is done. The authors should clarify how this uptake coefficient is derived and how it is considered when computing $P(O_3)_{net}$ from Eq. 1. The amplitude of this correction should also be clearly stated. It would be useful to add time series of P(O3)net with and without correction in the supplementary material to show how this correction changes over time.

We apologize for the unclear description. We have added the light-enhanced $O_3$ loss quantification method and the amplitude of this correction in the amended supplementary materials on page 9:

"**S3. The experiments concerning the light-enhanced loss of $O_3$**

***The light-enhanced loss of $O_3$ in the reaction and reference chambers*** at 5 L min$^{-1}$ (the flow rate used during the observation campaign in this study) was investigated by carrying out the following experiment: the $O_3$ was injected at a mixing ratio of approximately 130 ppbv generated by the $O_3$ generator (P/N 97-0067-02, Analytic Jena US, USA) to ensure that no photochemical $O_3$ was produced during the outdoor experiment. The $J(O^1D)$, $T$, RH, $P$ and $O_3$ mixing ratios at the inlet and outlet of the reaction and reference chambers were measured simultaneously. The $T$ and RH were measured with a thermometer (Vaisala, HMP110, USA). The light-enhanced loss coefficient of $O_3$ ($\gamma$) was calculated using Eq. (4) described in the main text, and the relationship between $J(O^1D)$ and $\gamma$ is shown in Fig. S8a. The obtained $\gamma$-$J(O^1D)$ equation listed in Eq. (4) was used to correct for the light-enhanced loss of $O_3$ in the reaction and reference chambers during the daytime to exclude the influence of light-enhanced loss. The change in the $O_3$ mixing ratio after correcting for the light-enhanced loss of $O_3$ ($d[O_3]$) showed no clear correlation with RH for both the reaction and reference chambers, as shown in Fig. S8b, indicating that the RH had no influence on the change in the $O_3$ mixing ratio during the observation period.

[Figure]

**Figure S8: The relationship between (a) $\gamma$ and $J(O^1D)$ and (b) RH and $d[O_3]$ in the reaction and reference chambers, calculated from the 68.3 % confidence interval of the fit lines between $\gamma$ and $J(O^1D)$), the shaded areas represent the maximum range of fluctuation under this confidence level.**"

Accordingly, we have added the $d[O_3]$ correction method on page 6-7, lines 188-198 in the revised manuscript:

"where $\gamma$ is the light-enhanced loss coefficient of $O_3$, which is derived from $J(O^1D)$ according to the relationship obtained from the outdoor experiments (for more details, see supplementary materials: S3.). $d[O_3]$ represents the difference between the $O_3$

mixing ratios at the inlet and outlet of the reaction and reference chambers, D is the diameter of the chambers, ω is the average velocity of $O_3$ molecules, $[O_3]$ is the injected $O_3$ mixing ratio at the inlet of the reaction and reference chambers, and $\tau$ is the average residence time of the air in the reaction and reference chambers. When quantifying the light-enhanced $O_3$ loss ($d[O_3]$) during the ambient air measurement, we first calculate $\gamma$ using the measured $J(O^1D)$ and the $\gamma$ -$J(O^1D)$ equations listed in Fig. S8 in the reaction and reference chambers, then use the measured $[O_3]$ and Eq. 2 to calculate $d[O_3]$."

Furthermore, we quantified the amplitude of this correction by comparing the $P(O_3)_{net}$ with and without the correction for the light-enhanced loss of $O_3$; the corresponding time series are shown in Fig. S9 in the supplementary material to show how this correction changes with time:

"Furthermore, we quantified the light-enhanced loss of $O_3$ correction by comparing the $P(O_3)_{net}$ with and without the correction, the corresponding time series are shown in Fig. S9. Results show that such a correction can increase the measured $P(O_3)_{net}$ by 10% (25% percentile) to 24% (75% percentile), with the median value of 17%.

[Figure]

**Figure S9: The time series of $P(O_3)_{net}$ with and without the light-enhanced loss of $O_3$ correction.**"

The corresponding description is added on page 7, lines 198-199 in the modified manuscript:

"The results show that such kind of correction can increase the measured $P(O_3)_{net}$ by 10% (25% percentile) to 24% (75% percentile), with a median of 17%."

- L180-181: "The limit of detection (LOD) of the NPOPR detection system is 2.3 ppbv h-1 at the sampling air flow rate of 5 L min-1,

which is obtained as three times the measurement error of P(O3)net." – It's not clear how the authors derive a LOD from the error associated to P(O3)net. This error will scale with P(O3)net. Please clarify.

Ok, we have clarified the error associated with $P(O_3)_{net}$ and described the derive method in detail in the supplementary material on pages 9-11:

"**S4. Measurement error of $P(O_3)_{net}$ and the LOD of the NPOPR detection system**

According to the $P(O_3)_{net}$ evaluation method listed in Eq. (1) in the main text, the measurement error of $P(O_3)_{net}$ depends on the estimation error of Ox in the reaction and reference chambers, which includes the measurement error of $O_X$ of CAPS-NO$_2$ monitor and the error caused by $\gamma$, and can be calculated according to Eq. (S1) :

$$(O_X)_{error} = \sqrt{(O_{X_\gamma})_{error}^2 + (O_{X_{CAPS}})_{error}^2} \qquad (S1)$$

where $(O_{X_\gamma})_{error}$ and $(O_{X_{CAPS}})_{error}$ represents the $\gamma$ corrected error of the Ox of the reaction and reference chambers and the measurement error of the $O_X$ of the CAPS-NO$_2$ monitor, respectively. The measurement error of the CAPS NO$_2$ monitor was obtained by fitting the NO$_2$ calibration results with a 68.3 % confidence level. The blue line in Fig. S10 represents the maximum range of fluctuation under this confidence level.

[Figure]

**Figure S10: Calibration results of the CAPS NO₂ monitor at different NO₂ mixing ratios. The y-axis represents the NO₂ mixing ratios measured by the CAPS NO₂ monitor, and the x-axis represents the prepared NO₂ mixing ratios prepared from the diluted NO₂ standard gas.**

$(O_{X_{CAPS}})_{error}$ was then calculated from the fluctuation range of the 68.3 % confidence

interval of the calibration curve. The relationship between $(O_{X_{CAPS}})_{error}$ and the measured Ox value ($[Ox]_{measured}$) can be expressed as a power function curve, as shown in Eq. (S2):

$$(O_{X_{CAPS}})_{error} = 9.72 \times [O_X]_{measured}^{-1.0024} \tag{S2}$$

Dry pure air was sequentially introduced into the NPOPR detection system for ~ 2 h to adjust the system, followed by dry pure air or ambient air when the CAPS $NO_2$ monitor time resolution was 1 s and the integration period was 100 s (the measurement durations for the reaction and reference chambers were both 2 min).

The measured $O_X$ errors may also be influenced by the light-enhanced loss of $O_3$ in the reaction and reference chambers under ambient conditions when the light intensity (especially $J(O^1D)$) and $O_3$ mixing ratios are high. Therefore, when injecting ambient air into the NPOPR system, the error of $P(O_3)_{net}$ with a residence time of $\tau$ can be calculated using Eq. (S3):

$$P(O_3)_{net\_error} = \frac{\sqrt{(O_{X_\gamma})_{rea\_error}^2 + ((9.72 \times [(O_X]_{rea\_measured}^{-1.0024})_{rea\_std})^2 + (O_{X_\gamma})_{ref\_error}^2 + ((9.72 \times [(O_X]_{ref\_measured}^{-1.0024})_{ref\_std})^2}}{\tau}$$

(S3)

where $(O_{X_\gamma})_{rea\_error}$ and $(O_{X_\gamma})_{ref\_error}$ represent the measurement error due to light-enhanced loss of $O_3$ in the reaction and reference chambers, respectively, and $(9.72 \times [O_X]_{measured}^{-1.0024})_{rea\_std}$ and $(9.72 \times [O_X]_{measured}^{-1.0024})_{ref\_std}$ represent the standard deviation of $O_X$ in the reaction and reference chambers, respectively, caused by the CAPS $NO_2$ monitor with an integration time period of 100 s. Combined with the associated residence time $\langle \tau \rangle$ under different flow rates, i.e., $\langle \tau \rangle$ was 0.063 h at a flow rate of 5 L min$^{-1}$.

The LOD of the NPOPR detection system was determined to be three times $P(O_3)_{net\_error}$. Since the measurement error of the CAPS $NO_2$ monitor decreases with increasing Ox mixing ratios (as shown in Eq. S2), higher LODs could be obtained when injecting dry pure air into the NPOPR detection system, which were approximately 0.07, 1.4, and 2.3

ppbv h⁻¹ at air flow rates of 1.3, 3, and 5 L min⁻¹, respectively. Given that the background

ppbv h$^{-1}$ at air flow rates of 1.3, 3, and 5 L min$^{-1}$, respectively. Given that the background O$_X$ mixing ratios (measured by the CAPS NO$_2$ monitor of the air in the reference chamber) changed when the ambient air was measured, the measured O$_X$ errors in the reaction and reference chambers changed with the Ox mixing ratios, and the LOD must also be a function of the intrinsic ambient and photochemically formed O$_3$ and NO$_2$ mixing ratios (i.e., the Ox mixing ratios measured by the CAPS NO$_2$ monitor)."

Accordingly, we have added the relevant description to the main text on pages 7, lines 202-205:

"More details about the measurement error of $P(O_3)_{net}$ are described in the supplementary materials: S4: The measurement error of $P(O_3)_{net}$ and the LOD of the NPOPR detection system. More details can be found in our previous work (Hao et al., 2013)."

- L194-195: "Therefore, we corrected the measured P(O3)net using the quantified P(O3)net in the reference chamber." – How was P(O3)net quantified in the reference chamber?

We apologize for the confusing description. We quantified $P(O_3)_{net}$ using the modelled values of $P(O_3)_{net}$ in the reference chamber according to the method described in our previous study (Hao et al., 2023). We have added such descriptions in Sect. 2.2.2 (page 7, lines 210-216):

"According to our previous investigation, the modelled $P(O_3)_{net}$ in the reaction chamber is similar to that modelled in ambient air, with the modelled $P(O_3)_{net}$ in the reference chamber accounting for 0-13.9% of that in the reaction chamber (Hao et al., 2023). Here, we employed the same modelling method described in Hao et al. (2013) to quantify the $P(O_3)_{net}$ in the reference chamber and corrected the bias caused by the $P(O_3)_{net}$ in reference chamber accordingly (more details can be found in Sect. 2.2.1)."

- L223-225: The authors should address the specificity of their NO2 measurements since chemiluminescence instruments also detect some NOy species in the NO2 channel. Were O3, NO and NO2 corrected for O3+NO→ NO2 in the sampling line? If so, please indicate the amplitude of this correction.

Thanks for the suggestion. We used a commercially available chemiluminescence NOx monitor with the interference of HNO$_3$ and PANs on NO$_2$ measurement, and

compared the $NO_2$ measured by the chemiluminescence NOx monitor with that measured by the Cavity Attenuated Phase Shift (CAPS, which is considered to be a reliable $NO_2$ measurement technique without chemical interference) and found that a 5% bias could be caused by the chemiluminescence NOx monitor, which is shown in Zhou et al (2025). Therefore, we simulated $P(O_3)_{net}$ by reducing and increasing the mixing ratios of $NO_2$ by 5% to check the interference caused by using the chemiluminescence NOx monitor to model $P(O_3)_{net}$. The results show that increasing and decreasing $NO_2$ by 5% resulted in a decrease in $P(O_3)_{net}$ of 1.64 % and 3.68 %, respectively, which is much smaller than the bias caused by $P(O_3)_{net}$ in the reference chamber (~ 13.9%), these tests are shown in Hao et al. (2023). However, this won't affect our measured $P(O_3)_{net}$ values, as we used a CAPS $NO_2$ monitor (Aerodyne research, Inc., Billerica MA, USA) in the net photochemical ozone production rate (NPOPR) detection system to avoid such interference, and quantified Ox ($=O_3+NO_2$) differences in the reaction and reference chambers to correct for the effects of fresh NO titration to $O_3$. We have specified the interference of $NO_2$ measurements using the chemiluminescence technique on pages 9, lines 263-269 in the amended manuscript:

"According to our test (Zhou et al., 2025), a 5% overestimation could be caused in the $NO_2$ measurement using the chemiluminescence technique compared to the CAPS technique, due to some $NO_Z$ species (i.e., $HNO_3$, peroxyacetyl nitrate (PANs), HONO, etc.)(Dunlea et al., 2007), this will result in a decrease of the modelled $P(O_3)_{net}$ by < 4%, which is negligible compared to the bias caused by the $P(O_3)_{net}$ in the reference chamber (~ 14%) (Zhou et al., 2023)."

And indicated the quantification method for $O_3$ in the reaction and reference chamber on pages 6, lines 166-168 in the modified manuscript:

"To correct for the effect of fresh NO titration to $O_3$, we use $O_X$ ($=O_3+NO_2$) instead of $O_3$ to quantify the $O_3$ generated by photochemical reactions (Pan et al., 2015; Tan et al., 2018)."

- Section 2.2.3: The authors should provide some details about the GC measurements. It is stated that an offline GC was used. How were the VOCs sampled? How were the sampled analyzed? How and how often were calibration and zeroing done on the GC instrument?

OK, we have added more details about offline GC measurements on page 8, lines 243-256:

"For the off-line GC-MS-FID measurement, whole-air samples were collected using 3.2

L electro-polished stainless-steel canisters (Entech, USA) at 5 and 120 m at time intervals of two hours. Two automatic canister samplers connected to 12 canisters were used to collect the whole-air samples, with each of canister collecting the sample for 10 min. The canisters were analyzed within one week (Zhu et al., 2018). The concentrations of 56 NMHC species in the canister analyzed by GC-MS/FID were calibrated daily using the mixture of a photochemical assessment monitoring stations (PAMS) standard gas and pure $N_2$. In addition, the mixture of PAMS standard gas and pure $N_2$ with species concentrations of 1 ppbv was injected into the analytical system every 10 samples to check the operational stability of the instrument. Pure $N_2$ was injected into the analytical system at the start and end of each day's analysis to provide reference blank measurements. A full list of all 56 non-methane hydrocarbons (NMHCs) can be found in the supplementary material (Table S2)."

- Section 2.3: The authors should add a subsection to explain how Ozone Formation Potential (OFP) values are computed. OFP values are reported in Table 1 and Figure 3, and discuss in the result section.

In deeded. We have added "Sect. 2.3.5 $O_3$ formation potential" on pages 12, lines 351-357:

"The ozone formation potential is calculated using the product of the VOCs concentration and the maximum incremental reactivity (MIR) coefficient (dimensionless, gram of $O_3$ produced per gram of VOCs) (Carter et al., 2012):

$$OFP_i = \sum_i [VOC]_i \times MIR_i \tag{10}$$

Where $OFP_i$ is the ozone formation potential of species $i$, $[VOC]_i$ is the mass concentration or emission of species $i$, and $MIR_i$ denotes the maximum increment reactivity of species $i$ (g $O_3$/g VOCs)."

- Section 2.3.1: The authors should indicate which chemical species were constrained in the box model and how they were constrained. It is not clear how the authors deal with ozone in the model. In section 3.2.2, the authors compare simulated ozone concentrations to field observations, which seems to indicate that measured ozone concentrations were not

directly constrained in the model. However, if the O3 advected to the site was not constrained, the simulations would very likely not reproduce the measured ozone concentrations. How did the authors constrain the ozone transported to the observation site in their model?

Yes, we did not constrain the $O_3$ concentrations in our model, nor the $O_3$ advected to the site. We used a constant dilution factor of 1/432000 s$^{-1}$ throughout the modeling period and a sedimention factor of 0.42 cm s$^{-1}$ for the physical processes of $O_3$. We also added an $O_3$ background of 30 ppbv to represent the ozone transported to the observation site. We added the detailed chemical species constrained method in Sect. 2.3.1:

"$P(O_3)_{net}$ and $O_3$ concentrations were simulated by constraining $T$, RH, $P$, organic and inorganic substances in gases, including 12 OVOCs (methanol, ethanol, formaldehyde, acetaldehyde, acrolein, acetone, hydroxyacetone, phenol, $m$-cresol, methyl vinyl ketone, methacrylaldehyde, methyl ethyl ketone), 56 NMHCs (toluene, benzene, isoprene, styrene, etc. as listed in Table S2), conventional pollutants ($O_3$, NO, $NO_2$, and CO), and photolysis rate values ($J(O^1D)$, $J(NO_2)$, $J(H_2O_2)$, $J(HONO)$, $J(HCHO\_M)$, $J(HCHO\_R)$, $J(NO_3\_M)$, $J(NO_3\_R)$, etc.). The VOCs, NOx, $T$, RH and $P$ were constrained throughout the modelling period, while $O_3$ was not constrained after providing initial concentration values. The effect of physical processes (such as vertical and horizontal transport) was considered by setting a constant dilution factor of 1/43200 s$^{-1}$ throughout the modelling period. Additionally, the dry deposition rate of $O_3$ was set to 0.42 cm s$^{-1}$, and the background of $O_3$, CO, and $CH_4$ were set to 30, 70, and 1800 ppbv, respectively. The modelling was run in a time-dependent mode with a resolution of 5 min, and it was run for spin-up time of 72 h to establish steady-state concentrations for secondary pollutants that were not constrained during the simulation."

- Figure 4: The authors should indicate how R(O3)tran is derived. Is it computed as d[O3]/dt – P(O3)net ? Please add error bars on the time series of d[O3]/dt, P(O3)net and R(O3)tran.

Yes, the $R(O_3)_{trans}$ is computed as $\frac{dO_x}{dt}$ manus $P(Ox)_{net}$, we have added how $R(O_3)_{trans}$ is derived on page 19-20, lines 531-535 in the revised manuscript:

"$R(O_X)_{trans}$ at 5 m ground level was derived from $\frac{dO_X}{dt}$ manus $P(Ox)_{net}$ according to Eq. (5) shown Sect. 2.3.2, their hourly averages and diurnal variations are shown in Figs. 4 and 5, respectively. From these figures, it is evident that the fluctuation of the $O_3$ concentration change rate $(d(O_3)/dt)$ at ground level is typically small and primarily dominated by the physical processes during nighttime."

We have added the error bars to the time series of $d[O_3]/dt$, $P(O_3)_{net}$ and $R(O_3)_{tran}$ in Fig. 4:

[Figure]

**Figure 4. Time series of $O_3$ concentration changes $(d(O_3)/dt)$ and contributions from local photochemical production $(P(O_3)_{net})$ and physical transport $(R(O_3)_{tran})$. The shaded areas of $d(O_3)/dt$, $P(O_3)_{net}$, and $R(O_3)_{tran}$ represent one standard deviation (denoted by $\sigma$) of the mean $d(O_3)/dt$, the uncertainty of measured $P(O_3)_{net}$, and the propagated error of $R(O_3)_{tran}$, respectively.**

- L535: Dilution was constrained in the model using a species lifetime of 12h. How sensitive are the simulation results to this parameter? Please indicate how modelled P(O3) changes when the lifetime is varied from 6 to 24 h. How does it affect the main conclusions?

To achieve the best agreement between the modelled $O_3$ concentrations and the observed values, we applied different dilution factors (the species lifetime) in the modelling, which varying from 6 h to 24 h. We found that the simulated $O_3$ is closest to the measured $O_3$ concentrations when we set the species lifetime to 12 h. We found that the modelled $P(O_3)_{net}$ increases with increasing dilution factor, but this doesn't affect our main conclusions as the influence of the dilution factor on the modelled $P(O_3)_{net}$ is negligible due to the very short lifetime of the $HO_2$ and $RO_2$ radicals that determine the $P(O_3)_{net}$ values.

According to our previous study, a 50 % change in the physical loss lifetime results in only 3 %, 6 % and 10 % changes in OH concentration, $HO_2$ concentration and ozone production rate, respectively (Wang et al., 2021). We have added the discussion in the main text on page 21-22, lines 598-607:

"To achieve the best agreement between the modelled $O_3$ concentrations and the observed values, we applied different dilution factors (the lifetime of the species) in the modelling, varying from 6 h to 24 h. We found that the simulated $O_3$ is closest to the measured $O_3$ concentrations when the lifetime of the species is set to 12 h. The modelled $P(O_3)_{net}$ increases with the decrease of the dilution factor, but this doesn't affect the main conclusions as the influence of the dilution factor on the modelled $P(O_3)_{net}$ is negligible due to the very short lifetime of the $HO_2$ and $RO_2$ radicals that determine the $P(O_3)_{net}$ values (Wang et al., 2021). Therefore, a constant dilution factor of $1/43200 \text{ s}^{-1}$ was set throughout the observation period."

- L541-543: Some values are provided for NMB and NME without addressing what it means for the model performance. The authors should comment these values in the text.

We have addressed the meaning of NMB and NME in the revised modified manuscript on page 22, lines 614-617:

"These analysis results indicate that the model underestimates the measured $P(O_3)_{net}$ by a factor ranging from 1.42 (25th percentile) to 1.31 (75th percentile), calculated as (1+|NMB|), and the simulation results are reliable (with -1<NME<1)."

**Minor comments:**

- L68-71: The authors should provide a brief summary of what is known about ozone formation in the PRD region.

Ok, we have made a brief summary of what is known about ozone formation in the PRD area on page 3, lines 72-77:

"Currently, many scholars have analyzed the relationship between tropospheric ozone pollution and its precursors and meteorological elements in the PRD region, results show that the surface $O_3$ pollution is determined by both local photochemistry and physical transport, with long-range transport contributing 30%-70% to surface $O_3$ concentrations (Mao et al., 2022; Shen et al., 2021; Li et al., 2012, 2013)."

We have replaced the reference of (Li et al., 2022a) with (Mao et al., 2022; Shen et al., 2022; Li et al., 2012, 2013) to properly support our findings.

- L121-122: What is the air residence time in the sampling lines?

The residence time of the sample gas is inversely proportional to the flow rate and the tube length. In our study, the tube length ranged from 5 m (at 5 m height) to 400 m (at 335 m height). At a tube length of 400 m, the residence time is less than 180 s at a flow rate of 13 SLPM. We have added a corresponding description on page 5, lines 142-145:

"Consequently, the flow rates of the air sample streams for the six tubes varied between 12.0 and 15.0 SLPM without subsampling and were less than 20 SLPM with subsampling. The residence time of the sample gas in the longest tube (~ 400 m) is less than 180 s at a flow rate of 13 SLPM."

- L158-168: This section is not necessary here. Please just indicate that pulse experiments were performed to quantify the residence time in the chambers and cite the paper of Hao et al. (2023).

Ok, we deleted the very detailed method for quantifying residence time and changed the related description of it on page 6, lines 181-183:

"A schematic of the NPOPR detection system is shown in Fig. S2. The pulse experiments were performed to quantify the residence time in the chambers (Hao et al., 2023)."

- L208-210: Please indicate the frequency and duration of zero measurements.

Ok, we have added the frequency and duration of zero measurements on page 8, lines 229-230:

"The background signal of each mode was measured every 30 min for at last 2 min by automatically switching the ambient measurement to a custom-built platinum catalytic converter heated to 365 °C."

- L212: Please provide the E/N value.

Ok, we have provided the E/N value on page 8, lines 232-235 in the modified manuscript:

"Eventually, we only used VOCs measured during the $H_3O^+$ mode, which was operated at a drift tube pressure of 3.8 mbar, a temperature of 120 °C, and a voltage of 760 V, resulting in an $E/N$ ($E$ refers to the electric field and $N$ refers to the number density of the buffer gas in the drift tube) value of ~ 120 Td (townsend)."

- L233: "the photolysis of $NO_2$ at 424 nm" – please provide a range of wavelength instead of a unique wavelength.

Sorry for this mistake, thanks for pointing this out. The photolysis wavelength of $NO_2$ should be less than 420 nm. We have changed the description on page 9, line 274-276 in the revised manuscript:

"The specific tropospheric $O_3$ photochemical formation process involves the photolysis of $NO_2$ at < 420 nm (Sadanaga et al., 2017)."

- L550: Please remove "and in Indiana in the United States (~ 30 ppbv h-1 in spring) (Sklaveniti et al., 2018)". Sklaveniti et al. did not measure ambient P(O3) but investigated the sensitivity of P(O3) to NO additions in the instrument.

We apologize for this mistake. We have checked the paper again and removed this sentence accordingly.

- L570: Please rephrase. It's not clear what is meant by "underestimate the NOx limited regime"

We apologize for the confusing description. We meant that the underestimation of the modelled $P(O_3)_{net}$ due to the unknown mechanisms could lead to the NOx-limited regime being shifted to the VOCs-limited regime, as illustrated in Wang et al. (2022, 2024), thus underestimating the NOx regime. We have changed the description on pages 23, lines 639-647 in the revised manuscript.

"For example, the reaction of OH with unknown VOCs (Tan et al., 2017), the lack of correction for the decomposition of $CH_3O_2NO_2$, the missing $RO_2$ production from photolysis $ClNO_2$ (Whalley et al., 2018; Tan et al., 2017), and the underestimation of OVOCs photolysis (Wang et al., 2022) in modelling approaches may lead to the underestimation of $RO_2$, thus underestimating the modelled $P(O_3)_{net}$. Further analysis showed that the underestimation of $P(O_3)_{net}$ can lead to the NOx-limited regime being

shifted to the VOCs-limited regime, thus underestimating the NOx-limited regime (Wang et al., 2022, 2024)."

- L793-794: "The maximum estimated error of modelled P(O3)net ranged from 22-45 % during different episodes and non-episodes.". This has not been discussed in the manuscript before the general conclusion. The authors should discuss this point in more details the manuscript. How is the 22-45% error estimated?

We apologize for the lack of discussion of this in the manuscript before the general conclusion. The 22-45% error was estimated as the median of the modelled $P(O_3)_{net}$ bias, i.e., the median value of [measured $P(O_3)_{net}$-modelled $P(O_3)_{net}$]/measured $P(O_3)_{net}$ during different episodes and non-episodes. We have added the related discussion in the main text on page 23, lines 632-634:

"The median value of [measured $P(O_3)_{net}$-modelled $P(O_3)_{net}$]/measured $P(O_3)_{net}$ ranged from 22% to 45% for different episodes and non-episodes."

And changed the related description on page 31, lines 875-877:

"The median value of the estimated error of the modelled $P(O_3)_{net}$ ranged from 22-45 % during different episodes and non-episodes."

- L32: "photochemical reactions play a dominate role" should read "photochemical reactions playing a major role"

We have changed this description accordingly on page 1-2, lines 30-32:

"The identified $O_3$ pollution episodes were found to be jointly influenced by both photochemical production and physical transport, with local photochemical reactions playing a major role."

- L56: "Tropospheric ozone (O3), which have adverse effects on ecosystems" should read "Tropospheric ozone (O3), which has adverse effects on ecosystems"

We have modified the description on page 2, line 56:

"Tropospheric ozone ($O_3$), which has adverse effects on ecosystems, climate change, and human health (Fiore et al., 2009; Anenberg Susan et al., 2012; Seinfeld, 2016),…"

- L58: "important factor resulting severe regional air pollution" should read "important factor resulting in severe regional air pollution"

We apologize for this mistake. We have modified this sentence on page 2, lines 58:

"Tropospheric ozone ($O_3$), which has adverse effects on ecosystems, climate change, and human health (Fiore et al., 2009; Anenberg Susan et al., 2012; Seinfeld, 2016), have become an important factor resulting in severe regional air pollution in China (Zhu et al., 2020)."

- L59: "Tropospheric O3 mainly comes from the external transport from the stratosphere" should read "Tropospheric O3 mainly comes from stratospheric intrusions"

We have modified the description on page 2, lines 59:

"Tropospheric $O_3$ mainly comes from stratospheric intrusions and the photochemical reactions of $O_3$ precursors, involving volatile organic compounds (VOCs) and nitrogen oxides ($NOx=NO+NO_2$),"

- L87-88: "thus largely hindered our in depth understanding of" should read "thus largely hindering our understanding of"

We have changed the description on page 3-4, lines 96-100:

"Owing to the low time resolution of these monitoring techniques, achieving continuous vertical coverage of VOCs and NOx measurements is challenging. Consequently, the vertical distribution structure of VOCs remains unclear, thus largely hindering our understanding of the vertical and temporal regional ozone formation mechanism."

- L90: "observation system based on the Shenzhen Meteorological Gradient Tower" should read "observation system located on the Shenzhen Meteorological Gradient Tower"

We actually meant that the vertical observation system was built based on the basis of the Shenzhen Meteorological Gradient Tower (SZMGT). We have changed this sentence on page 4, lines 101-103:

"To fill the gaps in the existing studies, we utilized a newly constructed vertical observation system based on the Shenzhen Meteorological Gradient Tower (SZMGT) (Li et al., 2023)."

- L92: "To diagnose the P(O3)net and O3 formation" should read "To diagnose the net ozone production rate, $P$(O3)net, and O3 formation"

We have changed this sentence on page 4, lines 104:

"To diagnose the net ozone production rate, $P(O_3)_{net}$, and $O_3$ formation sensitivities across various heights,"

- L94: "with the Master Chemical Mechanism (MCM v3.3.1)." should read "with the Master Chemical Mechanism (MCM v3.3.1), referred to as OBM-MCM in the following."

We have changed this description on page 4, lines 105-107:

"we employed an observation-based model coupled with the Master Chemical Mechanism (MCM v3.3.1), referred to as OBM-MCM in the following."

- L104: "while acknowledging potential biases associated modelling." Should read "while acknowledging potential biases associated to the modelling.

We have changed this sentence on page 4, lines 114-116:

"we have extensively discussed the vertical and temporal variability in $P(O_3)_{net}$ and $O_3$ formation sensitivity, while acknowledging potential biases associated to the modelling."

- L172: "[O3] represents the difference should read "d[O3] represents the difference"

We have changed the description on page 6, lines 190:

"$d$[O₃] represents the difference between the $O_3$ mixing ratios at the inlet and outlet of the reaction and reference chambers"

- L 173 & L184: "(Hao et al., 2013) » should read "(Hao et al., 2023)."

We apologize for this mistake. We have changed it to " (Hao et al., 2023)."

- L198: "mass spectrometry » should read « mass spectrometer"

We have changed it on page 7, lines 219:

"VOCs were measured using a high-resolution proton transfer reaction time-of-flight mass spectrometer (PTR-TOF-MS, Ionicon Analytik, Austria) (Wang et al., 2020a; Wu et al., 2020) and an off-line gas chromatography mass spectrometry flame ionization detector (GC-MS-FID) (Wuhan Tianlong, Co. Ltd, China) (Yuan et al., 2012)."

- L241-247: Please rephrase. This sentence does not have a conjugated verb.

We apologize for this mistake. We rephrased this sentence accordingly on page 9, lines 284-285:

"$P(O_3)_{net}$ and $O_3$ concentrations were simulated by constraining $T$, RH, $P$, organic and inorganic substances in gases, including…"

- L268: "$P(O3)net$ denotes the net photochemical O3 production rate (ppbv h-1)" should read "$P(Ox)net$ denotes the net photochemical O3 production rate (ppbv h-1)"

We have changed this sentence on page 11, line 319:

"Where $\frac{dO_x}{dt}$ is the change rate of the observed $O_x$ mixing ratio change (ppbv h$^{-1}$), $P(Ox)_{net}$ denotes the net photochemical $O_3$ production rate (ppbv h$^{-1}$), which was equal to $P(O_3)_{net}$ and measured directly by the NPOPR system, $R(O_X)_{trans}$ represents $O_3$ mixing ratio change due to the physical transport (ppbv h$^{-1}$), including the horizontal and vertical transport, dry deposition and the atmospheric mixing (Liu et al., 2022)."

- L297: "different kinds of VOCs groups together to investigated their influence to the gradient $P(O3)net$ change with heights in Sect. 3.2.3." should read "different kinds of VOC groups together to investigate their influence on the vertical gradient of $P(O3)net$ in Sect. 3.2.3."

We have changed the description on page 11, lines 347-349:

"In this study, we summarized the OH reactivities of different kinds of VOCs groups together to investigate their influence on the vertical gradient $P(O_3)_{net}$ in Sect. 3.2.3."

- L454: "As concluded in Sect. 3.3.1" should read "As concluded in Sect. 3.1.1"

We have changed the description on page 18, lines 508:

"As concluded in Sect. 3.1.1 $O_3$ pollution episodes may be jointly affected by the photochemical reactions and physical transport."

- L458: "As the dry deposition are usually contribute" should read "As dry deposition usually contributes"

We have changed the sentence on page 17, lines 512-514:

"Typically, as dry deposition contributes a relatively small portion and can often be considered negligible, making vertical and horizontal transport the main contributors to physical processes (Tan et al., 2019)."

- Legend Figure 5: "R(O3)net" should read "R(O3)tran"

We have changed the legend of "$R(O_3)_{net}$" to "$R(O_3)_{trans}$" in Fig. 5.

- L504: "concentration became stable, suggests that the photochemical reaction competed against physical transport and jointly affect O3 concentration change" should read "concentration became stable, suggesting that the photochemical reaction competed against physical transport and jointly affected O3 concentration change"

We have changed this sentence on page 21, lines 570:

"Around noon, $O_3$ concentrations stabilize, suggesting a balance between photochemical reactions and physical transport affecting $O_3$ concentration changes."

- L 506: "the O3 concentration decreases due to the diffuse of photochemically formed O3" should read "the O3 concentration decreases due to the transport of photochemically formed O3"

We modified the description on page 21, lines 571-573:

"$O_3$ concentration decreases due to the transport of photochemically formed $O_3$ from the observation site to upward directions or the surrounding areas."

- L514: "with O3 diffuse to »" should read "with O3 transport to"

We modified the description on page 21, lines 580:

"②elevated photochemical $O_3$ production, with $O_3$ transport to surrounding areas under favorable diffusion conditions (i.e., non-episodes II)."

- L558: "presence of missing RO2 under high NO conditions" should read "underestimation of RO2 under high NO conditions"

We modified this sentence on page 22-23, lines 628-632:

"The measured $P(O_3)_{net}$ were mostly higher than the modelled $P(O_3)_{net}$, which could be attributed to the underestimation of $RO_2$ under high NO conditions, leading to substantial disparities between calculated $P(O_3)_{net}$ derived from measured and modelled $RO_2$ concentrations, as highlighted in previous studies (Whalley et al., 2018, 2021; Tan et al., 2017, 2018)."

- L569-570: "OVOCs photolysis (Wang et al., 2022) in modelling approach, may result in the underestimation of RO2, thus underestimate the modelled P(O3)net" should read "OVOCs photolysis (Wang et al., 2022) in modelling approaches, may result in the underestimation of RO2, thus underestimating the modelled P(O3)net"

We modified the sentence on page 23, lines 639-644:

"For example, the reaction of OH with unknown VOCs (Tan et al., 2017), the lack of correction for the decomposition of $CH_3O_2NO_2$, the missing $RO_2$ production from photolysis $ClNO_2$ (Whalley et al., 2018; Tan et al., 2017), and the underestimation of OVOCs photolysis (Wang et al., 2022) in modelling approaches may lead to the underestimation of $RO_2$, thus underestimating the modelled $P(O_3)_{net}$."

- L700: "heights, indicates the similar photochemical O3 formation regime" should read "heights, indicating a similar photochemical O3 formation regime"

We modified the sentence on page 27, lines 769-771:

"As illustrated in Fig. 9, the RIR values for different $O_3$-precursors or precursor groups don't exhibit significant variation at different heights during specific episodes or non-episodes, indicating a similar photochemical $O_3$ formation regime."

- L708: "during polluted episode I, both reduce VOCs and NOx" should read "during polluted episode I, reducing both VOCs and NOx"

We modified the sentence on page 27-28, lines 778-779:

"For example, during polluted episode I, reducing both VOCs and NOx can mitigate photochemical $O_3$ formation, but during the other $O_3$ polluted episodes and non-episodes, reduce VOCs can effectively alleviate photochemical $O_3$ formation, while the reduction of NOx might aggravate photochemical $O_3$ formation."

- L724: "which located in the » should read "which is located in the"

We modified the description on page 28-29, lines 807-809:

"This suggests a transition in the photochemical $O_3$ formation regime throughout the day, shifting from a VOC-limited regime in the morning to a transition regime and more sensitive to NOx in the afternoon around 16:00 LT."

- L731: "ROx radicals cycle reactions involved Nox" should read "ROx radicals cycle reactions involving NOx"

We modified the description on page 29, lines 814:

"Through the sensitivity study, NOx is not found to be the limiting factor affecting $P(O_3)_{net}$, therefore, reactions involving NOx in the ROx radicals cycle, such as $RO_2+NO \rightarrow HO_2$ and $HO_2+NO \rightarrow OH$, should occurred efficiently."

- L754: "Given that NOx has a significant titration effect on ozone" should read "Given that NO has a significant titration effect on ozone"

We modified the sentence on page 29, lines 837:

"Given that NO has a significant titration effect on ozone, the lower $O_3$ concentration at ground level may be attributed to the increase in NOx concentration due to a more pronounced NO titration effect, besides the dry deposition near the ground."

- L766: "with local photochemical reactions play a dominate role" should read "with local photochemical reactions playing a key role"

We modified the sentence accordingly on page 30, lines 849-850:

"we found that the O$_3$ pollution episodes were jointly influenced by both photochemical production and physical transport, ==with local photochemical reactions playing a key role==."

- L771: "the measurement period, indicated the" should read "the measurement period, indicating the"

We modified it accordingly on page 30, lines 854-855:

"The index of agreement (IOA) ranged from 0.87 (25$^{th}$ percentile) to 0.90 (75$^{th}$ percentile) for the measured and modelled $P(O_3)_{net}$ across ==the measurement period, indicating the== rationality to investigate the vertical and temporal variability of O$_3$ formation mechanism using modelling results."

- L774: "differences of measured and modelled $P(O3)net$" should read "differences between measured and modelled $P(O3)net$"

We modified the sentence on page 30, lines 856-858:

"However, the measured $P(O_3)_{net}$ generally exceeded the modelled $P(O_3)_{net}$, ==the differences between measured and modelled $P(O_3)_{net}$== ($\Delta P(O_3)_{net}$) were found to be correlated with NO concentrations."

References:

Hao, Y., Zhou, J., Zhou, J. P., Wang, Y., Yang, S., Huangfu, Y., Li, X. B., Zhang, C., Liu, A., Wu, Y., Zhou, Y., Yang, S., Peng, Y., Qi, J., He, X., Song, X., Chen, Y., Yuan, B., and Shao, M.: Measuring and modeling investigation of the net photochemical ozone production rate via an improved dual-channel reaction chamber technique, Atmos. Chem. Phys., 23, 9891-9910, 10.5194/acp-23-9891-2023, 2023.

Wang, W., Yuan, B., Peng, Y., Su, H., Cheng, Y., Yang, S., Wu, C., Qi, J., Bao, F., Huangfu, Y., Wang, C., Ye, C., Wang, Z., Wang, B., Wang, X., Song, W., Hu, W., Cheng, P., Zhu, M., Zheng, J., and Shao, M.: Direct observations indicate photodegradable oxygenated volatile organic compounds (OVOCs) as larger contributors to radicals and ozone production in the atmosphere, Atmos. Chem. Phys., 22, 4117-4128, 10.5194/acp-22-4117-2022, 2022.

Wang, W.; Yuan, B.; Su, H.; Cheng, Y.; Qi, J.; Wang, S.; Song, W.; Wang, X.; Xue, C.; Ma, C.; Bao, F.; Wang, H.; Lou, S.; Shao, M.: A large role of missing volatile organic compound reactivity from anthropogenic emissions in ozone pollution regulation, Atmos. Chem. Phys., 24, (7), 4017-4027,10.5194/acp-24-

4017-2024, 2024.

Zhou, J.; Wang, W.; Wu, Y.; Zhang, C.; Liu, A.; Hao, Y.; Li, X.-B.; Shao, M.: Development and application of a nitrogen oxides analyzer based on the cavity attenuated phase shift technique, J. Environ. Sci., 150, 692-703, 10.1016/j.jes.2023.11.017, 2025.

---

## Referee Report (RR1)

**Review for: Measurement report: Vertical and temporal variability of near-surface ozone production rate and sensitivity in an urban area in Pearl River Delta (PRD) region, China.**

In this manuscript, the authors have presented a detailed study on ozone production rates in an urban area in the Pearl River Delta region of China, including an investigation into the vertical and temporal variability of ozone, it's production rate and its precursors. The subject area is in the scope of this journal and would be of interest to the urban air quality community. I would recommend this manuscript for publication, provided the following comments have sufficiently been addressed.

**Overall comments:**

I am not completely convinced that comparing measured and modelled $O_3$ mixing ratios is a good way to validate the model, or to decide on any dilution factors (e.g Line 601). Measured $O_3$ mixing ratios will be influenced by transport, whereas modelled $O_3$ will not so you cannot draw a direct comparison here. In my opinion, the authors should address the caveats with using this to method to derive your dilution rates in the text more clearly or use a different compound which is formed from secondary chemistry, such as glyoxal, to determine their dilution rates / model lifetimes. Alternatively, they could use a value quoted in the literature and discuss the caveats of this instead. However, comparing measured $PO_3$ vs modelled $PO_3$ seems reasonable.

The authors need to be much clearer why understanding the vertical distributions are important to the air quality community. It seems that daytime $O_3$ is well mixed, and so ground level $O_3$ measurements would be representative of the vertical column. Is the key message that although this is the case, the VOC profile is different at different heights, meaning that if the chemical box modelling community is constraining to ground-level concentrations, they may not be accurately representing in situ $O_3$ production in the vertical column? If this is the case, it would be a very interesting conclusion and should be outlined in the text.

**Minor comments:**

Line 44: Type, "either" included twice. Remove one instance.

Line 190: Do you mean Table S3? At first, I went to figure S3 but I think you meant table S3. Please clarify in the text.

Line 203: Same as above comment – do you mean Table S4?

Line 232: Please clarify why you only used VOCs measured during the $H_3O^+$ mode.

Line 288: I wouldn't call these "conventional" pollutants. Please rephrase. Perhaps "inorganic pollutants"?

Line 295: Please clarify how you have decided on these background concentrations. Can you reference anywhere else this has been used?

Line 377: Now referred to as "ozone" in the text. It's better to ensure that either "$O_3$" or "ozone" is used consistently throughout the text.

Line 400: In Table 1, episode II $O_3$ mixing ratios are lower than non-episode II $O_3$ mixing ratios. Please explain why this is the case in the text.

Line 784: I'm confused on which VOCs fall into which categories. I looked at Table S2 (although if that is where the reader should be looking, please direct them here in the text), and many of the VOCs fall under two categories (e.g. AVOC and NMHC). Does that mean some species are repeated in different categories in Figure 9?

Line 820: It would be nice to know which specific AVOCs/OVOCs might be key, so that potential sources that can be targeted for reduction could be identified. The conclusion to this section is fine, but very general and doesn't really add any new details.

Line 841: "The TVOC and OFP exhibited variable trends with increased height during both daytime and nighttime" – what are the implications of this to the modelling or measurement community?

Line 847: I don't think you performed a test for statistical significance in this part, so perhaps rephrase.

Line 899: Could you expand a bit more in your conclusions on why an in-depth understanding of vertical variability of $O_3$ formation mechanisms is important? What could this new knowledge mean for the air quality community?

---

## Author Response (AR2)

We appreciate the reviewers' insightful comments and constructive suggestions, which will help us to provide a more accurate description of our work. Our responses are detailed below, presented in red, following the reviewers' comments that are in black. The revisions made to the manuscript are highlighted in yellow.

**Response to reviewer # 1:**

**Review for: Measurement report: Vertical and temporal variability of near-surface ozone production rate and sensitivity in an urban area in Pearl River Delta (PRD) region, China.**

In this manuscript, the authors have presented a detailed study on ozone production rates in an urban area in the Pearl River Delta region of China, including an investigation into the vertical and temporal variability of ozone, it's production rate and its precursors. The subject area is in the scope of this journal and would be of interest to the urban air quality community. I would recommend this manuscript for publication, provided the following comments have sufficiently been addressed.

We are honored by your positive evaluation of our manuscript. Thanks for your recognition of this study's potential contribution to the urban air quality community. We have taken your feedback with the utmost seriousness and have revised the manuscript to ensure that all points are considered.

**Overall comments:**

I am not completely convinced that comparing measured and modelled $O_3$ mixing ratios is a good way to validate the model, or to decide on any dilution factors (e.g Line 601). Measured $O_3$ mixing ratios will be influenced by transport, whereas modelled $O_3$ will not so you cannot draw a direct comparison here. In my opinion, the authors should address the caveats with using this to method to derive your dilution rates in the text more clearly or use a different compound which is formed from secondary chemistry, such as glyoxal, to determine their dilution rates / model lifetimes. Alternatively, they could use a value quoted in the literature and discuss the caveats of this instead. However, comparing measured $PO_3$ vs modelled $PO_3$ seems reasonable.

Thank you for the insightful suggestions. Unfortunately, we haven't measure glyoxal during the observation campaign. The dilution factors may only partially

reflect the impact of physical transport on $O_3$ mixing ratios, particularly the outflow of $O_3$ from the observation site due to physical processes. We have added the caveats regarding the use of the comparison between measured and modelled $O_3$ to derive the dilution rates in lines 596-598:

"Previous studies have utilized the comparison of measured and modelled $O_3$ concentrations to determine the dilution factor in modelling studies, discovering that suitable dilution factors vary by location (Yang et al., 2021)."

and lines 602-605 in the modified manuscript:

"However, given that $O_3$ concentrations are affected by physical transport processes, the dilution factor might only represent the outflow of $O_3$ from the observation site. Therefore, there may be limitations in using this method for precise comparisons."

To make the description more accurate, we have also changed the sentence in lines 292-294 in the original text: "The effect of physical processes (such as vertical and horizontal transport) was considered by setting a constant dilution factor of 1/43200 $s^{-1}$ throughout the modelling period." to "To avoid the build-up of long-lived species to unreasonable levels, we also considered the physical dilution process by setting a constant dilution factor of 1/43200 $s^{-1}$ throughout the modelling period (Liu et al., 2021; Decker et al., 2019)." in lines 295-298 in the modified manuscript.

Furthermore, we conducted a comparison between the measured and modelled $P(O_X)_{net}$ under different dilution rates, the results show that the modelled $P(O_3)_{net}$ first increases and then decreases as the dilution factor decreases, which corresponds to an extension of the species' lifetime. Notably, the best agreement between the modelled and modelled $P(O_3)_{net}$ was achieved when the species lifetime was set to 12 h. This discussion has been integrated into the revised manuscript in lines 605-613:

"We further compared the measured and modelled $P(O_3)_{net}$ under different dilution factors. The modelled $P(O_3)_{net}$ initially increases and then decreases as the dilution factor decreases (equivalent to an increase of species lifetime). However, the influence of varying dilution rates on the modelled $P(O_3)_{net}$ is minimal, constituting less than 30 %, due to the short lifetimes of the $HO_2$ and $RO_2$ radicals, which determine the $P(O_3)_{net}$ values (Wang et al., 2021). Notably, the modelled $P(O_3)_{net}$ closely matched the measured values when the species lifetime was set to 12 h, as illustrated in Fig. S3b.

Consequently, a constant dilution factor of $1/43200 \text{ s}^{-1}$ was applied throughout the observation period."

The authors need to be much clearer why understanding the vertical distributions are important to the air quality community. It seems that daytime $O_3$ is well mixed, and so ground level $O_3$ measurements would be representative of the vertical column. Is the key message that although this is the case, the VOC profile is different at different heights, meaning that if the chemical box modelling community is constraining to ground-level concentrations, they may not be accurately representing in situ $O_3$ production in the vertical column? If this is the case, it would be a very interesting conclusion and should be outlined in the text.

We concur with your opinion that the ozone ($O_3$) concentrations are well mixed in the boundary layer during daytime hours. However, the $O_3$ formation mechanisms exhibited significant vertical variability, driven by fluctuations in the concentrations of volatile organic compounds (VOCs) and nitrogen oxides (NOx). Chemical box modelling, when solely constrained by ground-level measurements, fails to accurately reproduce the $O_3$ production across the vertical column. Consequently, relying solely on ground-level $O_3$ formation mechanisms to devise control strategies for ozone precursors is inadequate. A more comprehensive approach is necessary to effectively address the complexities of $O_3$ production throughout the atmospheric column. We have outlined this conclusion in lines 488-496 in Sect. 3.1.2 in the modified manuscript:

"In conclusion, our daytime observations revealed minimal vertical gradients in the concentrations of $O_3$, NOx, Ox, and TVOC, attributed to the rapid vertical mixing effects driven by surface heating effects (Tang et al., 2017). This suggests that ground-level $O_3$ concentrations would be representative of the entire vertical column. Nonetheless, the OFP varies for different VOCs profiles at various heights, and the vertical mixing effects facilitates the downward transport of $O_3$ photochemically formed from higher altitudes to the near-ground layer. Consequently, a box model constraining to ground-level NOx and VOCs concentrations may not accurately reflect the in situ $O_3$ production in the vertical atmospheric column."

as well as lines 878-887 in the conclusion section:

"The findings provide us in-depth understanding of near-ground vertical variability in $O_3$ formation mechanisms, which are influenced by the concentrations of VOCs and

NOx, and the distinct OFP associated with different VOCs profiles. During daytime, the vertical mixing of air masses is substantially enhanced due to the effect of surface heating. Consequently, photochemically formed $O_3$ at higher altitudes can be vertically transported downward to the near-ground layer. Under this condition, control strategies for $O_3$ precursors based on the $O_3$ formation mechanisms on the ground-level are insufficient. A more comprehensive approach is necessary to effectively address the complexities of $O_3$ production throughout the atmospheric column. The vertical variability of $O_3$ formation mechanisms should be taken into account when making effective $O_3$ control strategies in the PRD area of China."

**Minor comments:**

Line 44: Type, "either" included twice. Remove one instance.

Ok, we removed one "either" in the modified manuscript.

Line 190: Do you mean Table S3? At first, I went to figure S3 but I think you meant table S3. Please clarify in the text.

We actually meant Sect. S3 in the supplementary materials: "**S3. The experiments concerning the light-enhanced loss of $O_3$**" We added Sect. S3 in the main text.

Line 203: Same as above comment – do you mean Table S4?

We actually meant Sect. S4 in the supplementary materials: "**S4. Measurement error of $P(O_3)_{net}$ and the LOD of the NPOPR detection system**" We added Sect. S4 in the main text.

Line 232: Please clarify why you only used VOCs measured during the $H_3O^+$ mode.

The PTR-ToF-MS instrument, when operated at NO+ mode, primarily detects higher alkanes, which significantly contribute to the formation of secondary organic aerosols (SOA) but negligible contributions to photochemical $O_3$ formation (Wang et al., 2020). Therefore, only the VOCs species identified during the PTR-ToF-MS measurements in $H_3O^+$ mode were used in this study. We have added this explanation in the main text in lines 232-234:

"Operating the PTR-ToF-MS instrument in $NO^+$ mode primarily detects higher alkanes, which are known significantly contribute to the formation of secondary organic aerosols (SOA) but negligible contributions to photochemical $O_3$ formation (Wang et al., 2020)."

Line 288: I wouldn't call these "conventional" pollutants. Please rephrase. Perhaps "inorganic pollutants"?

We agree with the reviewer. we changed it to "inorganic gaseous pollutants".

Line 295: Please clarify how you have decided on these background concentrations. Can you reference anywhere else this has been used?

We have set the background concentrations of $O_3$, CO, and $CH_4$ according to the findings of Wang et al. (2011), Wang et al. (2022a), and WMO greenhouse gas bulletin (2022), respectively. We have cited these references accordingly in the main text (lines 300-301):

"Additionally, the dry deposition rate of $O_3$ was set to 0.42 cm s$^{-1}$, and the background of $O_3$, CO, and $CH_4$ were set to 30, 70, and 1800 ppbv, respectively, based on the findings of Wang et al. (2011), Wang et al. (2022a), and WMO greenhouse gas bulletin (2022)."

Line 377: Now referred to as "ozone" in the text. It's better to ensure that either "$O_3$" or "ozone" is used consistently throughout the text.

We have changed "ozone" to "$O_3$" throughout the manuscript, after defining it on line 55: "Tropospheric ozone ($O_3$), which has adverse effects on ecosystems, climate change, and human health…"

Line 400: In Table 1, episode II $O_3$ mixing ratios are lower than non-episode II $O_3$ mixing ratios. Please explain why this is the case in the text.

Through the average $O_3$ mixing ratio during episode II was lower than during non-episode II, the variability of these average values, as indicated by the standard deviation, is actually higher during episode II. This means that even though there are days with very high hourly average $O_3$ concentrations-which define $O_3$ pollution episodes, where levels exceed the Grade II standard of 102 ppbv-the overall average $O_3$ concentrations for episode II is not higher than that of non-episode II, it shows greater fluctuations, as suggested by the larger standard deviation. We have added the explanation in lines 413-419 in the revised manuscript:

"For example, through there are days with very high hourly average $O_3$ concentrations which define $O_3$ pollution episodes-where levels exceed the Grade II standard of 102 ppbv-the overall average $O_3$ concentrations for episode II is not higher than that of non-episode II. This suggests that despite the occurrence of peak hourly levels, the average concentration for episode II remains lower, highlighting the fluctuating pattern of $O_3$ levels during these episodes."

Line 784: I'm confused on which VOCs fall into which categories. I looked at Table S2 (although if that is where the reader should be looking, please direct them here in the text), and many of the VOCs fall under two categories (e.g. AVOC and

NMHC). Does that mean some species are repeated in different categories in Figure 9?

Yes, we categorized the VOCs according to Table S2, and many of the VOCs fall under two categories. Therefore, the VOCs species in different categories in Figure 9 might be repeated. We have added this explanation on lines 745-747 in the modified manuscript:

"The VOCs species, categorized into different precursor groups as listed in Table S2, indicate that some species depicted in Figure 9 may appear in multiple categories and hence could be repeated."

Line 820: It would be nice to know which specific AVOCs/OVOCs might be key, so that potential sources that can be targeted for reduction could be identified. The conclusion to this section is fine, but very general and doesn't really add any new details.

Thanks for your suggestion. However, in our study, we have lumped the individual AVOCs/OVOCs species together to assess their relative incremental reactivity (RIR). This approach does not allow us to analyze which specific AVOCs or OVOCs are the most critical in terms of RIR. Alternatively, we have identified and presented the three VOC species with the highest OFP in Table S4, distinguishing between episodes and non-episodes. We acknowledged and discussed the significance of pointing these key species for targeted reduction strategies in the revised manuscript in lines 786-797:

"Given that photochemical $O_3$ formation is most sensitive to AVOC and OVOC groups, we further identified and presented the three VOC species with the highest OFP during different episodes and non-episodes in Table S4. Results show that compounds such as toluene, *m/p*-xylene, and n-butane in AVOC group, formaldehyde, hydroxyacetone, and ethanol in OVOC group have identified as the most significant contributors to the total OFP in all episodes and non-episodes. Toluene, *m/p*-xylene, and n-butane are often associated with specific industrial processes (Shi et al., 2022; Liang et al., 2017), while formaldehyde, hydroxyacetone, and acetaldehyde can originate from both the industrial processes and natural sources (Parrish et al., 2012; Fan et al., 2021; Spaulding et al., 2003; Salthammer 2023). Priority of these emission sources should be given to reducing AVOC and OVOC to mitigate $O_3$ pollution in the PRD area of China."

Line 841: "The TVOC and OFP exhibited variable trends with increased height during both daytime and nighttime" – what are the implications of this to the modelling or measurement community?

As mentioned above, we have added more discussion concerning the variable OFP for different VOCs profiles at various heights in lines 492-496 in Sect. 3.1.2 in the modified manuscript:

"Nonetheless, the OFP varies for different VOCs profiles at various heights, and the vertical mixing effects facilitates the downward transport of $O_3$ photochemically formed from higher altitudes to the near-ground layer. Consequently, a box model constraining to ground-level NOx and VOCs concentrations may not accurately reflect the in situ $O_3$ production in the vertical atmospheric column."

And further added the implications to the modelling or measurement community in lines 820-821 in the revised manuscript:

"However, the TVOC and their OFP exhibited variable trends with increased height during both daytime and nighttime, observed in episodes and non-episodes, which indicates the complexities of $O_3$ formation mechanisms at different heights throughout the atmospheric column."

Line 847: I don't think you performed a test for statistical significance in this part, so perhaps rephrase.

Sorry for the inaccurate description. We meant the mean concentrations of $O_3$ precursors, including CO, NO, $NO_2$, and TVOC, were not necessarily higher during episodes than those during non-episodes. We have changed this sentence to "The mean concentrations of $O_3$ precursors, including CO, NO, $NO_2$, and TVOC, were not consistently elevated during episodes compared to their levels during non-episodes."

Line 899: Could you expand a bit more in your conclusions on why an in-depth understanding of vertical variability of $O_3$ formation mechanisms is important? What could this new knowledge mean for the air quality community?

During daytime, the vertical mixing of air masses is substantially enhanced due to the effect of surface heating. Consequently, photochemically formed $O_3$ at higher altitudes can be vertically transported downward to the near-ground layer. Under this condition, control strategies for $O_3$ precursors based on the ozone formation mechanisms on the ground-level are insufficient. The vertical variability of $O_3$

formation mechanisms should be taken into account when making effective $O_3$ control strategies. We added more discussion concerning the importance of in-depth understanding of vertical variability of $O_3$ formation mechanisms in lines 878-887 in the modified manuscript:

"The findings provide us in-depth understanding of near-ground vertical variability in $O_3$ formation mechanisms, which are influenced by the concentrations of VOCs and NOx, and the distinct OFP associated with different VOCs profiles. During daytime, the vertical mixing of air masses is substantially enhanced due to the effect of surface heating. Consequently, photochemically formed $O_3$ at higher altitudes can be vertically transported downward to the near-ground layer. Under this condition, control strategies for $O_3$ precursors based on the $O_3$ formation mechanisms on the ground-level are insufficient. A more comprehensive approach is necessary to effectively address the complexities of $O_3$ production throughout the atmospheric column. The vertical variability of $O_3$ formation mechanisms should be taken into account when making effective $O_3$ control strategies in the PRD area of China."

References:
Fan, J., Ju, T., Wang, Q., Gao, H., Huang, R., and Duan, J.: Spatiotemporal variations and potential sources of tropospheric formaldehyde over eastern China based on OMI satellite data, Atmos. Pollut. Res., 12, 272-285, 10.1016/j.apr.2020.09.011, 2021.

Liang, X., Chen, X., Zhang, J., Shi, T., Sun, X., Fan, L., Wang, L., and Ye, D.: Reactivity-based industrial volatile organic compounds emission inventory and its implications for ozone control strategies in China, Atmos. Environ., 162, 115-126, 10.1016/j.atmosenv.2017.04.036, 2017.

Parrish, D. D., Ryerson, T. B., Mellqvist, J., Johansson, J., Fried, A., Richter, D., Walega, J. G., Washenfelder, R. A., de Gouw, J. A., Peischl, J., Aikin, K. C., McKeen, S. A., Frost, G. J., Fehsenfeld, F. C., and Herndon, S. C.: Primary and secondary sources of formaldehyde in urban atmospheres: Houston Texas region, Atmos. Chem. Phys., 12, 3273-3288, 10.5194/acp-12-3273-2012, 2012.

Salthammer, T.: Acetaldehyde in the indoor environment, Environ. Sci. Atmos., 3, 474-493, 10.1039/D2EA00146B, 2023.

Shi, J., Bao, Y., Ren, L., Chen, Y., Bai, Z., and Han, X.: Mass concentration, source and health risk assessment of volatile organic compounds in nine cities of Northeast China, Int. J. Environ. Res. Public Health, 19, 4915, 10.3390/ijerph19084915, 2022.

Spaulding, R. S., Schade, G. W., Goldstein, A. H., and Charles, M. J.: Characterization of secondary atmospheric photooxidation products: Evidence for biogenic and anthropogenic sources, J. Geophys. Res.,108, 4247, 10.1029/2002JD002478, 2003.

Wang, C., Yuan, B., Wu, C., Wang, S., Qi, J., Wang, B., Wang, Z., Hu, W., Chen, W., Ye, C., Wang, W., Sun, Y., Wang, C., Huang, S., Song, W., Wang, X., Yang, S., Zhang, S., Xu, W., Ma, N., Zhang, Z., Jiang, B., Su, H., Cheng, Y., Wang, X., and Shao, M.: Measurements of higher alkanes using NO$^+$ chemical ionization in PTR-ToF-MS: important contributions of higher alkanes to secondary organic aerosols in China, Atmos. Chem. Phys., 20, 14123-14138, 10.5194/acp-20-14123-2020, 2020.

**Response to reviewer # 2:**

The authors have provided additional information that answers most of the comments from my initial review (reviewer #2, report #3). However, there are still a few points that need to be addressed before publication.

We would like to express our sincere gratitude for your thorough initial review (reviewer #2, report #3) and for the time you have invested in evaluating our manuscript. We have taken your comments seriously and have made substantial revisions. We believe that these revisions have significantly improved the quality of the manuscript.

Major comments:

1/ Supplement S3

- The authors should clarify whether the "outdoor experiment" was carried out by flowing zero air or ambient air in the reaction and reference chambers. From the text it seems that ambient air was used.

Yes, the "outdoor experiment" was carried out by flowing zero air in the reaction and reference chambers, but the reaction and reference chambers are located outdoor. We modified the description in S3 to make it clearer:

"***The light-enhanced loss of O₃ in the reaction and reference chambers*** at 5 L min$^{-1}$ (the flow rate used during the observation campaign in this study) was investigated by carrying out the following outdoor experiment: the O$_3$ with a mixing ratio of approximately 130 ppbv generated by the O$_3$ generator (P/N 97-0067-02, Analytic Jena US, USA) was injected into both the reaction and reference chambers. We flowed zero air together with the generated O$_3$ into these chambers, which are located outdoors, to

ensure there was no photochemical $O_3$ production. This setup allowed us to observe the real changes in photolysis frequencies of different species during daytime."

- The authors indicate that "O3 was injected at a mixing ratio of approximately 130 ppbv …. to ensure that no photochemical O3 was produced during the outdoor experiment." – If ambient air was used during these experiments, how can the authors be sure that there is no ozone production in the chambers?

As described above, zero air is used for the tests of light-enhanced loss of $O_3$ in the reaction and reference chambers.

2/ Supplement S4

- Please clarify whether (Ox)error is an absolute or a relative error. From equation S1 it seems that this is a quadratic propagation of absolute errors.

Yes, the (Ox)error represents an absolute error, resulting from the quadratic propagation of individual absolute errors. To elucidate this concept better, we have included the following explanation in Supplement S4:

"where $(O_X)_{error}$ represents the absolute error in the estimated $O_X$ concentration in the reaction and reference chambers, which results from the quadratic propogation of the absolute errors $(O_{X_\gamma})_{error}$ and $(O_{X_{CAPS}})_{error}$. Here, $(O_{X_\gamma})_{error}$ denotes the error associated with the $\gamma$-corrected Ox of the chambers, while $(O_{X_{CAPS}})_{error}$ signifies the measurement error of the $O_X$ measured by the CAPS-NO2 monitor."

- The power function referenced as S2 would lead to a low error on measured Ox concentrations when [Ox]>20-30 ppb. However, it would lead to a large error for low Ox concentrations. For instance, plugging a Ox concentration of 1ppb in this equation would lead to an error of 9.7ppb, which does not seem reasonable. The authors should comment on this.

Yes, as indicated by the power function in equation S2, the error increases as the measured $O_X$ concentration decreases, resulting a more significant $(O_{X_{CAPS}})_{error}$ at lower $O_X$ levels. However, this power function has been derived from the calibration for $O_X$ concentrations ranging from 20 ppbv to 160 ppbv. Applying it outside this range, especially at very low $O_X$ concentrations, could lead to disproportionately large errors that do not reflect the true variability of the measurement errors. In this study, the $O_X$ concentrations varied between 28 ppbv and 145 ppbv, which falls into the calibration range, validating the use of this power function for estimating $(O_{X_{CAPS}})_{error}$ throughout the measurement period. We have added the related comment in Supplementary S4:

"We acknowledge that this power function has been derived from calibration data of the $O_X$ concentrations ranged from 20 ppbv to 160 ppbv. Utilizing this function outside this calibrated range, especially at very low $O_X$ concentrations, may result in errors that are disproportionately large and may not accurately capture the true variability of the measurement errors. In this study, the $O_X$ concentrations ranged from 28 to 145 ppbv, which falls into the calibration range. Consequently, this power function is deemed appropriate for estimating the $(O_{X_{CAPS}})_{error}$ throughout the whole measurement period."

- Equation S3 assumes no error associated to the residence time. Is it correct to do so? What is the uncertainty associated to the residence time?

We did not include the error associated with the residence time in Equation 3. In a previous study (Hao et al., 2023), we assessed the error in residence time and determined it to be approximately 0.0007, with an average residence time of 0.063 h at a flow rate of 5 L min$^{-1}$. Upon incorporating this residence time error into the calculation of '$P(O_3)_{net\_error}$' using the 'error in the quotient' principle from the 'error propagation rules', we observed that the '$P(O_3)_{net\_error}$' value decreased, with a reduction ranging from 0 to 2% [0.25-0.75 percentile]. This reduction is negligible compared to the '$P(O_3)_{net\_error}$' calculated without accounting for the residence time error. Consequently, we chose not to consider the uncertainty related to the residence time in our calculations. We have provided this explanation in Supplement **S4. Measurement error of $P(O_3)_{net}$ and the LOD of the NPOPR detection system**:

"In our previous research (Hao et al., 2023), we evaluated the residence time error and determined it to be approximately 0.0007, with an average residence time of 0.063 hours at a flow rate of 5 L min$^{-1}$. When we considered this error in the calculation of '$P(O_3)_{net\_error}$', we observed a minimal reduction in the '$P(O_3)_{net\_error}$' values, ranging from 0 to 2% [0.25-0.75 percentile]. This impact is considered negligible in relation to the overall '$P(O_3)_{net\_error}$' as presented in Eq. (3). Consequently, we did not consider the uncertainty associated with the residence time in our calculations."

3/ Main paper

L205-207: "The measurement accuracy of NPOPR detection system is determined as 13.9 %, which is the maximum systematic error caused by the photochemical O3 productions in the reference chamber." – I do not understand this statement since the authors indicate on L214 that the measurement bias introduced by ozone production in the reference chamber is corrected for. The P(O3) measurement accuracy should therefore depend on the error associated with this correction. In addition, the measurement accuracy should account for other sources of errors such as the error associated to the residence time in the reaction and reference chambers.

Sorry for the confusing description. We agree with the reviewer that the $P(O_3)_{net}$ corrections mentioned in lines 212-215 in the original manuscript, "Here, we employed the same modelling method described in Hao et al. (2013) to quantify the $P(O_3)_{net}$ in the reference chamber and corrected the bias caused by the $P(O_3)_{net}$ in reference chamber accordingly (more details can be found in Sect. 2.2.1)." are indeed related to the measurement accuracy of the NPOPR detection system as described in lines 206-208: "The measurement accuracy of the NPOPR detection system is determined as 13.9 %, representing the maximum systematic error resulting from photochemical $O_3$ production in the reference chamber." We refer to this as the "accuracy of the NPOPR detection system" because it accounts for the systematic errors inherent in the system. These errors arise from photochemical $O_3$ productions in the reference chamber, as a result of the UV protection Ultem film that only filters out the sunlight with wavelengths less than 390 nm. Consequently, photochemical $O_3$ production from sunlight wavelengths between 390 nm and 790 nm still exist in the reference chamber.

However, the error calculated in Eq. (S3) as described in Supplement **S4. Measurement error of $P(O_3)_{net}$ and the LOD of the NPOPR detection system** is considered as the measurement precision. This error refers to the degree of consistency or repeatability observed in a set of measurements by the NPOPR detection system. Here we have not taken in to account the uncertainty associated to the residence time, for the reasons outlined above.

We have added this explanation in lines 199- 216 in the main text:

"The limit of detection (LOD) of the NPOPR detection system is 2.3 ppbv h$^{-1}$ at the sampling air flow rate of 5 L min$^{-1}$, which is obtained as three times the measurement error of $P(O_3)_{net}$ (Hao et al., 2013). The measurement error of $P(O_3)_{net}$ is determined by the estimation error of Ox in the reaction and reference chambers, which includes the measurement error associated with the $O_X$ of the CAPS-NO$_2$ monitor and the error due to the light-enhanced loss of $O_3$. This collective measurement error is referred to as the measurement precision of the NPOPR detection system, with further details provided in the supplementary materials, specifically in Sect. S4. The measurement accuracy of the NPOPR detection system is determined as 13.9 %, representing the maximum systematic error resulting from photochemical $O_3$ production in the reference chamber. Our earlier research indicated that the modelled $P(O_3)_{net}$ in the reaction chamber is similar to that modelled in ambient air, with the modelled $P(O_3)_{net}$ in the reference chamber accounting for 0-13.9% of that in the reaction chamber (Hao et al., 2023). This is due to the UV protection Ultem film covered on the reference chamber, which only filtered out the sunlight with wavelengths < 390 nm, allowing photochemical $O_3$ production persist at the sunlight wavelength between 390 nm and 790 nm. Here,

we have utilized the same modelling approach described in Hao et al. (2013) to quantify the $P(O_3)_{net}$ in the reference chamber and corrected for the bias introduced by the measurement accuracy."

And supplement S4:

"We note that this collective measurement error of $P(O_3)_{net}$ is referred to as the measurement precision of the NPOPR detection system, which is different with the measurement accuracy of the NPOPR detection system described in Sect. 2.2.2."

Minor comments for the main paper:

L65-66: Shouldn't "while a VOCs-limited regime has lower VOCs/NOx ratios and the O3 formation is sensitive to NOx concentration changes" read "while a VOCs-limited regime has lower VOCs/NOx ratios and the O3 formation is sensitive to VOC concentration changes"?

Yes, it should be read 'while a "VOCs-limited" regime has lower VOCs/NOx ratios and the $O_3$ formation is sensitive to VOCs concentration changes.", we have corrected this sentence in the modified manuscript in lines 63-65.

L154: Please replace "self-developed" by "home-made"

Ok, we replaced "self-developed" by "home-made".

L184-185: It should be clearly stated that [Ox] concentrations plugged in Eq.1 are measured concentrations corrected from Ox losses in the reaction and reference chambers. Please replace "We further quantified and corrected the wall losses of Ox and the light-enhanced loss of O3 (d[O3]) in the reaction and reference chambers during daytime" by "[Ox] values plugged in Eq.1 to derive P(O3)net are measured values corrected for wall losses of Ox and the light-enhanced loss of O3 (d[O3]) in the reaction and reference chambers during daytime"

Thanks for pointing this out. We changed the sentence accordingly in lines 183-185 in the modified manuscript.

L410: "5.1E-4" should read "5.1×10-4". Please also correct other instances of wrong formatting in the rest of the paragraph.

Ok, we corrected all instances of wrong formatting in the rest of the paragraph.

Figures 4 & 5: Please homogenize the notation between the main text and these figures - d(O3) vs. d(Ox), R(O3) vs. R(Ox), P(O3) vs. P(Ox)

We homogenized the notation between the main text and Figs. 4 & 5 to $d(O_X)/dt$, $P(O_X)_{net}$, and $R(O_X)_{tran}$.

L877-878: "The median value of the estimated error of the modelled P(O3)net ranged from 22-45 % during different episodes and non-episodes." Should read "The median relative difference between measured and modelled P(O3)net ranged from 22-45 % during different episodes and non-episodes."

Thank you for the revision. We have corrected the sentence accordingly to make it clearer.

Minor comments for S3:

- "The light-enhanced loss coefficient of O3 ($\gamma$) was calculated using Eq. (4) described in the main text" should read "The light-enhanced loss coefficient of O3 ($\gamma$) was calculated using Eq. (2) described in the main text"

Sorry for this mistake. We corrected the sentence in S3 to "The light-enhanced loss coefficient of $O_3$ ($\gamma$) was calculated using Eq. (2) described in the main text".

Appendix:

We detected other errors during the manuscript review and revised them as follows:

In S3: we changed "The obtained $\gamma$-$J(O^1D)$ equation listed in Eq. (4) was used to correct for the light-enhanced loss of $O_3$ in the reaction and reference chambers during the daytime to exclude the influence of light-enhanced loss." to "The obtained $\gamma$-$J(O^1D)$ equation listed in Fig. S8a was used to correct for the light-enhanced loss of $O_3$ in the reaction and reference chambers during the daytime to exclude the influence of light-enhanced loss."

References:

Hao, Y., Zhou, J., Zhou, J. P., Wang, Y., Yang, S., Huangfu, Y., Li, X. B., Zhang, C., Liu, A., Wu, Y., Zhou, Y., Yang, S., Peng, Y., Qi, J., He, X., Song, X., Chen, Y., Yuan, B., and Shao, M.: Measuring and modeling investigation of the net photochemical ozone production rate via an improved dual-channel reaction chamber technique, Atmos. Chem. Phys., 23, 9891-9910, 10.5194/acp-23-9891-2023, 2023.

---

## Author Response (AR3)

Author's response to the editor's comments:

I have checked through your responses to the reviewers and the changes that you have made to the manuscript and I am happy that the remaining queries they raised have been adequately addressed. There are a few typos that I spotted whilst reading the manuscript listed below which need to be corrected. Once these changes have been made, I am happy for the manuscript to be published in ACP.

We extend our sincere gratitude to the editor's thorough review and valuable guidance provided throughout the review process, which have significantly contributed to the paper's quality. Our responses are listed below, presented in red, following the reviewers' comments, which are in black. The revisions made to the manuscript are highlighted in yellow.

Line 64: VOC-sensitive regime $O_3$ decreases with increasing NOx and increases with increasing VOC (so will impacted by both changes in NOx and VOC concentrations). Also change 'VOCs' to 'VOC'

Thanks for the correction. We have changed the sentence to "A "NOx-limited" regime has higher VOCs/NOx ratios and the $O_3$ formation is sensitive to NOx concentration changes, while a "VOCs-limited" regime has lower VOCs/NOx ratios and the $O_3$ formation decreases with increasing $NO_X$ and increases with increasing VOC."

Line 77: 'VOCs' to 'VOC'

Ok. We changed 'VOCs' to 'VOC'.

Line 213: add 'to' before 'persist'

Ok. We added 'to' before 'persist'.

Line230: 'last' to 'least'

Sure. We changed 'last' to 'least'.

Line 251: Change 'The concentrations of 56 NMHC species in the canister analyzed by GC-MS/FID were calibrated daily using the mixture of a photochemical assessment monitoring stations (PAMS) standard gas and pure N2' to 'The concentrations of 56 NMHC species in the canister were analyzed by GC-MS/FID which was calibrated daily using the mixture of a photochemical assessment monitoring stations (PAMS) standard gas and pure N2'

Ok. We changed the sentence to "The concentrations of 56 NMHC species in the

canister were analyzed by GC-MS/FID which was calibrated daily using the mixture of a photochemical assessment monitoring stations (PAMS) standard gas and pure $N_2$." in lines 252-253 in the modified manuscript.

Line 279: 'recylces' to 'cycles'

Ok. We changed 'recylces' to 'cycles'.

Line 301: 'modelling' to 'model'

Ok. We changed 'modelling' to 'model'.

Line 356: ' VOCs' to 'VOC'

Ok. We changed 'VOCs' to 'VOC'.

Line 413: 'through' to 'though'

Ok. We changed 'through' to 'though'.

Line 498: ' explored' to 'explore'

Ok. We changed 'explored' to 'explore'.

Line 505: 'discussed' to 'discuss'

Ok. We changed 'discussed' to 'discuss'.

Line 527: 'manus' to 'minus'

Ok. We changed 'manus' to 'minus'.

Line 534: 'error caused by light-enhanced loss of O3' this shouldn't contribute at night, so suggest removing this error as the possible cause of P(Ox)net

Sorry for the confusion description. We primarily meant this is included in the measurement uncertainty of $P(O_X)_{net}$. We have changed the sentence to "During nighttime, $P(O_X)_{net}$ should be zero without sun radiation, the significant $P(O_X)_{net}$ shown in Fig. 5 may be due to the measurement uncertainty of $P(O_X)_{net}$, which is determined by the measurement error of $O_X$ of CAPS-NO$_2$ monitor in the reaction and reference chambers (as discussed in Sect. S4)." in lines 535-537 in the modified manuscript.

Line 569: 'to upward directions' to 'upwards'

Sorry for the unappropriated description, we changed 'to upward directions' to 'upwards'.

Line 647: 'the lack of correction for the decomposition of CH3O2NO2' is a potential interference in RO2 measurements – it isn't a modelling problem, so I suggest removing this as a possible factor.

Indeed. We removed 'the lack of correction for the decomposition of CH₃O₂NO₂' as a potential interference' in modelling approaches in the modified manuscript.

Line 696: 'exhibiting' to 'exhibit'

Ok. We changed 'exhibiting' to 'exhibit' in the modified manuscript.

Line 756: remove 'that the'

Ok. We removed 'that the' in the modified manuscript.

Line 784: remove 'should'

Ok. We removed 'should' in the modified manuscript.

Line 838: 'Base' to 'Based'

Ok. We changed 'Base' to 'Based' in the modified manuscript.

Line 823: 'the OFP was primarily attributed to OVOCs..' but on line 871 ' most sensitive to AVOC..' need to be consistent.

Sorry for the confusing description. The description in line 823 'the OFP was primarily attributed to OVOCs.' was based on the OFP calculated from different VOCs categories, including OVOCs, aromatics, alkyne, alkene, and alkane as shown in Fig. 3. To make the description clearer, we changed the sentence lines 479-480 to "We further plotted the OFP of different VOCs categories at various altitudes, including OVOCs, aromatics, alkyne, alkene, and alkane, …" .

The description in line 871 'O₃ formation is most sensitive to AVOC,' is derived from the relative incremental reactivity (RIR) of various VOC groups, including anthropogenic volatile organic compounds (AVOC), biogenic organic compounds (BVOC), oxygenated volatile organic compounds (OVOC), and the non-methane hydrocarbons (NMHC). We clarify in the revised manuscript that AVOC in this study includes both OVOCs and NMHC. As shown in Fig. 9, the combined RIR of OVOC and NMHC is nearly identical to that of AVOC alone. Additionally, it is evident that the RIR of OVOC significantly exceeds that of NMHCs. The VOC species categorized under OVOC primarily originate from anthropogenic sources, but can also originate from biogenic precursors (Wu et al., 2020; Park et al., 2013). This potential overlap suggests that the RIR of AVOC may be overestimated. Consequently, it is more accurate to conclude that the O₃ formation is most sensitive to OVOC rather than AVOC. We have added this discussion in the revised manuscript concerning this in lines 747-753:

"We note that AVOC includes both NMHC and OVOC. Figure 9 demonstrates that the aggregate RIR of OVOCs and NMHCs is nearly identical to that of AVOC alone. Recognizing that VOC species within the OVOC category are primarily originate from anthropogenic sources, but can also originate from biogenic precursors (Wu et al., 2020; Park et al., 2013), we acknowledge the possibility of an overestimated RIR for AVOC and due to this overlap."

And lines 768-772:

"given that the AVOC includes NMHC and OVOC, there is urgent need to reduce NMHC and OVOC emissions to mitigate $O_3$ pollution in this area. Additionally, it is evident that OVOCs have a substantially higher RIR than NMHC, therefore, it is more accurate to conclude that the $O_3$ formation is most sensitive to OVOC rather than AVOC."

This conclusion is consistent with the conclusion of OFP was primarily attributed to OVOCs in Sect. 3.1.2.

Accordingly, we changed the conclusion in lines 796 in the modified manuscript and Table S4:

"We further identified and presented the three VOC species with the highest OFP in NMHC and OVOC groups during different episodes and non-episodes in Table S4."

we changed the "AVOC group" to "NMHC group" in lines 798 in the modified manuscript:

"Results show that compounds such as toluene, *m/p*-xylene, and n-butane in NMHC group, formaldehyde, hydroxyacetone, and ethanol in OVOC group have identified as the most significant contributors to the total OFP in all episodes and non-episodes."

Also, we changed in lines 804-805 "Priority of these emission sources should be given to reducing AVOC and OVOC to mitigate $O_3$ pollution in the PRD area of China." to "Priority of these emission sources reducing should be given to mitigate $O_3$ pollution in the PRD area of China."

Furthermore, we modified the sentence in lines 48-50 in the introduction:

"The vertical and temporal $O_3$ formation is most sensitive to OVOC, suggesting that targeting specific VOCs for control measures is more practical and feasible at the observation site."

And the sentence in lines 878-880 in the conclusion:

"Nonetheless, throughout all episodes and non-episodes, $O_3$ formation is most sensitive to OVOC at various heights, emphasizing the urgent need to reduce emissions of these compounds and their precursors to mitigate $O_3$ pollution in this area."

References:

Park, J. H., Goldstein, A. H., Timkovsky, J., Fares, S., Weber, R., Karlik, J., and Holzinger, R.: Active atmosphere-ecosystem exchange of the vast majority of detected volatile organic compounds, Science, 341, 643–647, 10.1126/science.1235053, 2013.

Wu, C., Wang, C., Wang, S., Wang, W., Yuan, B., Qi, J., Wang, B., Wang, H., Wang, C., Song, W., Wang, X., Hu, W., Lou, S., Ye, C., Peng, Y., Wang, Z., Huangfu, Y., Xie, Y., Zhu, M., Zheng, J., Wang, X., Jiang, B., Zhang, Z., and Shao, M.: Measurement report: Important contributions of oxygenated compounds to emissions and chemistry of volatile organic compounds in urban air, Atmos. Chem. Phys., 20, 14769-14785, 10.5194/acp-20-14769-2020, 2020.